# SAPD+ : An Accelerated Stochastic Method for Nonconvex-Concave Minimax Problems

**Xuan Zhang**
Department of Industrial and Manufacturing Engineering
Pennsylvania State University
University Park, PA,USA.
xxz358@psu.edu

**Necdet Serhat Aybat**
Department of Industrial and Manufacturing Engineering
Pennsylvania State University
University Park, PA,USA.
nsa10@psu.edu

**Mert Gürbüzbalaban**
Department of Management Science and Information Systems
Rutgers University
Piscataway, NJ, USA
mg1366@rutgers.edu

## Abstract

We propose a new stochastic method SAPD+ for solving nonconvex-concave minimax problems of the form $\min \max \mathcal{L}(x,y) = f(x) + \Phi(x,y) - g(y)$, where $f, g$ are closed convex and $\Phi(x,y)$ is a smooth function that is weakly convex in $x$, (strongly) concave in $y$. For both strongly concave and merely concave settings, SAPD+ achieves the best known oracle complexities of $\mathcal{O}(L\kappa_y \epsilon^{-4})$ and $\mathcal{O}(L^3 \epsilon^{-6})$, respectively, without assuming compactness of the problem domain, where $\kappa_y$ is the condition number and $L$ is the Lipschitz constant. We also propose SAPD+ with variance reduction, which enjoys the best known oracle complexity of $\mathcal{O}(L\kappa_y^2 \epsilon^{-3})$ for weakly convex-strongly concave setting. We demonstrate the efficiency of SAPD+ on a distributionally robust learning problem with a nonconvex regularizer and also on a multi-class classification problem in deep learning.

## 1 Introduction

We consider the following saddle-point (SP) problem:

$$\min_{x \in \mathcal{X}} \max_{y \in \mathcal{Y}} \mathcal{L}(x,y) \triangleq f(x) + \Phi(x,y) - g(y), \tag{1}$$

where $\mathcal{X}$ and $\mathcal{Y}$ are, $n$ and $m$ dimensional Euclidean spaces, the function $\Phi : \mathcal{X} \times \mathcal{Y} \to \mathbb{R}$ is smooth and possibly nonconvex in $x \in \mathcal{X}$ and $\mu_y$-strongly concave in $y \in \mathcal{Y}$ for some $\mu_y \geq 0$ –with the convention that for $\mu_y = 0$, $\Phi$ is merely concave (MC) in $y$, and the functions $f$ and $g$ are closed, convex and possibly nonsmooth. In this paper, we consider a particular case of nonconvexity, i.e., we assume that $\Phi(\cdot, y)$ is weakly convex (WC) for any fixed $y \in \mathbf{dom}\, g \subset \mathcal{Y}$. Weakly convex functions constitute a rich class of non-convex functions and arise naturally in many practical settings for machine learning (ML) applications [9, 35], precise definitions will be given later in Section 2. In practice, WC assumption is widely satisfied, e.g., under smoothness –see remark 1; most of

36th Conference on Neural Information Processing Systems (NeurIPS 2022).

the work in related literature considering nonconvex-(strongly) concave SP problems provide their analyses under the premise of weak convexity. The problem (1) with $\mu_y > 0$ is called a weakly convex-strongly concave (WCSC) saddle-point problem, whereas for $\mu_y = 0$, it is called a weakly convex-merely concave (WCMC) saddle-point problem. Both problems arise frequently in many ML settings including constrained optimization of WC objectives based on Lagrangian duality [22], Generative Adversarial Networks (GAN) (where $x$ denotes the parameters of the *generator* network whereas $y$ represents the parameters of the *discriminator* network [13]), distributional robust learning with weakly convex loss functions such as those arising in deep learning [14, 35] and learning problems with non-decomposable losses [35].

There are two important settings for (1): (i) the *deterministic setting*, where the partial gradients of $\Phi$ are exactly available, (ii) the *stochastic setting*, where only stochastic estimates of the gradients are available. Although, recent years have witnessed significant advances in the deterministic setting [6, 17, 19, 23, 24, 25, 33, 36, 38]; our focus in this paper will be mainly on the *stochastic setting*, which is more relevant and more applicable to ML problems. Indeed, due to large-dimensions and the sheer size of the modern datasets, computing gradients exactly is either infeasible or impractical in ML practice, and gradients are often estimated stochastically based on mini-batches (randomly sampled subset of data points) as in the case of stochastic gradient-type algorithms.

There is a growing literature on the WCSC and WCMC problems in the stochastic setting. Several metrics for quantifying the quality of an approximate solution to (1) have been proposed in the literature. A common way to assess the performance is to define the *primal function* $\phi(\cdot) \triangleq \max_{y \in \mathcal{Y}} \mathcal{L}(\cdot, y)$ and measure the violation of first-order necessary conditions for the non-convex problem $\min_{x \in \mathcal{X}} \phi(x)$. Given the primal iterate sequence $\{x_k\}_{k \geq 0}$ of a stochastic SP algorithm and a threshold $\epsilon > 0$, a commonly used metric is the *gradient norm of the Moreau envelope* (GNME); indeed, the objective is to provide a bound $K_\epsilon$ such that $\mathbb{E}[\|\nabla \phi_\lambda(x_k)\|] \leq \epsilon$ for all $k \geq K_\epsilon$, where $\phi_\lambda$ denotes the Moreau envelope of the primal function $\phi$ –see Definitions 3, 4 and 5. Another commonly used natural metric is the *gradient norm of the primal function* $\phi(\cdot)$ [4, 17, 16, 26, 37], abbreviated as GNP, where the aim is to derive $K_\epsilon$ such that $\mathbb{E}[\|\nabla \phi(x_k)\|] \leq \epsilon$ for all $k \geq K_\epsilon$. Other metrics such as the notion of $\epsilon$-first-order Nash equilibrium (FNE) and its generalized versions also exist in the literature [32, 33].

When using any of the aforementioned metrics, the ultimate goal is to establish a bound on the oracle (sampling) complexity, i.e., $\sum_{k=0}^{K_\epsilon} b_k$, where $b_k$ denotes the batch-size for iteration $k \geq 0$. For the WCSC setting, it crucial to note that GNME, GNP and FNE metrics are all equivalent in the sense that convergence in either of them implies convergence in the other two metric for WCSC problems [23]. In this paper, for the WCSC setting, we adopt both GNME and GNP as the main performance metrics to analyze our algorithms; indeed, in Theorem 2 we show that, when the non-smooth part $f(\cdot) = 0$, we can convert a GNME guarantee to a GNP guarantee by incurring only little additional cost compared to the computational cost required for the GNME guarantee, and the overall worst-case complexity (in terms of worst-case dependency to the target accuracy $\epsilon$) remains the same for both metrics. When the non-smooth part $f(\cdot) \neq 0$, we also obtain similar guarantees and show equivalence between the metrics based on GNME and *the generalized gradient mapping*. On the other hand, for the WCMC setting, we provide our guarantees in GNME metric as $\phi$ is not necessarily differentiable for this scenario. Moreover, our work accounts for the individual effects of $L_{xx}$, $L_{xy}$, $L_{yx}$ and $L_{yy}$, i.e., the Lipschitz constants of $\nabla_x \Phi(\cdot, y)$, $\nabla_x \Phi(x, \cdot)$, $\nabla_y \Phi(x, \cdot)$ and $\nabla_y \Phi(\cdot, y)$ (see Assumption 2), respectively, instead of using the worst-case parameters $L \triangleq \max\{L_{xx}, L_{xy}, L_{yx}, L_{yy}\}$, while the majority of related work ignore the influence of these block Lipschitz constants in their analyses. We emphasize that using the worst-case parameters will lead to a theoretically conservative step sizes, and this phenomenon has been validated in the work [43].

**Contributions.** Table 1 summarizes the relevant existing work for WCSC and WCMC problems closest to our setting. More specifically, in Table 1, for the stochastic setting, we report the (oracle) complexity with respect to the GNP and GNME as the performance metrics for WCSC and WCMC problems, respectively, and the batch-size (number of data points in the mini-batches) required at every iteration. We also report whether the method is based on a variance-reduction (VR) technique. VR-based methods mentioned in Table 1 use a small batch-size $b'$ all iterations except for few, where they need a large batch-size $b \geq b'$ once in every $q$ iterations. The period $q$ is equal to the number of times small batches are sampled consecutively plus one, and it is also an algorithm parameter. Therefore, for VR-methods, we report the batch size as a triplet $(b', b, q)$. In the column "Compactness", we list whether achieving the specific complexity requires assuming compactness of the primal and/or dual domains.

| Ref. | Complexity | Compactness | VR-based | Batchsize |
|---|---|---|---|---|
| **Weakly Convex-Strongly Concave (WCSC) problems** | | | | |
| [*]Rafique *et al.* [35] | $\mathcal{O}(\epsilon^{-4}\log(\epsilon^{-1}))$ | (n, n) | ✗ | $\mathcal{O}(1)$ |
| [†]Yan *et al.* [39] | $\mathcal{O}(\epsilon^{-4}\log(\epsilon^{-1}))$ | (y, y) | ✗ | $\mathcal{O}(1)$ |
| [†]Yang *et al.* [41] | $\mathcal{O}(L\kappa_y^2\epsilon^{-4})$ | (n, n) | ✗ | $\mathcal{O}(1)$ |
| Lin *et al.* [23] | $\mathcal{O}(L\kappa_y^3\epsilon^{-4})$ | (n, y) | ✗ | $\mathcal{O}(\kappa_y\epsilon^{-2})$ |
| Bot and Böhm [4] | $\mathcal{O}(L\kappa_y^3\epsilon^{-4})$ | (n, n) | ✗ | $\mathcal{O}(\kappa_y\epsilon^{-2})$ |
| [‡]Huang *et al.* [17] | $\mathcal{O}(\kappa_y^5\mu_y^{-1}\epsilon^{-3})$ | (n, n) | ✓ | $\mathcal{O}(\kappa_y\epsilon^{-1}),\ \mathcal{O}(\kappa_y^2\epsilon^{-2}),\ \mathcal{O}(\kappa_y\epsilon^{-1})$ |
| [§]Huang *et al.* [16] | $\tilde{\mathcal{O}}(L^{1.5}\kappa_y^{3.5}\epsilon^{-3})$ | (y, y) | ✓ | $\mathcal{O}(\sqrt{\kappa_y})$ |
| Luo *et al.* [26] | $\mathcal{O}(L\kappa_y^3\epsilon^{-3})$ | (y, y) | ✓ | $\mathcal{O}(\kappa_y\epsilon^{-1}),\ \mathcal{O}(\kappa_y^2\epsilon^{-2}),\ \mathcal{O}(\kappa_y\epsilon^{-1})$ |
| Xu *et al.* [37] | $\mathcal{O}(L\kappa_y^3\epsilon^{-3})$ | (y, y) | ✓ | $\mathcal{O}(\kappa_y\epsilon^{-1}),\ \mathcal{O}(\kappa_y^2\epsilon^{-2}),\ \mathcal{O}(\kappa_y\epsilon^{-1})$ |
| SAPD+, Theorem 3 | $\mathcal{O}(L\kappa_y\epsilon^{-4})$ | (n, n) | ✗ | $\mathcal{O}(1)$ |
| SAPD+, Theorem 4 | $\mathcal{O}(L\kappa_y^2\epsilon^{-3})$ | (n, n) | ✓ | $\mathcal{O}(\kappa_y\epsilon^{-1}),\ \mathcal{O}(\kappa_y\epsilon^{-2}),\ \mathcal{O}(\epsilon^{-1})$ |
| **Weakly Convex-Merely Concave (WCMC) problems** | | | | |
| Rafique *et al.* [35] | $\mathcal{O}(L^3\epsilon^{-6}\log^3(L\epsilon^{-2}))$ | (y, y) | - | $\mathcal{O}(1)$ |
| Bot and Böhm [4] | $\mathcal{O}(L^5\epsilon^{-8})$ | (n, y) | - | $\mathcal{O}(1)$ |
| Lin *et al.* [23] | $\mathcal{O}(L^3\epsilon^{-8})$ | (n, y) | - | $\mathcal{O}(1)$ |
| SAPD+, Theorem 5 | $\mathcal{O}(L^3\epsilon^{-6})$ | (n, y) | - | $\mathcal{O}(1)$ |

Table 1: Summary of relevant work for WCSC and WCMC problems. For the column "Compactness", we use y and n to indicate when the results require compactness and when do not require it, respectively; the first argument is for primal domain and the second is for dual domain. For batchsize, we use $(b', b, q)$ format for VR-based methods to state *small batch* $(b')$, *large batch* $(b)$, and *frequency* $(q)$ employed within the algorithm.

**Table notes:** [*]For WCSC setting, [35] assumes $\Phi(\cdot, y) \triangleq c^\top(\cdot)y$ is weakly convex and $g(\cdot)$ is strongly convex. [†] In [39], $\mathcal{L} = \Phi$ and $\Phi$ need not be smooth, rather second moment of stochastic subgradients is assumed to be uniformly bounded. When $\Phi$ is $L$-smooth, $\Phi(\cdot, y)$ and $\Phi(x, \cdot)$ are $L_\Phi$-Lipschitz, the results in [39] imply $\mathcal{O}(L_\Phi^2\kappa_y^2\epsilon^{-4}\log^2(\sqrt{\kappa_y}L_\Phi/\epsilon))$ complexity. [‡,§]The complexity results reported here are different than those in [17, 16]. The issues in their proofs leading to the wrong complexity results are explained in Appendix I. The notation $\tilde{\mathcal{O}}$ ignores logarithmic factors.

To make the comparison of our results with the existing work easier, we provide the results in the table for the worst-case setting, where $\kappa_y \triangleq \frac{L}{\mu_y}$, and we report the $\epsilon$-, $\kappa_y$- and $L$-dependency of the complexity results for the existing algorithms. That being said, our results have finer granularity in terms of their dependence to the individual effects of $L_{xx}$, $L_{xy}$, $L_{yx}$ and $L_{yy}$ as we mentioned earlier.

Our contributions (also summarized in section 1) are as follows:

- We propose a new stochastic method, SAPD+, based on the inexact proximal point method (iPPM). In this framework, one inexactly solves strongly convex-strongly concave (SCSC) saddle point sub-problems using an accelerated primal-dual method, SAPD [43]. In Theorem 3, we establish an oracle complexity of $\mathcal{O}(L\kappa_y\epsilon^{-4})$ for WCSC problems, and unlike the majority of existing work we do not require compactness for neither the primal nor the dual domain. To our knowledge, our bound has the best $\kappa_y$ dependence in the literature; indeed, prior to this work, without using variance reduction, the best known complexity was $\mathcal{O}(L\kappa_y^2\epsilon^{-4})$ shown in [41]; hence, we establish a $\mathcal{O}(\kappa_y)$ improvement.
- We propose a variance-reduced version of SAPD+ in Theorem 4. For WCSC setting, SAPD+ using variance reduction achieves an oracle complexity of $\mathcal{O}(L\kappa_y^2\epsilon^{-3})$ –this bound has the best $\epsilon$-dependency in the literature to our knowledge, and among all the methods with the $\mathcal{O}(\epsilon^{-3})$ complexity, our approach has the best condition number, $\kappa_y$, dependency; indeed, prior to this work, the best known complexity was $\mathcal{O}(L\kappa_y^3\epsilon^{-3})$; hence, we establish $\mathcal{O}(\kappa_y)$ factor improvement.
- For the WCMC case, our proposed algorithm SAPD+ results in $\mathcal{O}(L^3\epsilon^{-6})$ complexity, which is the best to our knowledge, improving the best known complexity by $\log^3(L/\epsilon^2)$ factor.
- Finally, we demonstrate the efficiency of SAPD+ on a distributionally robust learning problem and also on a (worst-case) multi-class classification problem in deep learning.

**Notation.** Throughout the paper, $\|\cdot\|$ denotes the Euclidean norm. Given $f: \mathbb{R}^n \to \mathbb{R} \cup \{\infty\}$ a closed convex function, $\mathbf{prox}_{\lambda f}(x) \triangleq \operatorname{argmin}_w f(w) + \frac{1}{2\lambda}\|w - x\|^2$ denotes the proximal map of $f$. Given random $\omega$, let $\tilde{\nabla}_x\Phi(x, y; \omega)$ and $\tilde{\nabla}_y\Phi(x, y; \omega)$ denote unbiased estimators of $\nabla\Phi_x(x, y)$ and $\nabla\Phi_y(x, y)$. Moreover, given a random mini-batch $\mathcal{B} = \{\omega_i\}_{i=1}^b$, we let $\tilde{\nabla}_x\Phi_\mathcal{B}(x, y) \triangleq \frac{1}{b}\sum_{i=1}^b \tilde{\nabla}_x\Phi(x, y; \omega_i)$ to denote the stochastic gradient estimate based on the batch $\mathcal{B}$, and we define $\tilde{\nabla}_y\Phi_\mathcal{B}(\cdot, \cdot)$ similarly.

## 2 Preliminaries

We start with describing the notion of weak convexity.

**Definition 1.** $h : \mathbb{R}^d \to \mathbb{R} \cup \{+\infty\}$ *is $\gamma$-weakly convex if $x \mapsto h(x) + \frac{\gamma}{2}\|x\|^2$ is convex.*

**Definition 2.** *A differentiable function $h : \mathbb{R}^d \to \mathbb{R} \cup \{+\infty\}$ is L-smooth if $\exists L > 0$ such that for $\forall x, x' \in \operatorname{\mathbf{dom}} h$, $\|\nabla h(x) - \nabla h(x')\| \leq L\|x - x'\|$.*

**Remark 1.** *If a function is L-smooth, then it is also L-weakly convex.*

Remark 1 shows that weak convexity is a rich class containing the class of smooth functions. In the rest of the paper, we consider the SP problem in (1). Next, we introduce our assumptions.

**Assumption 1.** $f : \mathcal{X} \to \mathbb{R} \cup \{+\infty\}$ *and $g : \mathcal{Y} \to \mathbb{R} \cup \{+\infty\}$ are proper, closed, convex functions. Let $\Phi : \mathcal{X} \times \mathcal{Y} \to \mathbb{R}$ be such that (i) for any $y \in \operatorname{\mathbf{dom}} g \subset \mathcal{Y}$, $\Phi(\cdot, y)$ is $\gamma$-weakly convex and bounded from below; (ii) for any $x \in \operatorname{\mathbf{dom}} f \subset \mathcal{X}$, $\Phi(x, \cdot)$ is $\mu_y$-strongly concave for some $\mu_y \geq 0$; (iii) $\Phi$ is differentiable on an open set containing $\operatorname{\mathbf{dom}} f \times \operatorname{\mathbf{dom}} g$.*

**Assumption 2.** *There exist $L_{xx}, L_{yy} \geq 0$, $L_{xy}, L_{yx} > 0$ such that $\|\nabla_x \Phi(x, y) - \nabla_x \Phi(\bar{x}, \bar{y})\| \leq L_{xx}\|x - \bar{x}\| + L_{xy}\|y - \bar{y}\|$, and $\|\nabla_y \Phi(x, y) - \nabla_y \Phi(\bar{x}, \bar{y})\| \leq L_{yx}\|x - \bar{x}\| + L_{yy}\|y - \bar{y}\|$ for all $x, \bar{x} \in \operatorname{\mathbf{dom}} f \subset \mathcal{X}$, and $y, \bar{y} \in \operatorname{\mathbf{dom}} g \subset \mathcal{Y}$.*

Assumption 1 allows non-convexity in $x$ while requiring (strong) concavity in the $y$ variable. Assumption 2 is standard in the analysis of first-order methods for solving SP problems. It should be noticed that when $L_{yx} = L_{xy} = 0$, the problem in (1) can be solved separately for the primal and dual variables; hence, it is natural to assume $L_{yx}, L_{xy} > 0$.

Suppose that we implement SAPD, stated in Algorithm 1, on the SCSC problem

$$\min_{x \in \mathcal{X}} \max_{y \in \mathcal{Y}} \mathcal{L}(x, y) + \frac{\mu_x + \gamma}{2}\|x - x_0\|^2 \tag{2}$$

for some given $\mu_x > 0$ and $x_0 \in \mathcal{X}$ –strong convexity follows from $\mathcal{L}(\cdot, y)$ being $\gamma$-weakly convex.

We make the following assumption on the statistical nature of the gradient noise as in, e.g., [5, 11, 43].

**Assumption 3.** *Given arbitrary $x_0 \in \mathcal{X}$ and $\mu_x > 0$, let $\{x_k, y_k\}$ sequence be generated by SAPD, stated in Algorithm 1, running on (2). There exist $\delta_x, \delta_y \geq 0$ such that for all $k \geq 0$, the stochastic gradients $\tilde{\nabla}_x \Phi(x_k, y_{k+1}; \omega_k^x)$, $\tilde{\nabla}_y \Phi(x_k, y_k; \omega_k^y)$ and random sequences $\{\omega_k^x\}_k$, $\{\omega_k^y\}_k$ satisfy the conditions:*

---

**Algorithm 1** SAPD Algorithm
1: **Input:** $\tau, \sigma, \theta, \mu_x, x_0, y_0, N$
2: $\bar{\Phi}(x, y) \leftarrow \Phi(x, y) + \frac{\mu_x + \gamma}{2}\|x - x_0\|^2$
3: $\tilde{q}_0 \leftarrow 0$
4: **for** $k = 0, 1, 2, ..., N$ **do**
5: $\quad \tilde{s}_k \leftarrow \tilde{\nabla}_y \Phi(x_k, y_k; \omega_k^y) + \theta \tilde{q}_k$
6: $\quad y_{k+1} \leftarrow \operatorname{\mathbf{prox}}_{\sigma g}(y_k + \sigma \tilde{s}_k)$
7: $\quad x_{k+1} \leftarrow \operatorname{\mathbf{prox}}_{\tau f}(x_k - \tau \tilde{\nabla}_x \bar{\Phi}(x_k, y_{k+1}; \omega_k^x))$
8: $\quad \tilde{q}_{k+1} \leftarrow \tilde{\nabla}_y \Phi(x_{k+1}, y_{k+1}; \omega_{k+1}^y) - \tilde{\nabla}_y \Phi(x_k, y_k; \omega_k^y)$
9: **end for**
10: **Output:** $(\bar{x}_N, \bar{y}_N) = \frac{1}{N}\sum_{k=0}^{N-1}(x_{k+1}, y_{k+1})$

---

*(i)* $\mathbb{E}[\tilde{\nabla}_x \Phi(x_k, y_{k+1}; \omega_k^x)|x_k, y_{k+1}] = \nabla_x \Phi(x_k, y_{k+1})$;

*(ii)* $\mathbb{E}[\tilde{\nabla}_y \Phi(x_k, y_k; \omega_k^y)|x_k, y_k] = \nabla_y \Phi(x_k, y_k)$;

*(iii)* $\mathbb{E}[\|\tilde{\nabla}_x \Phi(x_k, y_{k+1}; \omega_k^x) - \nabla_x \Phi(x_k, y_{k+1})\|^2|x_k, y_{k+1}] \leq \delta_x^2$;

*(iv)* $\mathbb{E}[\|\tilde{\nabla}_y \Phi(x_k, y_k; \omega_k^y) - \nabla_y \Phi(x_k, y_k)\|^2|x_k, y_k] \leq \delta_y^2$.

Assumption 3 says that the gradient noise conditioned on the iterates is unbiased with a finite variance[1]. Such assumptions are common in the literature, e.g., [5, 11, 43], and are satisfied when gradients are estimated from randomly sampled data points with replacement.

For WCSC minimax problems, a commonly adopted definition for $\epsilon$-stationary is based on Moreau envelope, e.g., see [23, 39]. It is inspired by Davis and Drusvyatskiy's work [9] for solving weakly convex minimization problems. For the sake of completeness, we briefly review this idea below.

**Definition 3.** *Let $\phi : \mathbb{R}^d \to \mathbb{R} \cup \{+\infty\}$ be $\gamma$-weakly convex. Then, for any $\lambda \in (0, \gamma^{-1})$, Moreau envelope of $\phi$ is defined as $\phi_\lambda : \mathbb{R}^d \to \mathbb{R}$ such that $\phi_\lambda(x) \triangleq \min_{w \in \mathcal{X}} \phi(w) + \frac{1}{2\lambda}\|w - x\|^2$.*

---

[1]When we run SAPD, stated in Algorithm 1, on (2), we use the convention that $\tilde{\nabla}_x \bar{\Phi}(x_k, y_{k+1}; \omega_k^x) \triangleq \tilde{\nabla}_x \Phi(x_k, y_{k+1}; \omega_k^x) + (\mu_x + \gamma)(x_k - x_0)$.

**Lemma 1.** *Let $\phi : \mathbb{R}^d \to \mathbb{R} \cup \{+\infty\}$ be a $\gamma$-weakly convex function. For any given $\lambda \in (0, \gamma^{-1})$, $\phi_\lambda(\cdot)$ is well-defined on $\mathcal{X}$. Moreover, $\nabla \phi_\lambda(x) = \lambda^{-1}(x - \mathbf{prox}_{\lambda\phi}(x))$ for $x \in \mathcal{X}$; hence, $\phi_\lambda$ is $\lambda^{-1}$-smooth, where $\mathbf{prox}_{\lambda\phi}(x) \triangleq \operatorname{argmin}_{w \in \mathcal{X}} \{\phi(w) + \frac{1}{2\lambda}\|w - x\|^2\}$.*

**Definition 4.** *Under Assumption 1, let $\phi, \phi^s : \mathbb{R}^d \to \mathbb{R} \cup \{+\infty\}$ such that $\phi(x) \triangleq \max_{y \in \mathcal{Y}} \mathcal{L}(x, y)$ and $\phi^s(x) = \phi(x) - f(x)$ for $x \in \mathbf{dom}\, f$, i.e., $\phi^s(x) \triangleq \max_{y \in \mathcal{Y}} \Phi(x, y) - g(y)$ for $x \in \mathbf{dom}\, f$.*

**Remark 2.** *Under Assumption 1, since $\Phi(\cdot, y)$ is $\gamma$-weakly convex for any $y \in \mathbf{dom}\, g$, $\phi^s$ is $\gamma$-weakly convex[2]; hence, $\phi$ is also $\gamma$-weakly convex. Note that*

$$\mathbf{prox}_{\lambda\phi}(x) = \operatorname*{argmin}_{w \in \mathcal{X}} \{\phi(w) + \tfrac{1}{2\lambda}\|w - x\|^2\} = \operatorname*{argmin}_{w \in \mathcal{X}} \max_{y \in \mathcal{Y}} \mathcal{L}(w, y) + \tfrac{1}{2\lambda}\|w - x\|^2. \quad (3)$$

*Furthermore, when $\mu_y > 0$, $\phi^s$ is differentiable on $\mathbf{dom}\, f$.*

In the following definition, we introduce the notion of $\epsilon$-stationary with respect to the GNME metric.

**Definition 5.** *A point $x_\epsilon$ is an $\epsilon$-stationary point of a $\gamma$-weakly convex function $\phi$ if $\|\nabla\phi_\lambda(x_\epsilon)\| \leq \epsilon$ for some $\lambda \in (0, \gamma^{-1})$. If $\epsilon = 0$, then $x_\epsilon$ is a stationary point of $\phi$.*

Thus, from Lemma 1, computing an $\epsilon$-stationary point $x_\epsilon$ for $\phi$ is equivalent to searching for $x_\epsilon$ such that $\|x_\epsilon - \mathbf{prox}_{\lambda\phi}(x_\epsilon)\|$ is small. Recall that for any $\lambda \in (0, \gamma^{-1})$, $\mathbf{prox}_{\lambda\phi}(x)$ is well-defined and unique. We also observe from (3) that $\mathbf{prox}_{\lambda\phi}(\cdot)$ computation is indeed an SCSC SP problem. To compute $x_\epsilon$ such that $\|x_\epsilon - \mathbf{prox}_{\lambda\phi}(x_\epsilon)\|$ is small, it is natural to consider the iPPM algorithm – e.g., see [18]. A generic iPPM generates $\{x_0^t\}_{t \geq 0}$ such that $x_0^{t+1} \approx \mathbf{prox}_{\lambda\phi}(x_0^t)$, i.e., proximal steps are "inexactly" computed for $t \geq 0$, starting from an arbitrary given point $x_0^0 \in \mathcal{X}$.

In the next section, we describe the proposed `SAPD+` method, an iPPM algorithm employing `SAPD` to *inexactly* solve the SCSC subproblems arising in the iPPM iterations.

## 3 The proposed algorithm `SAPD+` and its analysis

The convergence and robustness properties of `SAPD` for SCSC SP problems are analyzed in [43]. For the WCSC SP problems, as we explained in the previous section, the main idea is to apply the iPPM framework as stated in `SAPD+` (see Algorithm 2) which requires successively solving SCSC SP problems. In the rest, the counter for iPPM outer iterations is denoted with $t \in \mathbb{Z}_+$. At each outer iteration $t \geq 1$, we inexactly compute the prox map, i.e., $x_0^{t+1} \approx \mathbf{prox}_{\lambda\phi}(x_0^t)$, which is well-defined for $\lambda \in (0, \gamma^{-1})$; hence, to derive our preliminary results, we fix $\lambda = (\mu_x + \gamma)^{-1}$ for some given $\mu_x > 0$ – thus, $\mathcal{L}(x, y) + \frac{\mu_x + \gamma}{2}\|x - x_0^t\|^2$ is SCSC in $(x, y)$ with moduli $(\mu_x, \mu_y)$ and has a unique saddle point. Consider the following SCSC SP problem:

$$\min_{x \in \mathcal{X}} \max_{y \in \mathcal{Y}} \mathcal{L}^t(x, y) \triangleq f(x) + \Phi^t(x, y) - g(y), \quad \text{where } \Phi^t(x, y) \triangleq \Phi(x, y) + \frac{\mu_x + \gamma}{2}\|x - x_0^t\|^2. \quad (4)$$

We will construct $\{x_0^t\}_{t=1}^T \subset \mathbf{dom}\, f$ by *inexactly* solving (4) at each outer iteration $t \in \mathbb{Z}_+$ through running `SAPD` for $N_t \in \mathbb{Z}_+$ iterations –we will specify $N_t \in \mathbb{Z}_+$ later. Next, we briefly explain the main step of `SAPD+` with `VR-flag=false`. The statement in line 4 of Algorithm 2 means that $(x_0^{t+1}, y_0^{t+1})$ is generated using `SAPD`, where is dispalyed in Algorithm 1 –indeed, `SAPD` is run on (4) for $N_t$ iterations with `SAPD` parameters $(\tau, \sigma, \theta)$ and starting from the initial point $(x_0^t, y_0^t)$. To analyze the convergence of `SAPD+`, we first define the gap function $\mathcal{G}^t$ for $t$-th `SAPD+` iteration:

---

**Algorithm 2** `SAPD+` Algorithm

1: **Input:** $\{\tau, \sigma, \theta, \mu_x\}$, $(x_0^0, y_0^0) \in \mathcal{X} \times \mathcal{Y}$, $\{N_t\}_{t \geq 0} \in \mathbb{Z}^+$
2: **for** $t = 0, 1, 2, ..., T$ **do**
3:     **if** `VR-flag == false` **then**
4:         $(x_0^{t+1}, y_0^{t+1}) \leftarrow \texttt{SAPD}(\tau, \sigma, \theta, \mu_x, x_0^t, y_0^t, N_t)$
5:     **else**
6:         $(x_0^{t+1}, y_0^{t+1}) \leftarrow \texttt{VR-SAPD}(\tau, \sigma, \theta, \mu_x, x_0^t, y_0^t, N_t)$
7:     **end if**
8: **end for**

---

$$\mathcal{G}^t(x, y) \triangleq \max_{y' \in \mathcal{Y}} \mathcal{L}^t(x, y') - \min_{x' \in \mathcal{X}} \mathcal{L}^t(x', y). \quad (5)$$

Recall that $\mathcal{L}^t$ is an SCSC function; therefore, *i)* it has a unique saddle point denoted by $(x_*^t, y_*^t)$, and it is important to note that $x_*^t = \mathbf{prox}_{\lambda\phi}(x_0^t)$ for $\phi(x) = \max_{y \in \mathcal{Y}} \mathcal{L}(x, y)$ and $\lambda = (\gamma + \mu_x)^{-1}$; *ii)*

---

[2]One can argue that $\phi^s(\cdot) + \frac{\gamma}{2}\|\cdot\|^2$ is a pointwise supremum of convex functions.

for any $(x, y) \in \mathbf{dom} f \times \mathbf{dom} g$, the following quantities are well-defined:

$$x_*^t(y) \triangleq \underset{x' \in \mathcal{X}}{\operatorname{argmin}} \mathcal{L}^t(x', y), \quad y_*(x) \triangleq \underset{y' \in \mathcal{Y}}{\operatorname{argmax}} \mathcal{L}^t(x, y') = \underset{y' \in \mathcal{Y}}{\operatorname{argmax}} \mathcal{L}(x, y'). \tag{6}$$

Thus, it follows that $\mathcal{G}^t(x, y) = \mathcal{L}^t(x, y_*(x)) - \mathcal{L}^t(x_*^t(y), y)$. Moreover, for $(x, y) \in \mathbf{dom} f \times \mathbf{dom} g$, we also define $\mathcal{G}(x, y) \triangleq \sup_{y' \in \mathcal{Y}} \mathcal{L}(x, y') - \inf_{x' \in \mathcal{X}} \mathcal{L}(x', y)$. Assumption 1 ensures that $\mathcal{G}$ is well defined.

Next, we first provide our oracle complexity in the GNME metric under the compactness assumption of the primal-dual domains; later, in section 3.1, we show that under a particular subdifferentiability assumption compactness requirement can be avoided.

**Assumption 4.** $\mathbf{dom} f$ and $\mathbf{dom} g$ are compact sets.

**Theorem 1.** *Suppose Assumptions 1, 2, 3, and 4 hold. Let $\mu_x = \gamma$, $\theta = 1$, $\tau, \sigma$ and $N$ be chosen as*

$$N = 33 \max\{\tfrac{4}{\gamma \tau}, \tfrac{8}{\mu_y \sigma}\}, \quad \tau = \min\{\tfrac{1}{L_{yx} + L_{xx} + 2\gamma}, \tfrac{1}{L_{xy}}, \tfrac{1}{480\gamma} \cdot \tfrac{\epsilon^2}{\delta_x^2}\}, \quad \sigma = \min\{\tfrac{1}{L_{yx} + 2L_{yy}}, \tfrac{1}{4512\gamma} \cdot \tfrac{\epsilon^2}{\delta_y^2}\}. \tag{7}$$

*Then, for any $\epsilon > 0$, when* `VR-flag=false`*, * `SAPD+` *guarantees $\epsilon$-stationary, $\min_{t=0,\dots,T} \mathbb{E}\left[\|\nabla \phi_\lambda(x_0^t)\|\right] \leq \epsilon$, for $T \geq 96 \mathcal{G}(x_0^0, y_0^0) \cdot \frac{\gamma}{\epsilon^2} + 1$, which requires $C_\epsilon$ stochastic first-order oracle calls in total where*

$$C_\epsilon = \mathcal{O}\Big(\Big(\tfrac{\max\{L_{xx}, L_{yx}, L_{xy}\}}{\gamma} + \tfrac{\max\{L_{yy}, L_{yx}\}}{\mu_y}\Big)\gamma \cdot \epsilon^{-2} + \Big(\tfrac{\delta_x^2}{\gamma} + \tfrac{\delta_y^2}{\mu_y}\Big)\gamma^2 \cdot \epsilon^{-4}\Big)\mathcal{G}(x_0^0, y_0^0).$$

*Proof.* See appendix A for the proof. $\qquad\qquad\square$

**Remark 3.** *Since $\mathbb{E}\left[\min_{t=0,\dots,T} \|\nabla\phi_\lambda(x_0^t)\|\right] \leq \min_{t=0,\dots,T} \mathbb{E}\left[\|\nabla\phi_\lambda(x_0^t)\|\right]$, the guarantees given in Theorem 1 also hold for achieving $\mathbb{E}\left[\min_{t=0,\dots,T} \|\nabla\phi_\lambda(x_0^t)\|\right] \leq \epsilon$.*

**Remark 4.** *For any $y \in \mathbf{dom} g$, since $\Phi(\cdot, y)$ $L_{xx}$-smooth, it is necessarily $L_{xx}$-weakly convex; hence, $\gamma \leq L_{xx}$. To get a worst-case complexity, let*

$$L \triangleq \max\{L_{xy}, L_{yx}, L_{xx}, L_{yy}\}, \quad \kappa_y \triangleq L/\mu_y, \quad \delta \triangleq \max\{\delta_x, \delta_y\}, \quad \gamma = L. \tag{8}$$

*Our oracle complexity $C_\epsilon$ in Theorem 1 can be simplified as $C_\epsilon = \mathcal{O}\left(\max\{1, \tfrac{\delta^2}{\epsilon^2}\}\tfrac{\kappa_y L \mathcal{G}(x_0^0, y_0^0)}{\epsilon^2}\right)$.*

*In fact, Li et al. [21] (see also [42]) provide a lower complexity bound for a class of first-order stochastic algorithms that do not use variance reduction. The lower bound for finding $\epsilon$-stationary points of smooth WCSC problems in GNP metric is $\Omega(L\Delta_\phi(\sqrt{\kappa_y}\epsilon^{-2} + \kappa_y^{\frac{1}{3}}\epsilon^{-4}))$, where $\Delta_\phi \triangleq \phi(x_0) - \min_{x \in \mathcal{X}} \phi(x)$ and $x_0$ is an arbitrary initial point.*

Consider $\phi = f + \phi^s$ as given in definition 4. For $\lambda > 0$, the map $G_\lambda : \mathbb{R}^d \to \mathbb{R}^d$ defined as

$$G_\lambda(\tilde{x}) \triangleq \frac{1}{\lambda}[\tilde{x} - \mathbf{prox}_{\lambda f}(\tilde{x} - \lambda\nabla\phi^s(\tilde{x}))] \tag{9}$$

is called the *generalized gradient mapping* and its norm is frequently used in optimization for assessing stationarity (see e.g. [10]). Theorem 1 provides guarantees in the GNME metric. Theorem 2 shows that given $x_\epsilon$, an $\epsilon$-stationary point in GNME metric (see definition 5) in expectation, we can generate $\tilde{x}$ such that $\mathbb{E}[\|G_\lambda(\tilde{x})\|] \leq \epsilon$ for some $\lambda > 0$, i.e., an $\epsilon$-stationary point in *generalized gradient mapping* metric, within $\tilde{\mathcal{O}}(1/\epsilon^2)$ `SAPD` iterations. Indeed, when $f(\cdot) = 0$, this metric and the GNP metric are the same.

**Theorem 2.** *Suppose Assumptions 1, 2, 3 hold, and $x_\epsilon$, an $\epsilon$-stationary point for the $\gamma$-weakly convex function $\phi(\cdot) = \max_{y \in \mathcal{Y}} \mathcal{L}(\cdot, y)$ in expectation, i.e., $\mathbb{E}[\|\nabla\phi_\lambda(x_\epsilon)\|] \leq \frac{\epsilon}{2}$ for some fixed $\lambda \in (0, \gamma^{-1})$ is given. Then, there exists some $\tau, \sigma, \theta$ – see eq. (35) in appendix B, such that initialized from $x_\epsilon$,* `SAPD`*, stated in Algorithm 1, can generate $\tilde{x}$ such that $\mathbb{E}[\|G_\lambda(\tilde{x})\|] \leq \epsilon$ within $\tilde{\mathcal{O}}(\frac{1}{\epsilon^2})$ stochastic first-order oracle calls, where $\phi^s(\cdot) = \max_{y \in \mathcal{Y}} \Phi(\cdot, y) - g(y)$ so that $\phi = f + \phi^s$ as in Definition 4.*

*Proof.* See appendix B for the proof. $\qquad\qquad\square$

**Remark 5.** *Based on Remark 3, the random vector $x_\varepsilon$ in Theorem 2 can be chosen as $x_0^{t_*}$ where $t_* \triangleq \operatorname{argmin}_{0 \leq t \leq T} \|\nabla\phi_\lambda(x_0^t)\|$. However, since $t_*$ can not be computed in practice, we provide an alternative method in the appendix to generate a point $x_\epsilon$ such that $\mathbb{E}[\|\nabla\phi_\lambda(x_\epsilon)\|] \leq \epsilon$ within $\tilde{\mathcal{O}}\left(\frac{L\kappa_y \mathcal{G}(x_0^0, y_0^0)}{\epsilon^2} + \frac{L\kappa_y \delta^2 \mathcal{G}(x_0^0, y_0^0)}{\epsilon^4}\right)$ stochastic first-order oracle calls – see Theorem 7 in appendix D.*

### 3.1 Relaxing the compactness assumption

In Theorem 1, we assume that $\mathbf{dom}\, f$ and $\mathbf{dom}\, g$ are compact sets, e.g., $f(\cdot) = \mathbb{1}_X(\cdot)$ and $g(\cdot) = \mathbb{1}_Y(\cdot)$, where $X \subset \mathcal{X}$ and $Y \subset \mathcal{Y}$ are compact convex sets. In this section, we show that SAPD+ can also handle unbounded domains under the following assumption.

**Assumption 5.** *For $f$ and $g$ closed convex, suppose $\exists B_f,\ B_g > 0$ such that $\inf\{\|s_f\| :\ s_f \in \partial f(x)\} \leq B_f$ for all $x \in \mathbf{dom}\, f$ and $\inf\{\|s_g\| :\ s_g \in \partial g(y)\} \leq B_g$ for all $y \in \mathbf{dom}\, g$.*

**Remark 6.** *Assumption 5 holds when $f$ is an indicator function of a closed convex set (not necessarily bounded) or for $f : \mathbb{R}^d \to \mathbb{R} \cup \{+\infty\}$ such that $\mathbf{dom}\, f$ is open and $f$ is Lipschitz. Two important examples for this scenario are: (i) $f(\cdot) = 0$, (ii) $f$ is a norm, e.g., $\ell_1$-, $\ell_2$-, or the Nuclear norms.*

The existing work based on iPPM framework either require compactness, e.g., [39], or some special structure on $\mathcal{L}$, e.g., [35]. This is also true for VR-based methods, e.g.,[16, 26, 37]. To our knowledge, ours is the first one to overcome this difficulty and strictly improve the best known complexity bound for the WCSC setting without compactness assumption; moreover, the same idea also works simultaneously with a variance reduction technique that will be discussed later (see section 4). Finally, the same trick for removing compactness assumption for the WCSC setting also helps removing the compactness assumption for the primal domain in WCMC setting and we still improve the best known complexity for this setting as well (see section 5).

**Remark 7.** *In [23], when $f = g = 0$, boundedness of dual space is required while Assumption 5 is a weaker requirement. Furthermore, based on the discussion with the authors of [39], compactness of the domain is needed for their proof to hold. In [17], the sub-level set $\{x : \phi(x) + f(x) \leq \alpha\}$ is required to be compact for all $\alpha > 0$. There are simple convex functions that do not satisfy this condition such as $f(x) = \max\{0, x\}$. Bot and B̈ohm [4] use milder assumptions than [23] without requiring compactness; however, their complexity is the same as the complexity of [23].*

**Theorem 3.** *The result of Theorem 1 continues to hold, if one replaces the compact domain assumption, i.e., Assumption 4, with Assumption 5.*

*Proof.* See appendix E for the proof. □

## 4 Variance reduction

Variance reduction techniques have been found useful for solving SCSC problems in finite sum form, e.g., [34] –see also [5] using Richardson-Romberg extrapolation in solving SCSC problems with noisy gradients to obtain improved practical performance.

In this section, we equip SAPD+ with SPIDER variance reduction technique [12], a variant of SARAH [31, 31] More precisely, for inexactly solving SCSC subproblems given in (4), we propose using VR-SAPD as stated in Algorithm 3. Note VR-SAPD employs a large batchsize of $b$ in every $q$ iterations and use small batchsizes of $b'_x$ and $b'_y$ for the rest. We prove that SAPD+ using variance reduction, i.e., with VR-flag=**true**, achieves an oracle complexity of $\mathcal{O}(L\kappa_y^2\epsilon^{-3})$; hence, we show an $\mathcal{O}(\kappa_y)$ factor improvement over the best known complexity in the literature to our knowledge.

---

**Algorithm 3** VR-SAPD Algorithm

1: **Input:** $\tau, \sigma, \theta, \mu_x, x_0, y_0, N, b, b'_x, b'_y, q$
2: $\bar{\Phi}(x, y) \leftarrow \Phi(x, y) + \frac{\mu_x + \gamma}{2}\|x - x_0\|^2$
3: Let $\mathcal{B}_0^x, \mathcal{B}_0^y$ be random mini-batch samples with $|\mathcal{B}_0^x| = |\mathcal{B}_0^y| = b$
4: $w_0 \leftarrow \tilde{\nabla}_y \Phi_{\mathcal{B}_0^y}(x_0, y_0), \quad \tilde{s}_0 \leftarrow w_0$
5: **for** $k \geq 0$ **do**
6: $\quad y_{k+1} \leftarrow \mathbf{prox}_{\sigma g}(y_k + \sigma \tilde{s}_k)$
7: $\quad$ **if** $\mathrm{mod}(k, q) == 0$ **then**
8: $\quad\quad v_k \leftarrow \tilde{\nabla}_x \bar{\Phi}_{\mathcal{B}_k^x}(x_k, y_{k+1})$
9: $\quad$ **else**
10: $\quad\quad$ Let $\mathcal{I}_k^x$ be random mini-batch sample with $|\mathcal{I}_k^x| = b'_x$
11: $\quad\quad v_k \leftarrow \tilde{\nabla}_x \bar{\Phi}_{\mathcal{I}_k^x}(x_k, y_{k+1}) - \tilde{\nabla}_x \bar{\Phi}_{\mathcal{I}_k^x}(x_{k-1}, y_k) + v_{k-1}$
12: $\quad$ **end if**
13: $\quad x_{k+1} \leftarrow \mathbf{prox}_{\tau f}(x_k - \tau v_k)$
14: $\quad$ Let $\mathcal{B}_{k+1}^x, \mathcal{B}_{k+1}^y$ be random mini-batch samples with $|\mathcal{B}_{k+1}^x| = |\mathcal{B}_{k+1}^y| = b$
15: $\quad$ **if** $\mathrm{mod}(k + 1, q) == 0$ **then**
16: $\quad\quad w_{k+1} \leftarrow \tilde{\nabla}_y \Phi_{\mathcal{B}_{k+1}^y}(x_{k+1}, y_{k+1})$
17: $\quad$ **else**
18: $\quad\quad$ Let $\mathcal{I}_{k+1}^y$ be mini-batch sample with $|\mathcal{I}_{k+1}^y| = b'_y$
19: $\quad\quad \tilde{q}_{k+1} \leftarrow \tilde{\nabla}_y \Phi_{\mathcal{I}_{k+1}^y}(x_{k+1}, y_{k+1}) - \tilde{\nabla}_y \Phi_{\mathcal{I}_{k+1}^y}(x_k, y_k)$
20: $\quad\quad w_{k+1} \leftarrow w_k + \tilde{q}_{k+1}$
21: $\quad$ **end if**
22: $\quad \tilde{s}_{k+1} \leftarrow (1 + \theta)w_{k+1} - \theta w_k$
23: **end for**
24: **Output:** $(\bar{x}_N, \bar{y}_N) = \frac{1}{N}\sum_{k=0}^{N-1}(x_{k+1}, y_{k+1})$

---

Here, we use $\tilde{\nabla}_y \Phi_{\mathcal{B}_k^y}^t(x_k, y_k)$ to represent $\frac{1}{|\mathcal{B}_k^y|}\sum_{\omega_k^i \in \mathcal{B}_k^y} \tilde{\nabla}_y \Phi(x_k, y_y; \vartheta_k^{y,i})$, where $\mathcal{B}_k^y = \{\vartheta_k^{y,i}\}_{i=1}^b$ is the mini-batch with $|\mathcal{B}_k^y| = b$ and we define $\tilde{\nabla}_x \Phi_{\mathcal{B}_k^x}^t(x_k, y_{k+1})$ similarly. In addition, $\mathcal{I}_k^x = \{\omega_k^{x,i}\}$ and

$\mathcal{I}_k^y = \{\omega_k^{y,i}\}$ with $|\mathcal{I}_k^x| = b_x'$ and $|\mathcal{I}_k^y| = b_y'$ denote the small mini-batches for generating $\tilde{\nabla}_y \Phi_{\mathcal{I}_k^y}^{t}(x_k, y_k)$ and $\tilde{\nabla}_x \Phi_{\mathcal{I}_k^x}^{t}(x_k, y_{k+1})$. When we run VR-SAPD on a generic subproblem as in (2), we use the convention that $\tilde{\nabla}_x \bar{\Phi}_{\mathcal{B}_k^x}(x_k, y_{k+1}) \triangleq \tilde{\nabla}_x \Phi_{\mathcal{B}_k^x}(x_k, y_{k+1}) + (\mu_x + \gamma)(x_k - x_0)$.

Throughout this section we make a continuity assumption on the stochastic first-order oracles similar to [17, 16, 26, 37].

**Assumption 6.** *$\exists L_{xx}, L_{xy}, L_{yx}, L_{yy} \geq 0$ such that $\forall x, \bar{x} \in \mathbf{dom}\, f \subset \mathcal{X}$ and $\forall y, \bar{y} \in \mathbf{dom}\, g \subset \mathcal{Y}$,*

$$
\begin{aligned}
\|\tilde{\nabla}_y \Phi(x, y; \omega) - \tilde{\nabla}_y \Phi(\bar{x}, \bar{y}; \omega)\| &\leq L_{yx}\|x - \bar{x}\| + L_{yy}\|y - \bar{y}\|, \quad w.p.\ 1, \\
\|\tilde{\nabla}_x \Phi(x, y; \omega) - \tilde{\nabla}_x \Phi(\bar{x}, \bar{y}; \omega)\| &\leq L_{xx}\|x - \bar{x}\| + L_{xy}\|y - \bar{y}\|, \quad w.p.\ 1.
\end{aligned}
\tag{10}
$$

**Assumption 7.** *Consider* SAPD+ *with* VR-flag = **true**. *We assume (i) for any $k \geq 0$, the random mini-batches $\mathcal{B}_k^x, \mathcal{B}_k^x, \mathcal{I}_k^x$ and $\mathcal{I}_k^y$ consist of independent elements, and $\mathcal{B}_x^k$ is independent from $\mathcal{B}_k^y$; (ii) for $i \in \{k - 1, k\}$ $\mathcal{B}_k^x, \mathcal{I}_k^x$ are independent of $(x_i, y_{i+1})$, and $\mathcal{B}_k^y, \mathcal{I}_k^y$ are independent of $(x_i, y_i)$.*

**Remark 8.** *For finite-sum type problems of the form $\min_x \max_y \frac{1}{n} \sum_{i=1}^n \Phi_i(x, y)$, we can set the stochastic gradient according to $\tilde{\nabla}_x \Phi(x, y; \omega) = \nabla_x \Phi_\omega(x, y)$ and $\tilde{\nabla}_y \Phi(x, y; \omega) = \nabla_y \Phi_\omega(x, y)$ where $\omega$ is uniformly drawn at random from $\{1, \ldots, n\}$. Therefore, if mini-batch samples are drawn from $\{1, \ldots, n\}$ uniformly at random with replacement; batches will be independent of the past iterates satisfying Assumption 7.*

**Theorem 4.** *Suppose Assumptions 1,3,6 and 7 hold. Moreover, either Assumption 4 or Assumption 5 holds. Let $\mu_x = \gamma$, $\theta = 1$, and $\tau, \sigma, b$ and $N$ be chosen as follows:*

$$
\tau = \left( L_{yx} + L_{xx} + 2\gamma + 2(q-1)\left(\frac{(L_{xx} + 2\gamma)^2}{\gamma b_x'} + \frac{10 L_{yx}^2}{\mu_y b_y'}\right) \right)^{-1}, \quad N = 2(1+\zeta)\max\left\{\frac{1}{\gamma\tau} - 1, \frac{1}{\mu_y \sigma}\right\},
$$

$$
\sigma = \left( 2L_{yy} + L_{yx} + 2(q-1)\left(\frac{L_{xy}^2}{\gamma b_x'} + \frac{10 L_{yy}^2}{\mu_y b_y'}\right) \right)^{-1}, \qquad b \geq \left\lceil \max\left\{\frac{144\delta_x^2}{\gamma}, 360\delta_y^2 \frac{1}{\mu_y}\right\} \frac{\gamma}{\epsilon^2} \right\rceil.
\tag{11}
$$

*For any $\epsilon > 0$ and parameters $b_x', b_y', q \in \mathbb{N}^+$, when* VR-flag = **true**, *SAPD+ guarantees $\epsilon$-stationary, $\min_{t=0,\ldots,T} \mathbb{E}\left[\|\nabla\phi_\lambda(x_0^t)\|\right] \leq \epsilon$, for $T \geq 288\mathcal{G}(x_0^0, y_0^0) \cdot \frac{\gamma}{\epsilon^2}$, which requires $T(Nb/q + N(b_x' + b_y'))$ stochastic first-order oracle calls in total, where*

$$
N = \mathcal{O}\left( \max\left\{ \frac{L_{yx} + L_{xx}}{\gamma} + \frac{q}{b_x'}\frac{L_{xx}^2}{\gamma^2} + \frac{q}{b_y'}\frac{L_{yx}^2}{\gamma\mu_y}, \quad \frac{L_{yy} + L_{yx}}{\mu_y} + \frac{q}{b_y'}\frac{L_{yy}^2}{\mu_y^2} + \frac{q}{b_x'}\frac{L_{xy}^2}{\gamma\mu_y} \right\} \right).
\tag{12}
$$

*Proof.* See appendix F for the proof. $\square$

**Remark 9.** *For any $y \in \mathbf{dom}\, g$, since $\Phi(\cdot, y)$ $L_{xx}$-smooth, it is necessarily $L_{xx}$-weakly convex; hence, $\gamma \leq L_{xx}$. To get a worst-case complexity, consider the setting in (8), and let $b_x' = b_y' = b'$. Then, Theorem 4 implies that setting $b = \mathcal{O}\left(\kappa_y \frac{\delta^2}{\epsilon^2}\right)$, $N = \mathcal{O}\left(\kappa_y + \kappa_y^2 \frac{q}{b'}\right)$, and $T = \mathcal{O}\left(\frac{L\mathcal{G}(x_0^0, y_0^0)}{\epsilon^2}\right)$ leads to $Nb/q + Nb' = \mathcal{O}\left(\kappa_y \frac{b}{q} + \kappa_y^2 \frac{b}{b'} + \kappa_y b' + \kappa_y^2 q\right)$. Thus, setting $q = \sqrt{\frac{b}{\kappa_y}}$ and $b' = \sqrt{b\kappa_y}$ leads to the oracle complexity of $T(Nb/q + Nb') = \mathcal{O}\left(\kappa_y^2 \frac{\delta}{\epsilon} \cdot \frac{L\mathcal{G}(x_0^0, y_0^0)}{\epsilon^2}\right)$.*

**Remark 10.** *The results in Theorem 4 continues to hold under a weaker form of Assumption 6 as in [26, 37], i.e., we replace eq. (10) with*

$$
\begin{aligned}
\mathbb{E}\left[\|\tilde{\nabla}_y \Phi(x, y; \omega) - \tilde{\nabla}_y \Phi(\bar{x}, \bar{y}; \omega)\|^2\right] &\leq 2L_{yx}\|x - \bar{x}\|^2 + 2L_{yy}\|y - \bar{y}\|^2, \\
\mathbb{E}\left[\|\tilde{\nabla}_x \Phi(x, y; \omega) - \tilde{\nabla}_x \Phi(\bar{x}, \bar{y}; \omega)\|^2\right] &\leq 2L_{xx}\|x - \bar{x}\|^2 + 2L_{xy}\|y - \bar{y}\|^2.
\end{aligned}
$$

## 5 Weakly convex-merely concave (WCMC) problems

In this section, we state the convergence guarantees of SAPD+ for solving WCMC problems. In particular, we will consider (1) such that $f(\cdot) = 0$ and $\mu_y = 0$, i.e., $\Phi(x, \cdot)$ is *merely* concave for all $x \in \mathcal{X}$. Instead of directly solving (1) in WCMC setting, we will solve an approximate model obtained by smoothing the primal problem in a similar spirit to the technique in [30]. More precisely, we approximate (1) with the following WCSC problem: given an arbitrary $\hat{y} \in \mathbf{dom}\, g$, consider

$$
\min_{x \in \mathcal{X}} \max_{y \in \mathcal{Y}} \hat{\mathcal{L}}(x, y) \triangleq \hat{\Phi}(x, y) - g(y), \quad \text{where} \quad \hat{\Phi}(x, y) \triangleq \Phi(x, y) - \frac{\hat{\mu}_y}{2}\|y - \hat{y}\|^2.
\tag{13}
$$

**Theorem 5.** *Under Assumptions 1, 2, 3, consider* (1) *such that* $f(\cdot) \equiv 0$, $\mu_y = 0$, *and* $\mathcal{D}_{\mathcal{Y}} \triangleq \sup_{y_1,y_2 \in \mathbf{dom}\, g} \|y_1 - y_2\| < \infty$. *When either Assumption 4 or Assumption 5 holds, for any given* $\epsilon > 0$, *SAPD+ with* `VR-flag` *= **false**, applied to* (13) *with* $\hat{\mu}_y = \Theta(\epsilon^2/(L\mathcal{D}_y^2))$, *is guaranteed to generate* $x_\epsilon \in \mathcal{X}$ *such that* $\mathbb{E}\left[\|\nabla \phi_\lambda(x_\epsilon)\|\right] \leq \epsilon$ *for* $\lambda = 1/(2\gamma)$ *within* $\mathcal{O}(L^3 \epsilon^{-6})$ *stochastic first-order oracle calls.*

*Proof.* See appendix G for the proof. $\qquad\square$

## 6   Numerical experiments

The experiments are conducted on a PC with 3.6 GHz Intel Core i7 CPU and NVIDIA RTX2070 GPU. We consider distributionally robust optimization and fair classification. In the rest, $n$ and $d$ represent the number of samples in the dataset and the dimension of each data point, respectively. In this section, SAPD+ means calling SAPD+ with VR-flag=**false**, and SAPD+VR means calling SAPD+ with VR-flag=**true**.

**Distributionally Robust Optimization (DRO).**   First, we consider nonconvex-regularized variant of DRO problem [1, 28, 20, 26, 43, 40] which arises in distributionally robust learning. Let $\{\mathbf{a}_i, b_i\}_{i=1}^n$ be the dataset where $\mathbf{a}_i \in \mathbb{R}^d$ are the features and $b_i \in \{-1, 1\}$ are labels. The DRO problem is

$$\text{(DRO):} \quad \min_{x \in \mathbb{R}^d} \max_{y \in Y} \frac{1}{n} \sum_{i=1}^n y_i \ell_i(x) + f(x) - g(y), \qquad (14)$$

where $\ell_i(x) = \log(1 + \exp(-b_i \mathbf{a}_i^\top \mathbf{x}))$ is the logistic loss, $f(x) = \eta_1 \sum_{i=1}^d \frac{\alpha x_i^2}{1+\alpha x_i^2}$ is a nonconvex regularizer [2], $g(y) = \frac{1}{2}\eta_2\|ny - \mathbf{1}\|^2$, and $Y \triangleq \{y \in \mathbb{R}_+^d : \mathbf{1}^\top y = 1\}$ – here, $\mathbf{1}$ denotes the vector with all entries equal to one. This problem can be viewed as a robust formulation of empirical risk minimization where the weights $y_i$ are allowed to deviate from $1/n$; and the aim is to minimize the worst-case empirical risk. We perform experiments on three data sets: *i*) a9a with $n = 32561$, $d = 123$; *ii*) gisette with $n = 6000$, $d = 5000$; *iii*) sido0 with $n = 12678$, $d = 4932$. The dataset sido0 is obtained from Causality Workbench[3] while the others can be downloaded from LIBSVM repository[4].

*Parameter tuning.* We set the parameters according to [40, 26, 20], i.e., , $\alpha = 10$, $\eta_1 = 10^{-3}$, $\eta_2 = 1/n^2$. We compare SAPD+ and SAPD+VR against PASGDA [4], SREDA [26], SMDA, SMDA-VR [17] algorithms. As suggested in [26], we tune the primal stepsizes of all the algorithms based on a grid-search over the set $\{10^{-3}, 10^{-2}, 10^{-1}\}$ and the ratio of the primal stepsize to dual stepsize, i.e., $\tau/\sigma$, is varied to take values from the set $\{10, 10^2, 10^3, 10^4\}$. For all variance reduction-based algorithms, i.e., for SAPD+VR, SREDA, SMDA-VR, we tune the large batch size $b \triangleq |\mathcal{B}|$ from the set $\{3000, 6000\}$, and the small batch size $b' \triangleq |I|$ from grid search over the set $\{10, 100, 200\}$. For the frequency parameter $q$, we let $q = b' = |I|$ for SAPD+VR and SMDA-VR (as suggested in [17]); for SREDA, when we set $q$ and $m$ (SREDA's inner loop iteration number) to $\mathcal{O}(n/|I|)$ as suggested in [26], we noticed that SREDA does not perform well against SAPD+VR and SMDA-VR. Therefore, to optimize the performance of SREDA further, we tune $q, m$ from a grid search over $\{10, 100, 200\}$. For methods without variance reduction, i.e., for SAPD+, SMDA and PASGDA, we also use mini-batch to estimate the gradients and tune the batch size from $\{10, 100, 200\}$ as well. For SAPD+ and SAPD+VR, we tune the momentum $\theta$ from $\{0.8, 0.85, 0.9\}$ and the inner iteration number from $N = \{10, 50, 100\}$.

*Results.* To fairly compare the performances of algorithms using different batch sizes, we plot loss against epochs in x-axis[5]. In fig. 1, we plot the average loss against the epoch number based on 30 simulations (runs). The standard deviations of the runs are also illustrated around the average in lighter color as shaded regions. We observe that SAPD+ and SAPD+VR consistently outperforms over other algorithms. For a9a, gisette, sido0 datasets, the average training accuracy of SAPD+ are $84.06\%$, $95.41\%$, $96.43\%$, and of SAPD+VR are $84.33\%$, $97.69\%$, $97.46\%$, respectively. The best performance for a9a, gisette, sido0 among all the other algorithms are $75.92\%$, $93.07\%$, $96.43\%$, respectively. More importantly, we observe that as an accelerated method, SAPD+VR enjoys fast convergence properties while still being robust to gradient noise.

---

[3]http://www.causality.inf.ethz.ch/challenge.php?page=datasets
[4]https://www.csie.ntu.edu.tw/ cjlin/libsvmtools/datasets/binary.html
[5]an epoch is completed whenever an algorithm does one pass over the whole data set through sampling mini-bathes without replacement.

Figure 1: Comparison of `SAPD+` and `SAPD+VR` against `PASGDA` [4], `SREDA` [26], `SMDA`, `SMDA-VR` [17] on real-data for solving eq. (14) with 30 times simulation.

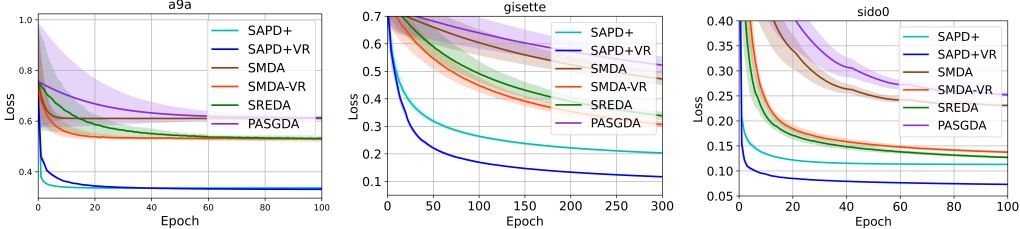

Figure 2: Comparison of `SAPD+VR` against other Variance Reduction algorithms, `SREDA` [26], `SMDA-VR` [17] on real-data for solving eq. (15) with 30 times simulation.

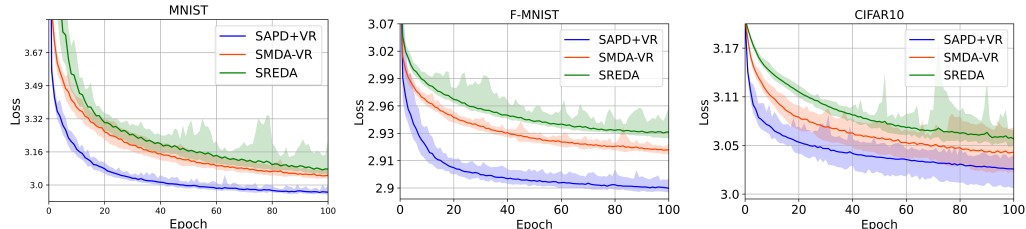

**Fair Classification.** In the context of multi-class classification, Mohri *et al.* [27] propose training a fair classifier thorough minimizing the worst-case loss over the classification categories. In the spirit of [32, 17], we adopt a nonconvex convolutional neural network (CNN) model as a classifier and set the number of categories to 3, resulting in a minimax problem of the form:

$$\min_{x \in \mathcal{X}} \max_{y \in \mathcal{Y}} \sum_{i=1}^{3} y_i \ell_i(x) - g(y), \quad s.t. \quad \sum_{i=1}^{3} y_i = 1, \ y_i \geq 0, \ \forall \ i \tag{15}$$

where $x \in \mathbb{R}^p$ represents the parameters of the CNN, and $\ell_1, \ell_2, \ell_3$ correspond to the loss of three categories whose details are given in appendix H, $g(y) = \frac{\eta}{2}\|y\|_2^2$ is a regularizer with $\eta > 0$. We train (15) on the datasets to classify: $i)$ gray-scale hand-written digits $\{0, 2, 3\}$ from `MNIST`; $ii)$ fashion images with target classes {T-shirt/top, Sandal, Ankle boot} from `F-MNIST`; $iii)$ RBG colored images with target classes {Plane, Truck, Deer} from `CIFAR10`. For both `MNIST` and `F-MNIST` $p = 43831$, $n = 18000$ and $d = 28 \times 28 \times 1$, and for `CIFAR10` $p = 61411$, $n = 15000$, and $d = 32 \times 32 \times 3$.

We let the regularization parameter $\eta = 0.1$ as suggested in [17]. We compare `SAPD+VR` against the other VR-based algorithms `SREDA` and `SMDA-VR` over 30 runs. We tune the primal stepsizes of `SAPD+VR` and `SREDA` by a grid search over the set $\{10^{-2}, 5 \times 10^{-3}, 10^{-3}\}$ and the ratio of primal to dual stepsizes, i.e., $\tau/\sigma$, is chosen from $\{10, 10^2, 5 \times 10^2, 10^3\}$. For `SMDA-VR`, the primal and dual stepsizes are $10^{-3}$ and $10^{-5}$ as suggested in [17] –we also tried stepsizes bigger than the suggested; but, it caused convergence issues in the experiments. We set the large batchsize $|\mathcal{B}| = 3000$ and the small batchsize $|\mathcal{I}| = 200$ for all algorithms and data sets; the frequency $q = 200$ is used for `SAPD+VR` and `SMDA-VR`, and we tune $q$ for `SREDA` taking values from $\{10, 50, 100, 200\}$. The momentum $\theta$ for `SAPD+VR` is tuned taking values from $\{0.8, 0.85, 0.9\}$ and inner iteration number is tuned from $N = \{10, 50, 100\}$. For `SREDA`, we tune the inner loop iteration from $\{10, 50, 100\}$. Fig. 2 shows that `SAPD+VR` outperforms the other VR-based algorithms clearly in terms of both the average loss and the standard deviation of the loss.

## 7 Conclusion

In this paper, we considered both WCSC and WCMC saddle-point problems assuming we only have an access to an unbiased stochastic first-oracle with a finite variance. This setting arises in many applications ranging from distributionally robust learning to GANs. We proposed a new method `SAPD+`, which achieves an improved complexity in terms of target accuracy $\epsilon$ for both WCSC and WCMC problems; moreover, our bound for `SAPD+` has a better dependency to the condition number $\kappa_y$ for the WCSC scenario. We also showed that our algorithm `SAPD+` can support the SPIDER variance-reduction technique. Finally, we provided numerical experiments demonstrating that `SAPD+` can achieve a state-of-the-art performance on distributionally robust learning and on multi-class classification problems arising in ML.

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
