# A The general construction used in the proof of Theorem 1

In general, the proof of Theorem 1 can be divided into two parts: (1) inner loop and outer loop convergence analysis, (2) combining these results to derive the overall complexity.

- We first study the convergence properties of Algorithm 1 for solving the SCSC subproblems in eq. (4). In Lemma 2, we provide guarantees for the inner loop iterates using the expected gap function as our metric.

- Since the convergence guarantee for the inner loop is provided in terms of $\mathcal{G}^t$, we also consider the relationship between $\mathcal{G}^t(x_0^t, y_0^t)$ and GNME, i.e., $\|\nabla_x \phi_\lambda(x_0^t)\|$. Indeed, Lemmas 3,4, and 5 allow us to translate the expected gap result of inner loops to the convergence in terms of GNME for the outer loops. In Theorem 6, we provide the convergence result in the GNME metric and state the requirements on the parameters to be able to derive the complexity bound in Theorem 1.

- In Lemma 6, we provide a particular step size rule for solving the SCSC subproblems in eq. (4), and we use this specific choice to compute the overall complexity for solving the WCSC problem eq. (1) by using SAPD+.

## A.1 The construction for the convergence analysis

Based on Lemma 1, the key step for establishing SAPD+ convergence is to bound $\|x_0^t - \mathbf{prox}_{\lambda\phi}(x_0^t)\|$, where $\phi(x) \triangleq \max_{y \in \mathcal{Y}} \mathcal{L}(x, y)$ for every $x \in \mathcal{X}$ and $\lambda = (\gamma + \mu_x)^{-1}$. To achieve this, we first give a bound on the gap function $\mathcal{G}^t$ at the $t$-th outer iteration.

**Lemma 2.** *Suppose Assumptions 1, 2, 3 hold. Given $\{N_t\}_{t \geq 0} \subset \mathbb{Z}_+$, let $\{x_0^t, y_0^t\}_{t \geq 1}$ be generated by SAPD+, stated in Algorithm 2, when VR-flag=false, initialized from $(x_0^0, y_0^0) \in \mathbf{dom} f \times \mathbf{dom} g$ and using $\tau, \sigma, \theta, \mu_x > 0$ that satisfy*

$$
\begin{pmatrix}
\mu_y & (\theta - 1)L_{yx} & (\theta - 1)L_{yy} & 0 \\
(\theta - 1)L_{yx} & \frac{1}{\tau} - L'_{xx} & 0 & -\theta L_{yx} \\
(\theta - 1)L_{yy} & 0 & \frac{1}{\sigma} - \alpha & -\theta L_{yy} \\
0 & -\theta L_{yx} & -\theta L_{yy} & \alpha
\end{pmatrix} \succeq 0
\tag{16}
$$

*for some $\alpha \in [0, \frac{1}{\sigma})$, where $L'_{xx} \triangleq L_{xx} + \mu_x + \gamma$. Then for all $t \geq 0$, it holds that*

$$
\mathbb{E}\left[\mathcal{G}^t(x_0^{t+1}, y_0^{t+1})\right] \leq \frac{M_{\tau,\sigma,\theta}}{N_t}\left(\frac{\mu_x}{4}\mathbb{E}\left[\|x_*^t(y_0^{t+1}) - x_0^t\|^2\right] + \frac{\mu_y}{4}\mathbb{E}\left[\|y_*(x_0^{t+1}) - y_0^t\|^2\right]\right) + \Xi_{\tau,\sigma,\theta},
\tag{17}
$$

*where $N_t \in \mathbb{N}^+$ and $M_{\tau,\sigma,\theta} \triangleq \max\{\frac{4}{\mu_x \tau}, \frac{4 + 4\theta}{\mu_y \sigma}\}$,*

$$
\Xi_{\tau,\sigma,\theta} \triangleq \tau\left(\Xi_{\tau,\sigma,\theta}^x + \frac{1}{2}\right)\delta_x^2 + \sigma\left(\Xi_{\tau,\sigma,\theta}^y + \frac{1 + 2\theta}{2}\right)\delta_y^2,
$$

$$
\Xi_{\tau,\sigma,\theta}^x \triangleq \left(1 + \frac{\sigma\theta(1+\theta)L_{yx}}{2}\right),
\tag{18a}
$$

$$
\Xi_{\tau,\sigma,\theta}^y \triangleq (1 + 3\theta + \sigma\theta(1+\theta)L_{yy} + \tau\sigma\theta(1+\theta)L_{yx}L_{xy})(1 + 2\theta) + \frac{\tau\theta(1+\theta)L_{yx}}{2}.
\tag{18b}
$$

*Proof.* For easier readability, we provide the proof in a separate subsection, see appendix C. □

The following lemma provides a relation between $\mathcal{G}^t(x_0^t, y_0^t)$ and $\mathcal{G}^t(x_0^{t+1}, y_0^{t+1})$.

**Lemma 3.** *Under the premise of Lemma 2 and Assumption 4, for all $t \geq 0$,*

$$
\left(1 - \frac{M_{\tau,\sigma,\theta}}{N_t}\right)\mathbb{E}[\mathcal{G}^t(x_0^{t+1}, y_0^{t+1})] \leq \frac{M_{\tau,\sigma,\theta}}{N_t}\mathbb{E}[\mathcal{G}^t(x_0^t, y_0^t)] + \Xi_{\tau,\sigma,\theta}.
$$

*Proof.* It is shown in [39, Lemma 1] that

$$
\frac{\mu_x}{4}\|x_*^t(y) - x'\|^2 + \frac{\mu_y}{4}\|y_*(x) - y'\|^2 \leq \mathcal{G}^t(x, y) + \mathcal{G}^t(x', y')
$$

holds for all $(x, y), (x', y') \in \mathbf{dom}\, f \times \mathbf{dom}\, g$. It is important to note that since $\mathbf{dom}\, f$ and $\mathbf{dom}\, g$ are compact sets, (17) implies that $\mathbb{E}[\mathcal{G}^t(x_0^{t+1}, y_0^{t+1})] < \infty$. Furthermore, since $\mathcal{G}^t(\cdot, \cdot) \geq 0$, we also have $\mathbb{E}[\mathcal{G}^t(x_0^{t+1}, y_0^{t+1})] > -\infty$; hence, $-\infty < \mathbb{E}[\mathcal{G}^t(x_0^{t+1}, y_0^{t+1})] < \infty$ for all $t \geq 0$. Then (17) and above inequality with the choice of $x = x_0^{t+1}$, $y = y_0^{t+1}$, $x' = x_0^t$, $y' = y_0^t$ together yield the desired result –one can subtract $\frac{M_{\tau,\sigma,\theta}}{N_t} \mathbb{E}[\mathcal{G}^t(x_0^{t+1}, y_0^{t+1})]$ from both sides $\mathbb{E}[\mathcal{G}^t(x_0^{t+1}, y_0^{t+1})]$ is finite. $\qquad\square$

For the sake of completeness, we state [39, Lemma 8] below, which will be used in our analysis.

**Lemma 4.** *[39, Lemma 8]. Under the premise of Lemma 2, for any $\beta_1, \beta_2 \in (0, 1)$ and $t \geq 0$,*

$$\mathcal{G}^t(x_0^{t+1}, y_0^{t+1}) \geq \left(1 - \frac{\gamma + \mu_x}{\gamma}\left(\frac{1}{\beta_1} - 1\right)\mathcal{G}_{t+1}(x_0^{t+1}, y_0^{t+1})\right) - \frac{\gamma + \mu_x}{2}\frac{\beta_1}{1 - \beta_1}\|x_0^{t+1} - x_0^t\|^2,$$

$$\mathcal{G}^t(x_0^{t+1}, y_0^{t+1}) \geq \phi(x_0^{t+1}) - \phi(x_0^t) + \frac{\gamma + \mu_x}{2}\|x_0^{t+1} - x_0^t\|^2,$$

$$\mathcal{G}^t(x_0^{t+1}, y_0^{t+1}) \geq \frac{\gamma\beta_2}{2}\|x_0^t - x_*^t\|^2 - \frac{\gamma\beta_2}{2(1 - \beta_2)}\|x_0^{t+1} - x_0^t\|^2,$$

$$\tag{19}$$

*hold w.p. 1, where $x_*^t = \mathbf{prox}_{\lambda\phi}(x_0^t)$.*

Recall that we aim to control $x_0^t - \mathbf{prox}_{\lambda\phi}(x_0^t)$ as it directly determines $\nabla\phi_\lambda(x_0^t)$, and we also have $\|x_0^t - \mathbf{prox}_{\lambda\phi}(x_0^t)\| = \|x_0^t - x_*^t\|$. Thus, in the following result, we bound $\mathbb{E}[\|x_0^t - x_*^t\|^2]$. Moreover, this result will also help us construct a telescoping sum for analyzing the convergence of $\{x_0^t\}_{t \geq 0}$ to a stationary point.

**Lemma 5.** *Under the premise of Lemma 2 and Assumption 4, for any $\beta_1, \beta_2 \in (0, 1)$, and $p_1, p_2, p_3 > 0$ such that $p_1 + p_2 + p_3 = 1$, it holds for all $t \geq 0$ that*

$$\left(1 - \frac{M_{\tau,\sigma,\theta}}{N_t}\right)\frac{\gamma p_3 \beta_2}{2}\mathbb{E}\left[\|x_0^t - x_*^t\|^2\right]$$

$$\leq \frac{M_{\tau,\sigma,\theta}}{N_t}\mathbb{E}\left[\mathcal{G}^t(x_0^t, y_0^t)\right] - \left(1 - \frac{M_{\tau,\sigma,\theta}}{N_t}\right)p_1\left(1 - \frac{\gamma + \mu_x}{\gamma}\left(\frac{1}{\beta_1} - 1\right)\right)\mathbb{E}\left[\mathcal{G}^{t+1}(x_0^{t+1}, y_0^{t+1})\right]$$

$$\tag{20}$$

$$+ \left(1 - \frac{M_{\tau,\sigma,\theta}}{N_t}\right)p_2\mathbb{E}\left[\phi(x_0^t) - \phi(x_0^{t+1})\right]$$

$$+ \frac{1}{2}\left(1 - \frac{M_{\tau,\sigma,\theta}}{N_t}\right)\left(p_1(\gamma + \mu_x)\frac{\beta_1}{1 - \beta_1} - p_2(\gamma + \mu_x) + p_3\gamma\frac{\beta_2}{1 - \beta_2}\right)\mathbb{E}\left[\|x_0^{t+1} - x_0^t\|^2\right] + \Xi_{\tau,\sigma,\theta}.$$

*Proof.* Using Lemma 4 and $\mathcal{G}^t(x_0^{t+1}, y_0^{t+1}) = (p_1 + p_2 + p_3)\mathcal{G}^t(x_0^{t+1}, y_0^{t+1})$ leads to

$$\mathbb{E}\left[\mathcal{G}^t(x_0^{t+1}, y_0^{t+1})\right] \geq -\left(p_1\frac{\gamma + \mu_x}{2}\frac{\beta_1}{1 - \beta_1} - p_2\frac{\gamma + \mu_x}{2} + p_3\frac{\gamma\beta_2}{2(1 - \beta_2)}\right)\mathbb{E}\left[\|x_0^{t+1} - x_0^t\|^2\right]$$

$$+ p_1\left(1 - \frac{\gamma + \mu_x}{\gamma}\left(\frac{1}{\beta_1} - 1\right)\right)\mathbb{E}\left[\mathcal{G}^{t+1}(x_0^{t+1}, y_0^{t+1})\right]$$

$$+ p_2\mathbb{E}\left[\phi(x_0^{t+1}) - \phi(x_0^t)\right] + p_3\frac{\gamma\beta_2}{2}\mathbb{E}\left[\|x_0^t - x_*^t\|^2\right].$$

Then, combining this inequality with Lemma 3 yields the desired result. $\qquad\square$

Finally, in the following result, we establish a preliminary convergence result for SAPD+ under compactness assumption stated in Assumption 4.

**Theorem 6.** *Under the premise of Lemma 2, given $T \in \mathbb{Z}_+$, suppose $N_t = N$ for all $t = 0, \ldots T$ for some $N \in \mathbb{Z}_+$ such that $N \geq (1 + \zeta)M_{\tau,\sigma,\theta}$ for some $\zeta > 0$, and the inequality system,*

$$\frac{M_{\tau,\sigma,\theta}}{N} - \left(1 - \frac{M_{\tau,\sigma,\theta}}{N}\right)p_1\left(1 - \frac{\gamma + \mu_x}{\gamma}\left(\frac{1}{\beta_1} - 1\right)\right) \leq 0, \tag{21a}$$

$$(\gamma + \mu_x)\left(p_1\frac{\beta_1}{1 - \beta_1} - p_2\right) + p_3\gamma\frac{\beta_2}{1 - \beta_2} \leq 0, \tag{21b}$$

*has a solution for some $\beta_1, \beta_2 \in (0, 1)$ and $p_1, p_2, p_3 > 0$ such that $p_1 + p_2 + p_3 = 1$. Then, for $\lambda = (\gamma + \mu_x)^{-1}$, under Assumption 4, the following bound holds for all $T \geq 1$:*

$$\frac{1}{T + 1}\sum_{t=0}^T \mathbb{E}\left[\|\nabla\phi_\lambda(x_0^t)\|^2\right] \leq \frac{2(1 + \zeta)(\gamma + \mu_x)^2}{\zeta\gamma p_3 \beta_2}\left(\frac{1}{T + 1}\mathcal{G}(x_0^0, y_0^0) + \Xi_{\tau,\sigma,\theta}\right). \tag{22}$$

*Proof.* Since $\mathbf{dom}\, f$ and $\mathbf{dom}\, g$ are compact sets, $\mathbb{E}[\mathcal{G}^t(x_0^t, y_0^t)] \in \mathbb{R}$ exist for $t = 0, \ldots, T$, i.e., $-\infty < \mathbb{E}[\mathcal{G}^t(x_0^t, y_0^t)] < \infty$ for all $t$. Therefore, if we sum up equation (20) from 0 to T, we get

$$
\sum_{t=0}^{T} \left(1 - \frac{M_{\tau,\sigma,\theta}}{N_t}\right) \frac{\gamma p_3 \beta_2}{2} \mathbb{E}\left[\|x_0^t - x_*^t\|^2\right]
$$

$$
\leq \frac{M_{\tau,\sigma,\theta}}{N_0} \mathcal{G}^0(x_0^0, y_0^0) - \left(1 - \frac{M_{\tau,\sigma,\theta}}{N_t}\right) p_1 \left(1 - \frac{\gamma + \mu_x}{\gamma}\left(\frac{1}{\beta_1} - 1\right)\right) \mathbb{E}\left[\mathcal{G}^{T+1}(x_0^{T+1}, y_0^{T+1})\right]
$$

$$
+ \sum_{t=0}^{T-1} \left(\frac{M_{\tau,\sigma,\theta}}{N_{t+1}} - \left(1 - \frac{M_{\tau,\sigma,\theta}}{N_t}\right) p_1 \left(1 - \frac{\gamma + \mu_x}{\gamma}\left(\frac{1}{\beta_1} - 1\right)\right)\right) \mathbb{E}\left[\mathcal{G}^{t+1}(x_0^{t+1}, y_0^{t+1})\right]
$$

$$
+ \left(1 - \frac{M_{\tau,\sigma,\theta}}{N_0}\right) p_2 \phi(x_0^0) - \left(1 - \frac{M_{\tau,\sigma,\theta}}{N_T}\right) p_2 \mathbb{E}\left[\phi(x_0^{T+1})\right] + p_2 \sum_{t=0}^{T-1} \underbrace{\left(\frac{M_{\tau,\sigma,\theta}}{N_t} - \frac{M_{\tau,\sigma,\theta}}{N_{t+1}}\right)}_{\textbf{part 1}} \mathbb{E}\left[\phi(x_0^{t+1})\right]
$$

$$
+ \sum_{t=0}^{T} \left(1 - \frac{M_{\tau,\sigma,\theta}}{N_t}\right) \left(p_1 \frac{\gamma + \mu_x}{2} \frac{\beta_1}{1 - \beta_1} - p_2 \frac{\gamma + \mu_x}{2} + p_3 \gamma \frac{\beta_2}{2(1 - \beta_2)}\right) \mathbb{E}\left[\|x_0^{t+1} - x_0^t\|^2\right]
$$

$$
+ (T + 1) \Xi_{\tau,\sigma,\theta}
$$

$$
\tag{23}
$$

Thus, using $N_t = N$ for $t = 0, \ldots, N$, it follows from the conditions in (21) that

$$
\frac{1}{T+1} \sum_{t=0}^{T} \left(1 - \frac{M_{\tau,\sigma,\theta}}{N}\right) \frac{\gamma p_3 \beta_2}{2} \mathbb{E}\left[\|x_0^t - x_*^t\|^2\right]
$$

$$
\leq \frac{1}{T+1} \frac{M_{\tau,\sigma,\theta}}{N} \mathbb{E}\left[\mathcal{G}^0(x_0^0, y_0^0)\right]
$$

$$
- \frac{1}{T+1} \left(1 - \frac{M_{\tau,\sigma,\theta}}{N}\right) p_1 \left(1 - \frac{\gamma + \mu_x}{\gamma}\left(\frac{1}{\beta_1} - 1\right)\right) \mathbb{E}\left[\mathcal{G}^{T+1}(x_0^{T+1}, y_0^{T+1})\right] \tag{24}
$$

$$
+ \frac{p_2 \left(1 - \frac{M_{\tau,\sigma,\theta}}{N}\right)}{T+1} \mathbb{E}\left[\phi(x_0^0) - \phi(x_0^{T+1})\right] + \Xi_{\tau,\sigma,\theta}
$$

$$
\leq \frac{1}{T+1} \frac{M_{\tau,\sigma,\theta}}{N} \mathcal{G}^0(x_0^0, y_0^0) + \frac{p_2 \left(1 - \frac{M_{\tau,\sigma,\theta}}{N}\right)}{T+1} \mathcal{G}(x_0^0, y_0^0) + \Xi_{\tau,\sigma,\theta},
$$

which follows from $(i)$ $\mathcal{G}_{T+1}(x_0^{T+1}, y_0^{T+1}) \geq 0$, $(ii)$ $\phi(x_0^0) - \phi(x_0^{T+1}) = \mathcal{L}(x_0^0, y_*(x_0^0)) - \mathcal{L}(x_0^{T+1}, y_*(x_0^{T+1})) \leq \mathcal{L}(x_0^0, y_*(x_0^0)) - \mathcal{L}(x_0^{T+1}, y_0^0) \leq \sup_{y' \in \mathcal{Y}} \mathcal{L}(x_0^0, y') - \inf_{x' \in \mathcal{X}} \mathcal{L}(x', y_0^0) = \mathcal{G}(x_0^0, y_0^0)$, and also from the fact that (21a) implies $\left(1 - \frac{M_{\tau,\sigma,\theta}}{N}\right) p_1 \left(1 - \frac{\gamma + \mu_x}{\gamma}\left(\frac{1}{\beta_1} - 1\right)\right) \geq 0$. Then dividing both sides by $\left(1 - \frac{M_{\tau,\sigma,\theta}}{N}\right) \frac{\gamma p_3 \beta_2}{2}$ gives us

$$
\frac{1}{T+1} \sum_{t=1}^{T} \|x_0^t - x_*^t\|^2 \tag{25}
$$

$$
\leq \frac{2}{(1 - \frac{M_{\tau,\sigma,\theta}}{N}) \gamma p_3 \beta_2} \left(\frac{1}{T+1} \frac{M_{\tau,\sigma,\theta}}{N} \mathcal{G}^0(x_0^0, y_0^0) + \frac{p_2 \left(1 - \frac{M_{\tau,\sigma,\theta}}{N}\right)}{T+1} \mathcal{G}(x_0^0, y_0^0) + \Xi_{\tau,\sigma,\theta}\right),
$$

$$
\leq \frac{2(1 + \zeta)}{\zeta \gamma p_3 \beta_2} \left(\frac{1}{T+1} \mathcal{G}(x_0^0, y_0^0) + \Xi_{\tau,\sigma,\theta}\right),
$$

where the second inequality follows from $\mathcal{G}(x_0^0, y_0^0) \geq \mathcal{G}^0(x_0^0, y_0^0)$, and for $p_2 \in (0, 1)$, we have $N \geq (1 + \zeta) M_{\tau,\sigma,\theta}$. Finally, we get the desired result using Lemma 1. $\qquad\square$

## A.2 A particular parameter choice

We employ the matrix inequality (MI) in eq. (16) to describe the admissible set of algorithm parameters that guarantee convergence of Algorithm 1, i.e., inner loop of `SAPD+` when `VR-flag` is **false**. In

this subsection, we compute a particular solution by exploiting the structure of MI in eq. (16). This particular solution is for solving the SCSC subproblems in eq. (4).

**Lemma 6.** *For any $\mu_x \geq 0$, let $L'_{xx} = L_{xx} + \gamma + \mu_x$. Suppose $\theta = 1$, and $\tau, \sigma > 0$, satisfy*

$$\tau \leq \frac{1}{L'_{xx} + L_{yx}}, \quad \sigma \leq \frac{1}{2L_{yy} + L_{yx}}. \tag{26}$$

*Then $\{\tau, \sigma, \theta, \alpha\}$ is a solution to (16) for $\alpha = L_{yx} + L_{yy}$.*

*Proof.* It follows from the choice of $\tau$ and $\sigma$ in (26) and $\theta = 1$ that a sufficient condition for (16) is given by the following smaller matrix inequality for $\alpha = L_{yx} + L_{yy}$,

$$\mathbf{0} \preceq \begin{pmatrix} \frac{1}{\tau} - L'_{xx} & 0 & -L_{yx} \\ 0 & \frac{1}{\sigma} - \alpha & -L_{yy} \\ -L_{yx} & -L_{yy} & \alpha \end{pmatrix} = \begin{pmatrix} \frac{1}{\tau} - L'_{xx} & 0 & -L_{yx} \\ 0 & \frac{1}{\sigma} - L_{yx} - L_{yy} & -L_{yy} \\ -L_{yx} & -L_{yy} & L_{yx} + L_{yy} \end{pmatrix} \triangleq M_1 + M_2,$$

where $M_1 \triangleq \begin{pmatrix} \frac{1}{\tau} - L'_{xx} & 0 & -L_{yx} \\ 0 & 0 & 0 \\ -L_{yx} & 0 & L_{yx} \end{pmatrix}$ and $M_2 \triangleq \begin{pmatrix} 0 & 0 & 0 \\ 0 & \frac{1}{\sigma} - L_{yx} - L_{yy} & -L_{yy} \\ 0 & -L_{yy} & L_{yy} \end{pmatrix}$. Therefore, the

Schur complement conditions together with eq. (26) imply $M_1 \succeq 0$ and $M_2 \succeq 0$, respectively. Thus, $M_1 + M_2 \succeq 0$. □

### A.3 Proof of Theorem 1

*Proof.* Using the results we derived in the previous two subsections, we are now ready to provide the proof of Theorem 1.

For the inner loop iterations, Lemma 6 ensures that eq. (16) holds for our $\{\tau, \sigma, \theta\}$ choice in eq. (7). For the outer loop, if we set $N$ as in eq. (7) and

$$p_1 = \frac{1}{16}, \ p_2 = \frac{19}{32}, \ p_3 = \frac{11}{32}, \ \beta_1 = \frac{4}{5}, \ \beta_2 = \frac{1}{2}, \ \zeta = 32, \tag{27}$$

all assumptions of Theorem 6 are satisfied, i.e., both the inequality system eq. (21) and $N \geq (1 + \zeta)M_{\tau,\sigma,\theta}$ hold.

Specifically, because $\mu_x = \gamma$ and $\theta = 1$, we have $M_{\tau,\sigma,\theta} = \max\{\frac{4}{\gamma\tau}, \frac{8}{\mu_y\sigma}\}$. Therefore, we know that $N \geq (1 + \zeta)M_{\tau,\sigma,\theta}$ is trivially true. Moreover, using $M_{\tau,\sigma,\theta}/N \leq (1 + \zeta)^{-1}$, it follows that eq. (21a) holds for $\mu_x = \gamma, p_1 = \frac{1}{16}$ and $\beta_1 = \frac{4}{5}$, i.e.,

$$\frac{M_{\tau,\sigma,\theta}}{N} - \left(1 - \frac{M_{\tau,\sigma,\theta}}{N}\right) p_1 \left(1 - \frac{\gamma + \mu_x}{\gamma}\left(\frac{1}{\beta_1} - 1\right)\right) = \frac{33}{32}\frac{M_{\tau,\sigma,\theta}}{N} - \frac{1}{32} \leq \frac{33}{32}\frac{1}{1 + \zeta} - \frac{1}{32} = 0.$$

Moreover, it is trivial to check that eq. (21b) holds for the parameter values given in eq. (27).

Since all assumptions of Theorem 6 are satisfied for parameters chosen as in eq. (7) and eq. (27), if we substitute eq. (27) into eq. (22), if follows that

$$\frac{1}{T+1}\sum_{t=0}^{T}\mathbb{E}\left[\|\nabla\phi_\lambda(x_0^t)\|^2\right] \leq 48\gamma\left(\frac{1}{T+1}\mathcal{G}(x_0^0, y_0^0) + \Xi_{\tau,\sigma,\theta}\right).$$

Thus, for any $\epsilon > 0$, the right side of the above inequality can be bounded by $\epsilon^2$ when

$$\frac{48\gamma}{T+1}\mathcal{G}(x_0^0, y_0^0) \leq \frac{\epsilon^2}{2}, \quad 48\gamma\Xi_{\tau,\sigma,\theta} \leq \frac{\epsilon^2}{2}. \tag{28}$$

Note that because $\Xi_{\tau,\sigma,\theta} = \tau\left(\Xi_{\tau,\sigma,\theta}^x + \frac{1}{2}\right)\delta_x^2 + \sigma\left(\Xi_{\tau,\sigma,\theta}^y + \frac{3}{2}\right)\delta_y^2$, a sufficient condition for the second inequality in eq. (28) is that

$$24\gamma\tau(1 + 2\Xi_{\tau,\sigma,\theta}^x)\delta_x^2 \leq \frac{\epsilon^2}{4}, \quad 24\gamma\sigma(3 + 2\Xi_{\tau,\sigma,\theta}^y)\delta_y^2 \leq \frac{\epsilon^2}{4}. \tag{29}$$

Moreover, recall that $\Xi^x_{\tau,\sigma,\theta}$ and $\Xi^y_{\tau,\sigma,\theta}$ are defined in Lemma 2; for $\theta = 1$, they can be simplified as follows:

$$\Xi^x_{\tau,\sigma,\theta} = 1 + \sigma L_{yx}, \quad \Xi^y_{\tau,\sigma,\theta} = 3\left(4 + 2\sigma L_{yy} + 2\tau\sigma L_{yx}L_{xy}\right) + \tau L_{yx}.$$

Because the choice of $\{\tau, \sigma\}$ in eq. (7) implies that

$$\tau L_{yx} \le 1, \quad \tau L_{xy} \le 1, \quad \sigma L_{yy} \le \frac{1}{2}, \quad \sigma L_{yx} \le 1,$$

we can upper bound $\Xi^x_{\tau,\sigma,\theta}$ and $\Xi^y_{\tau,\sigma,\theta}$ as follows:

$$\Xi^x_{\tau,\sigma,\theta} \le 2, \quad \Xi^y_{\tau,\sigma,\theta} \le 22.$$

Therefore, with the choice of $\{\tau,\sigma\}$ in eq. (7), we have a sufficient condition for eq. (29) as follows:

$$120\gamma\tau\delta_x^2 \le \frac{\epsilon^2}{4}, \qquad 1128\gamma\sigma\delta_y^2 \le \frac{\epsilon^2}{4}.$$

Indeed, the above condition is trivially satisfied by our choice of $\{\tau,\sigma\}$ given in eq. (7). Therefore, the second condition in (28), i.e., $48\gamma\Xi_{\tau,\sigma,\theta} \le \frac{\epsilon^2}{2}$, holds for the choice of $\{\tau,\sigma\}$ in eq. (7). Thus, from the first inequality in eq. (28), we get $\min_{t=0,\dots,T} \mathbb{E}[\|\nabla\phi_\lambda(x_0^t)\|^2] \le \epsilon^2$ for

$$T \ge 96\mathcal{G}(x_0^0, y_0^0) \cdot \frac{\gamma}{\epsilon^2} + 1. \tag{30}$$

Note that from Jensen's inequality, we have $(\mathbb{E}[\|\nabla\phi_\lambda(x_0^t)\|])^2 \le \mathbb{E}[\|\nabla\phi_\lambda(x_0^t)\|^2]$ for all $t = 0,\dots,T$; hence, it follows that $\min_{t=0,\dots,T} \mathbb{E}[\|\nabla\phi_\lambda(x_0^t)\|] \le \epsilon$ for all $T \in \mathbb{Z}_+$ satisfying (30). Finally, to show the complexity result, recall that $N = 33\max\{\frac{4}{\gamma\tau}, \frac{8}{\mu_y\sigma}\}$. Using the the choice of $\{\tau,\sigma\}$ in eq. (7) we derive that

$$N = \mathcal{O}\left(\frac{\max\{L_{xx}, L_{yx}, L_{xy}\}}{\gamma} + \frac{\max\{L_{yy}, L_{yx}\}}{\mu_y} + \left(\frac{\delta_x^2}{\gamma} + \frac{\delta_y^2}{\mu_y}\right)\frac{\gamma}{\epsilon^2}\right). \tag{31}$$

Moreover, since SAPD+ requires $NT$ oracle calls in total, combining (30) with (31) leads to $\mathcal{O}(\epsilon^{-4})$ bound on $C_\epsilon$ as stated in Theorem 1, which completes the proof. $\qquad\square$

## B  Proof of Theorem 2 and preliminary technical results

Suppose Assumptions 1, 2, 3 hold. Given $x_\epsilon$, an $\epsilon$-stationary point for the $\gamma$-weakly convex function $\phi(\cdot) = \max_{y\in\mathcal{Y}} \mathcal{L}(\cdot, y)$, i.e., $\mathbb{E}\left[\|\nabla\phi_\lambda(x_\epsilon)\|\right] \le \frac{\epsilon}{2}$ for some fixed $\lambda \in (0, \gamma^{-1})$. Let $\phi^s(\cdot) \triangleq \max_{y\in\mathcal{Y}} \Phi(\cdot, y) - g(y)$ so that $\phi = f + \phi^s$. In this section we show that initialized from $x_\epsilon$ and using appropriately selected step size parameters, within $\tilde{\mathcal{O}}(\frac{1}{\epsilon^2})$ stochastic first-order oracle calls, SAPD, stated in Algorithm 1, can generate $\tilde{x}$ such that $\mathbb{E}\left[\|G_\lambda(\tilde{x})\|\right] \le \epsilon$, where generalized gradient mapping $G_\lambda$ is defined in (9).

**Lemma 7.** *Suppose Assumptions 1, 2, 3 hold. Given some $(x_0, y_0) \in \mathbf{dom}\, f \times \mathbf{dom}\, g$, consider the SCSC problem in (2) for some $\mu_x > 0$. Let $\{x_k, y_k\}_{k\ge 0}$ be generated by SAPD, stated in Algorithm 1, initialized from $(x_0, y_0)$ and using $\tau, \sigma, \theta > 0$ that satisfy*

$$G \triangleq \begin{pmatrix} \frac{1}{\tau}(1 - \frac{1}{\rho}) + \frac{\mu_x}{\rho} & 0 & 0 & 0 & 0 \\ 0 & \frac{1}{\sigma}(1 - \frac{1}{\rho}) + \mu_y & (\frac{\theta}{\rho} - 1)L_{yx} & (\frac{\theta}{\rho} - 1)L_{yy} & 0 \\ 0 & (\frac{\theta}{\rho} - 1)L_{yx} & \frac{1}{\tau} - L'_{xx} & 0 & -\frac{\theta}{\rho}L_{yx} \\ 0 & (\frac{\theta}{\rho} - 1)L_{yy} & 0 & \frac{1}{\sigma} - \alpha & -\frac{\theta}{\rho}L_{yy} \\ 0 & 0 & -\frac{\theta}{\rho}L_{yx} & -\frac{\theta}{\rho}L_{yy} & \frac{\alpha}{\rho} \end{pmatrix} \succeq 0 \tag{32}$$

*for some $\alpha \in [0, \frac{1}{\sigma})$ and $\rho \in (0,1)$, where $L'_{xx} \triangleq L_{xx} + \mu_x + \gamma$. Define $\phi(x) = \max_{y\in\mathcal{Y}} \mathcal{L}(x, y)$; and let $\hat{x} = \mathbf{prox}_{\lambda\phi}(x_0)$ for $\lambda = (\mu_x + \gamma)^{-1}$ and $y_*(\hat{x}) = \operatorname{argmax}_{y\in\mathcal{Y}} \mathcal{L}(\hat{x}, y)$. Then for all $N \in \mathbb{Z}_+$, it holds that*

$$\mathbb{E}\left[\left(\frac{1}{\tau} - \mu_x\right)\|x_N - \hat{x}\|^2 + \left(\frac{1}{\sigma} - \alpha\right)\|y_N - y_*(\hat{x})\|^2\right]$$

$$\le \rho^N \left(\frac{1}{\tau}\|x_0 - \hat{x}\|^2 + \frac{1}{\sigma}\|y_0 - y_*(\hat{x})\|^2\right) + \frac{\rho}{1-\rho}\left(\tau\Xi^x_{\tau,\sigma,\theta}\delta_x^2 + \sigma\Xi^y_{\tau,\sigma,\theta}\delta_y^2\right), \tag{33}$$

*where $\Xi^x_{\tau,\sigma,\theta}$ and $\Xi^y_{\tau,\sigma,\theta}$ are defined in (18a) and (18b), respectively.*

*Proof.* For easier readability, we provide the proof in a separate subsection, see appendix C. $\quad\square$

In the following part, we will compute a particular solution by exploiting the structure of MI in eq. (32) and use this particular solution for the rest of the proof. First, in Lemma 8, we give an intermediate condition to help us construct the particular solution subsequently provided in Lemma 9 for solving the generic SCSC subproblems in eq. (2).

**Lemma 8.** *For any $\mu_x > 0$, let $L'_{xx} = L_{xx} + \gamma + \mu_x$. Suppose $\rho = \theta$, and $\tau, \sigma > 0$, $\theta \in (0,1)$ satisfy*

$$\tau \geq \frac{1-\theta}{\mu_x}, \quad \sigma \geq \frac{1-\theta}{\mu_y\theta}, \quad \frac{1}{\tau} \geq L'_{xx} + \pi_1 L_{yx}, \quad \frac{1}{\sigma} \geq \frac{\theta L_{yx}}{\pi_1} + \left(\frac{\theta}{\pi_2} + \pi_2\right) L_{yy}, \tag{34}$$

*for some $\pi_1, \pi_2 > 0$. Then $\{\tau, \sigma, \theta, \alpha\}$ is a solution to (32) for $\alpha = \frac{\theta L_{yx}}{\pi_1} + \frac{\theta L_{yy}}{\pi_2}$.*

*Proof.* It follows from the choice of $\tau$ and $\sigma$ in (34) and $\rho = \theta$ that a sufficient condition for eq. (32), i.e., for $G \succeq 0$, is given by the following smaller matrix inequality for $\alpha = \frac{\theta L_{yx}}{\pi_1} + \frac{\theta L_{yy}}{\pi_2}$,

$$\mathbf{0} \preceq \begin{pmatrix} \frac{1}{\tau} - L'_{xx} & 0 & -L_{yx} \\ 0 & \frac{1}{\sigma} - \alpha & -L_{yy} \\ -L_{yx} & -L_{yy} & \frac{\alpha}{\theta} \end{pmatrix} = \begin{pmatrix} \frac{1}{\tau} - L'_{xx} & 0 & -L_{yx} \\ 0 & \frac{1}{\sigma} - \frac{\theta L_{yx}}{\pi_1} - \frac{\theta L_{yy}}{\pi_2} & -L_{yy} \\ -L_{yx} & -L_{yy} & \frac{L_{yx}}{\pi_1} + \frac{L_{yy}}{\pi_2} \end{pmatrix} \triangleq M_1 + M_2,$$

where $M_1 \triangleq \begin{pmatrix} \frac{1}{\tau} - L'_{xx} & 0 & -L_{yx} \\ 0 & 0 & 0 \\ -L_{yx} & 0 & \frac{L_{yx}}{\pi_1} \end{pmatrix}$ and $M_2 \triangleq \begin{pmatrix} 0 & 0 & 0 \\ 0 & \frac{1}{\sigma} - \frac{\theta L_{yx}}{\pi_1} - \frac{\theta L_{yy}}{\pi_2} & -L_{yy} \\ 0 & -L_{yy} & \frac{L_{yy}}{\pi_2} \end{pmatrix}$. Therefore,

since $\pi_1, \pi_2 > 0$, the Schur complement conditions in (34), i.e., the third and the fourth inequalities, imply $M_1 \succeq 0$ and $M_2 \succeq 0$, respectively. Thus, $M_1 + M_2 \succeq 0$. $\quad\square$

Lemma 8 shows that every solution to (34) can be converted to a solution to (32). Next, based on Lemma 8, we will give another explicit parameter choice for Algorithm 1 in addition to the solution we provided earlier in Lemma 6.

**Lemma 9.** *For any $\mu_x > 0$, let $L'_{xx} = L_{xx} + \gamma + \mu_x$. For any given $\beta \in (0,1]$, let $\tau, \sigma > 0$ and $\theta \in (0,1)$ be chosen satisfying*

$$\tau = \frac{1-\theta}{\mu_x}, \quad \sigma = \frac{1-\theta}{\mu_y\theta}, \quad \theta \geq \bar{\theta}(\beta), \tag{35}$$

*where $\bar{\theta}(\beta) \triangleq \max\{\bar{\theta}_1(\beta), \bar{\theta}_2(\beta)\} \in (0,1)$ such that*

$$\bar{\theta}_1(\beta) \triangleq 1 - \frac{\beta \mu_y L'_{xx}}{2L_{yx}^2} \left(\sqrt{1 + \frac{4L_{yx}^2 \mu_x}{\beta L'^2_{xx}\mu_y}} - 1\right),$$

$$\bar{\theta}_2(\beta) \triangleq \begin{cases} 1 - \frac{(1-\beta)^2}{8} \frac{\mu_y^2}{L_{yy}^2} \left(\sqrt{1 + \frac{16L_{yy}^2}{(1-\beta)^2\mu_y^2}} - 1\right) & L_{yy} > 0 \\ 0 & L_{yy} = 0. \end{cases}$$

*Then, $\{\tau, \sigma, \theta, \alpha, \rho\}$ with $\alpha = \frac{1}{\sigma} - \sqrt{\theta}L_{yy} > 0$ and $\rho = \theta$ is a solution to (32).*

*Proof.* Consider arbitrary $\tau, \sigma, \pi_1, \pi_2 > 0$ and $\theta \in (0,1)$. By a straightforward calculation, $\{\tau, \sigma, \theta, \pi_1, \pi_2\}$ is a solution to (34) if and only if

$$\tau \geq \frac{1-\theta}{\mu_x}, \quad \sigma \geq \frac{1-\theta}{\theta\mu_y}, \quad \pi_1 \geq \frac{\sigma\theta L_{yx}}{1 - \sigma(\pi_2 + \frac{\theta}{\pi_2})L_{yy}}, \tag{36a}$$

$$\sigma(\pi_2 + \frac{\theta}{\pi_2})L_{yy} < 1, \quad \frac{1}{\tau} - L'_{xx} \geq \pi_1 L_{yx}. \tag{36b}$$

In the remainder of the proof, we fix $(\pi_1, \pi_2)$ as follows:

$$\pi_1 = \frac{\sigma\theta L_{yx}}{1 - \sigma\left(\pi_2 + \frac{\theta}{\pi_2}\right)L_{yy}}, \quad \pi_2 = \sqrt{\theta}. \tag{37}$$

Note the definition of $\bar{\theta}(\beta)$ implies that $\bar{\theta}(\beta) \in (0,1)$. Next, we show that $\theta \in [\bar{\theta}(\beta), 1)$ implies $\pi_1, \pi_2 > 0$; furthermore, we also show that $\tau, \sigma > 0$ defined as in (35) for $\theta \in [\bar{\theta}(\beta), 1)$ together with $(\pi_1, \pi_2)$ as in (37) is a solution to (36).

First, setting $\tau, \sigma$ as in (35) and $\pi_1, \pi_2$ as in (37) imply that (36a) is trivially satisfied. Next, by substituting $\{\tau, \sigma, \pi_1, \pi_2\}$, chosen as in (35) and (37), into (36b), we conclude that $\{\tau, \sigma, \theta, \pi_1, \pi_2\}$ satisfies (36) for any $\theta \in (0,1)$ such that

$$\frac{2L_{yy}}{\mu_y} \cdot \frac{1-\theta}{\sqrt{\theta}} \leq 1 - \beta, \tag{38}$$

$$\frac{\mu_x}{1-\theta} - L'_{xx} \geq (1-\theta)\frac{L_{yx}^2}{\mu_y} \cdot \left(1 - \frac{2L_{yy}}{\mu_y} \cdot \frac{1-\theta}{\sqrt{\theta}}\right)^{-1}, \tag{39}$$

for some $\beta \in (0,1]$. Clearly, a sufficient condition for (39) is

$$\frac{\mu_x}{1-\theta} - L'_{xx} \geq (1-\theta)\frac{L_{yx}^2}{\mu_y} \cdot \frac{1}{\beta}. \tag{40}$$

Note that (38) implies that $\pi_1 > 0$. We also have $\pi_2 = \sqrt{\theta} > 0$ trivially.

When $L_{yy} > 0$, given any $\beta \in (0,1)$, solving eqs. (38) and (40) for $\theta \in (0,1)$, we get the third condition in (35). Indeed, it can be checked that $\theta \in [\bar{\theta}_2(\beta), 1)$ satisfies (38) and $\theta \in [\bar{\theta}_1(\beta), 1)$ satisfies (40); thus, $\theta \in [\bar{\theta}(\beta), 1)$ satisfies (38) and (40) simultaneously. Moreover, when $L_{yy} = 0$, one does not need to solve eq. (38) as the first inequality in (36b) holds trivially; thus, the only condition on $\theta$ comes from (39) which is equivalent to (40) with $\beta = 1$. The rest follows from Lemma 8 by setting $\alpha = \frac{\theta L_{yx}}{\pi_1} + \frac{\theta L_{yy}}{\pi_2}$. Indeed, the particular choice of $(\pi_1, \pi_2)$ in (37) gives us $\alpha = \frac{1}{\sigma} - \sqrt{\theta}L_{yy}$. □

Now that we have provided a particular solution to eq. (32), we will next use this particular solution within Lemma 7 to derive an error bound customized for this choice of parameters. The following two technical results, i.e., Lemmas 10 and 11, will be used later within the proof of Theorem 2.

**Lemma 10.** *Consider $\mathcal{L}$ defined in (1). Suppose Assumptions 1, 2, 3 hold. Given arbitrary $x_0$, let $\hat{x} = \mathbf{prox}_{\lambda\phi}(x_0)$, where $\phi(\cdot) = \max_{y \in \mathcal{Y}} \mathcal{L}(\cdot, y)$ and $\lambda = (2\gamma)^{-1}$. For any given $\hat{\epsilon} > 0$, `SAPD`, displayed in Algorithm 1, can generate $\tilde{x}_* \in \mathcal{X}$ such that $\mathbb{E}\left[\|\tilde{x}_* - \hat{x}\|\right] \leq \hat{\epsilon}$ within $\tilde{\mathcal{O}}(\frac{1}{\hat{\epsilon}^2})$ stochastic first-order oracle calls.*

*Proof.* Recall that $y_*(x) = \text{argmax}_{y \in \mathcal{Y}} \mathcal{L}(x, y)$ for $x \in \mathbf{dom}\, f$. Hence, $(\hat{x}, y_*(\hat{x}))$ is the unique saddle point to the SCSC problem:

$$\min_{x \in \mathcal{X}} \max_{y \in \mathcal{Y}} \bar{\mathcal{L}}(x, y) \triangleq f(x) + \Phi(x, y) + \gamma\|x - x_0\|^2 - g(y), \tag{41}$$

which is equivalent to the SCSC problem in eq. (2) with $\mu_x = \gamma$. Let $\{x_k, y_k\}$ be the iterate sequence generated by `SAPD` running on (41), initialized from an arbitrary point $(x_0, y_0)$, with parameters $\{\tau, \sigma, \theta\}$ chosen as follows:

$$\tau = \frac{1-\theta}{\gamma}, \quad \sigma = \frac{1-\theta}{\mu_y \theta}, \quad \theta = \max\{\bar{\theta}(\beta), \hat{\theta}_1, \hat{\theta}_2\}, \tag{42}$$

for $\beta = \min\{\frac{1}{2}, \frac{\mu_y}{\gamma}, \frac{\gamma}{\mu_y}, \frac{L_{yx}}{L_{xy}}\}$, where $\bar{\theta}(\beta) \triangleq \max\{\bar{\theta}_1(\beta), \bar{\theta}_2(\beta)\} \in (0,1)$ such that

$$\bar{\theta}_1(\beta) \triangleq 1 - \frac{\beta\mu_y L'_{xx}}{2L_{yx}^2}\left(\sqrt{1 + \frac{4L_{yx}^2\gamma}{\beta L'^2_{xx}\mu_y}} - 1\right),$$

$$\bar{\theta}_2(\beta) \triangleq \begin{cases} 1 - \frac{(1-\beta)^2}{8}\frac{\mu_y^2}{L_{yy}^2}\left(\sqrt{1 + \frac{16L_{yy}^2}{(1-\beta)^2\mu_y^2}} - 1\right) & L_{yy} > 0 \\ 0 & L_{yy} = 0, \end{cases}$$

with $L'_{xx} = L_{xx} + 2\gamma$ and

$$\hat{\theta}_1 \triangleq \max\left\{0, 1 - \frac{1}{8}\cdot\gamma^2\cdot\frac{\hat{\epsilon}^2}{\delta_x^2}\right\}, \quad \hat{\theta}_2 \triangleq \left(1 + \frac{1}{8}\cdot\frac{\mu_y\gamma}{11}\cdot\frac{\hat{\epsilon}^2}{\delta_y^2}\right)^{-1}. \tag{43}$$

In fact, in Lemma 7 we provide a convergence guarantee for solving the above problem in (41) using Algorithm 1. Since the parameter choice above satisfies the condition (32) in Lemma 7, we can invoke eq. (33) to complete the rest of the analysis. To be more precise, the problem in eq. (41) is a generic form of the SCSC subproblems given in eq. (4) with $\mu_x = \gamma$; furthermore, by Lemma 9, $(\tau, \sigma, \theta)$ chosen as in (42) satisfies (32) with $\rho = \theta$, $\mu_x = \gamma$, $\alpha = \frac{1}{\sigma} - \sqrt{\theta} L_{yy} > 0$, and $L'_{xx} = L_{xx} + 2\gamma$.

Since $(\hat{x}, y_*(\hat{x}))$ is the saddle point of $\bar{\mathcal{L}}$, then by Lemma 7, we get

$$\mathbb{E}\left[\left(\frac{1}{\tau} - \gamma\right) \|x_N - \hat{x}\|^2\right] \leq \theta^N \left(\frac{1}{\tau}\|x_0 - \hat{x}\|^2 + \frac{1}{\sigma}\|y_0 - y_*(\hat{x})\|^2\right) + \frac{\theta}{1-\theta}\left(\tau \Xi^x_{\tau,\sigma,\theta}\delta_x^2 + \sigma \Xi^y_{\tau,\sigma,\theta}\delta_y^2\right).$$

If we substitute the choice of $\{\tau, \sigma\}$ in eq. (42) into the above inequality, it follows that

$$\mathbb{E}\left[\|x_N - \hat{x}\|^2\right] \leq \theta^{N-1} \max\left\{1, \frac{\mu_y}{\gamma}\right\} \left(\|x_0 - \hat{x}\|^2 + \|y_0 - y_*(\hat{x})\|^2\right) + \frac{1}{\gamma}\left(\tau \Xi^x_{\tau,\sigma,\theta}\delta_x^2 + \sigma \Xi^y_{\tau,\sigma,\theta}\delta_y^2\right).$$

Then, by Jensen's inequality, it follows that

$$\left(\mathbb{E}\left[\|x_N - \hat{x}\|\right]\right)^2 \leq \mathbb{E}\left[\|x_N - \hat{x}\|^2\right] \leq \theta^{N-1} \max\left\{1, \frac{\mu_y}{\gamma}\right\}\mathcal{D}_0^2 + \frac{1}{\gamma}\left(\tau \Xi^x_{\tau,\sigma,\theta}\delta_x^2 + \sigma \Xi^y_{\tau,\sigma,\theta}\delta_y^2\right),$$

where $\mathcal{D}_0 \triangleq \left(\|\hat{x} - x_0\|^2 + \|y_*(\hat{x}) - y_0\|^2\right)^{1/2}$. Thus, for any given $\hat{\epsilon} > 0$, $\mathbb{E}\left[\|x_N - \hat{x}\|\right]$ can be bounded by $\hat{\epsilon}$ when

$$\frac{1}{\gamma}\left(\tau \Xi^x_{\tau,\sigma,\theta}\delta_x^2 + \sigma \Xi^y_{\tau,\sigma,\theta}\delta_y^2\right) \leq \frac{\hat{\epsilon}^2}{2}, \tag{44a}$$

$$\theta^{N-1} \max\left\{1, \frac{\mu_y}{\gamma}\right\}\mathcal{D}_0^2 \leq \frac{\hat{\epsilon}^2}{2}. \tag{44b}$$

Recall that $\Xi^x_{\tau,\sigma,\theta}$, $\Xi^y_{\tau,\sigma,\theta}$ are defined in Lemma 7. Thus, the choice of $\tau$ and $\sigma$ in (42) further implies that

$$\Xi^x_{\tau,\sigma,\theta} = 1 + (1 - \theta^2)\frac{L_{yx}}{2\mu_y},$$

$$\Xi^y_{\tau,\sigma,\theta} = \left(1 + 3\theta + (1 - \theta^2)\frac{L_{yy}}{\mu_y} + (1+\theta)(1-\theta)^2\frac{L_{yx}L_{xy}}{\gamma\mu_y}\right)(1 + 2\theta) + \theta(1-\theta^2)\frac{L_{yx}}{2\gamma}.$$

Since $0 < \theta < 1$ and $1 - \theta^2 \leq 2(1 - \theta)$, we have

$$\Xi^x_{\tau,\sigma,\theta} \leq 1 + (1 - \theta)\frac{L_{yx}}{\mu_y}, \tag{45a}$$

$$\Xi^y_{\tau,\sigma,\theta} \leq 3\left(4 + 2(1 - \theta)\frac{L_{yy}}{\mu_y} + 2(1-\theta)^2\frac{L_{yx}L_{xy}}{\gamma\mu_y}\right) + (1 - \theta)\frac{L_{yx}}{\gamma}. \tag{45b}$$

On the other hand, since $\theta \geq \bar{\theta}(\beta) = \max\{\bar{\theta}_1(\beta), \bar{\theta}_2(\beta)\}$, the inequality $\sqrt{a+b} \leq \sqrt{a} + \sqrt{b}$ for all $a, b \geq 0$, and the definition of $\bar{\theta}(\beta)$ together imply that

$$1 - \theta \leq \min\left\{\frac{\sqrt{\beta\gamma\mu_y}}{L_{yx}}, \frac{1-\beta}{2}\frac{\mu_y}{L_{yy}}\right\}. \tag{46}$$

Therefore, by eq. (46), we can derive that

$$(1-\theta)\frac{L_{yx}}{\mu_y} \leq \sqrt{\frac{\beta\gamma}{\mu_y}}, \quad (1-\theta)\frac{L_{yy}}{\mu_y} \leq \frac{1-\beta}{2}, \quad (1-\theta)^2\frac{L_{yx}L_{xy}}{\gamma\mu_y} \leq \frac{\beta L_{xy}}{L_{yx}}, \quad (1-\theta)\frac{L_{yx}}{\gamma} \leq \sqrt{\frac{\beta\mu_y}{\gamma}};$$

thus, using those inequalities within eq. (45a) and eq. (45b), we get

$$\Xi^x_{\tau,\sigma,\theta} \leq 1 + \sqrt{\frac{\beta\gamma}{\mu_y}}, \tag{47a}$$

$$\Xi^y_{\tau,\sigma,\theta} \leq 15 - 3\beta + 6\beta\frac{L_{xy}}{L_{yx}} + \sqrt{\frac{\beta\mu_y}{\gamma}}. \tag{47b}$$

Note that $\beta = \min\{\frac{1}{2}, \frac{\mu_y}{\gamma}, \frac{\gamma}{\mu_y}, \frac{L_{yx}}{L_{xy}}\} \in (0,1)$ implies that

$$\Xi^x_{\tau,\sigma,\theta} \le 2, \qquad \Xi^y_{\tau,\sigma,\theta} \le 22.$$

Therefore, using the choice of $\{\tau, \sigma\}$ in eq. (42), we obtain a sufficient condition for eq. (44a) as given below:

$$\frac{1-\theta}{\gamma} \frac{2}{\gamma} \delta_x^2 + \frac{1-\theta}{\mu_y \theta} \frac{22}{\gamma} \delta_y^2 \le \frac{\hat{\epsilon}^2}{2}. \tag{48}$$

Our choice of $\theta \in (0,1)$ in (42) implies that $\theta \ge \max\{\hat{\theta}_1, \hat{\theta}_2\}$, where $\hat{\theta}_1$ and $\hat{\theta}_2$ are defined in eq. (43). Note $\theta \ge \max\{\hat{\theta}_1, \hat{\theta}_2\}$ immediately implies that the above sufficient condition in (48) holds. Therefore, with the choice of $\{\tau, \sigma, \theta\}$ in eq. (42) we obtain that eq. (44a) holds, i.e.,

$$\frac{1}{\gamma}\left(\tau \Xi^x_{\tau,\sigma,\theta} \delta_x^2 + \sigma \Xi^y_{\tau,\sigma,\theta} \delta_y^2\right) \le \frac{\hat{\epsilon}^2}{2}.$$

Furthermore, (44b) holds when $N \ge \ln\left(\frac{2\max\{1, \mu_y/\gamma\}\mathcal{D}_0^2}{\hat{\epsilon}^2}\right)/\ln\left(\frac{1}{\theta}\right) + 1$. Thus, we conclude that for any $\hat{\epsilon} > 0$, SAPD, stated in Algorithm 1, can generate $x_N$ such that $\mathbb{E}\left[\|x_N - \hat{x}\|\right] \le \hat{\epsilon}$ within $N_{\hat{\epsilon}}$ iterations for $\theta = \max\{\bar{\theta}(\beta), \hat{\theta}_1, \hat{\theta}_2\}$, where

$$N_{\hat{\epsilon}} = \mathcal{O}\left(\ln\left(\frac{\max\{1, \mu_y/\gamma\}}{\hat{\epsilon}}\right)/\ln\left(\frac{1}{\theta}\right) + 1\right). \tag{49}$$

Note $\frac{1}{\ln(\frac{1}{\theta})} \le (1-\theta)^{-1}$ for $\theta \in (0,1)$ implies that

$$\frac{1}{\ln(\frac{1}{\theta})} \le \mathcal{O}\left(\max\{(1-\bar{\theta}_1(\beta))^{-1}, (1-\bar{\theta}_2(\beta))^{-1}, (1-\hat{\theta}_1)^{-1}, (1-\hat{\theta}_2)^{-1}\}\right).$$

First, we equivalently rewrite $(1 - \bar{\theta}_1(\beta))^{-1}$ and $(1 - \bar{\theta}_2(\beta))^{-1}$ as follows:

$$(1-\bar{\theta}_1(\beta))^{-1} = \frac{1}{2}\frac{L'_{xx}}{\gamma} + \sqrt{\frac{1}{4}\frac{L'_{xx}}{\gamma^2}^2 + \frac{L_{yx}^2}{\beta\gamma\mu_y}}, \quad (1-\bar{\theta}_2(\beta))^{-1} = \frac{1}{2} + \sqrt{\frac{1}{4} + \frac{4L_{yy}^2}{(1-\beta)^2\mu_y^2}};$$

thus,

$$(1-\bar{\theta}_1)^{-1} \le \frac{L'_{xx}}{\gamma} + \frac{L_{yx}}{\sqrt{\beta\gamma\mu_y}}, \qquad (1-\bar{\theta}_2)^{-1} \le 1 + \frac{2}{1-\beta}\cdot\frac{L_{yy}}{\mu_y}.$$

Finally,

$$(1-\hat{\theta}_1)^{-1} = \mathcal{O}\left(\frac{1}{\gamma^2}\cdot\frac{\delta_x^2}{\hat{\epsilon}^2}\right), \qquad (1-\hat{\theta}_2)^{-1} = \mathcal{O}\left(\frac{1}{\gamma\mu_y}\cdot\frac{\delta_y^2}{\hat{\epsilon}^2} + 1\right).$$

Recall that $L'_{xx} = 2\gamma + L_{xx}$, using the above four identities that and our choice of $\beta = \min\{\frac{1}{2}, \frac{\mu_y}{\gamma}, \frac{\gamma}{\mu_y}, \frac{L_{yx}}{L_{xy}}\}$ we derive that

$$\frac{1}{\ln(\frac{1}{\theta})} = \mathcal{O}\left(\frac{\max\{L_{xx}, L_{yx}\}}{\gamma} + \frac{\max\{L_{yx}, L_{xy}\}}{\sqrt{\gamma\mu_y}} + \frac{\max\{L_{yy}, L_{yx}\}}{\mu_y} + \left(\frac{\delta_x^2}{\gamma} + \frac{\delta_y^2}{\mu_y}\right)\frac{1}{\gamma\hat{\epsilon}^2}\right),$$

From (49), we conclude that

$$N_{\hat{\epsilon}} = \mathcal{O}\left(\frac{\max\{L_{xx}, L_{yx}\}}{\gamma} + \frac{\max\{L_{yx}, L_{xy}\}}{\sqrt{\gamma\mu_y}} + \frac{\max\{L_{yy}, L_{yx}\}}{\mu_y} + \left(\frac{\delta_x^2}{\gamma} + \frac{\delta_y^2}{\mu_y}\right)\frac{1}{\gamma\hat{\epsilon}^2}\right)\cdot\ln\left(\frac{\max\{1, \mu_y/\gamma\}}{\hat{\epsilon}}\right),$$

which completes the proof. □

**Lemma 11.** *Suppose $f : \mathcal{X} \to \mathbb{R} \cup \{+\infty\}$ is closed convex, and $V$ is a strictly convex function on* **dom** *$f$ and differentiable on an open set containing* **dom** *$f$. Let $x_* = \operatorname{argmin}_{x\in\mathcal{X}} f(x) + V(x)$. Then, for any $\alpha > 0$, it holds that $x_* = \mathbf{prox}_{\alpha f}(x_* - \alpha\nabla V(x_*))$.*

*Proof.* From the first-order optimality condition, we have

$$0 \in \partial f(x_*) + \nabla V(x_*). \tag{50}$$

Moreover, from the definition of $\mathbf{prox}_{\alpha f}(\cdot)$ operator, it follows that

$$\mathbf{prox}_{\alpha f}(x_* - \alpha\nabla V(x_*)) = \operatorname*{argmin}_{x\in\mathcal{X}} f(x) + \nabla V(x_*)^\top(x - x_*) + \frac{1}{2\alpha}\|x - x_*\|^2. \tag{51}$$

Finally, (50) implies that $x_*$ is the unique minimizer of the problem on the rhs of (51). Therefore, we get that $x_* = \mathbf{prox}_{\alpha f}(x_* - \alpha\nabla V(x_*))$, which completes the proof. □

## B.1   Proof of Theorem 2

We are now ready to prove Theorem 2.

*Proof.* Let $\hat{x} = \mathbf{prox}_{\lambda\phi}(x_\epsilon)$, and $\phi^s$ be the smooth part of $\phi$, i.e., $\phi = f + \phi^s$. Moreover, since $\Phi(x, \cdot) - g(\cdot)$ is strongly concave and $\Phi(\cdot, y)$ is differentiable, we have that $\phi^s$ is differentiable; hence, for any $x \in \mathbf{dom}\, f$,

$$\nabla\phi^s(x) = \nabla_x\Phi(x, y_*(x)), \quad \text{where} \quad y_*(x) = \underset{y\in\mathcal{Y}}{\mathrm{argmax}}\, \Phi(x, y) - g(y).$$

Then we can explicitly write $\hat{x}$ as

$$\hat{x} = \underset{x\in\mathcal{X}}{\mathrm{argmin}}\, f(x) + \phi^s(x) + \frac{1}{2\lambda}\|x - x_\epsilon\|^2.$$

Since $\phi^s(\cdot) + \frac{1}{2\lambda}\|\cdot - x_\epsilon\|^2$ is smooth and strongly convex, for any $\alpha > 0$, Lemma 11 implies that

$$\hat{x} = \mathbf{prox}_{\alpha f}\left(\hat{x} - \alpha\big(\nabla\phi^s(\hat{x}) + \frac{1}{\lambda}(\hat{x} - x_\epsilon)\big)\right).$$

If we let $\alpha = \lambda$, it follows that

$$\hat{x} = \mathbf{prox}_{\lambda f}\big(x_\epsilon - \lambda\nabla_x\phi^s(\hat{x})\big).$$

Moreover, since $f$ is closed convex, $\mathbf{prox}_f(\cdot)$ is nonexpansive; hence,

$$\mathbb{E}\left[\|\hat{x} - \mathbf{prox}_{\lambda f}\big(\hat{x} - \lambda\nabla_x\phi^s(\hat{x})\big)\|\right] \le \mathbb{E}\left[\|x_\epsilon - \hat{x}\|\right] \le \frac{\lambda\epsilon}{2}, \tag{52}$$

where we used Lemma 1 for the last inequality, i.e., $\|x_\epsilon - \hat{x}\| = \lambda\|\nabla\phi_\lambda(x_\epsilon)\|$. On the other hand, for any $\tilde{x} \in \mathbf{dom}\, f$,

$$\mathbb{E}\left[\|\tilde{x} - \mathbf{prox}_{\lambda f}\big(\tilde{x} - \lambda\nabla_x\phi^s(\tilde{x})\big)\|\right]$$

$$\le \mathbb{E}\left[\|\tilde{x} - \mathbf{prox}_{\lambda f}\big(\tilde{x} - \lambda\nabla_x\phi^s(\tilde{x})\big) - \hat{x} + \mathbf{prox}_{\lambda f}\big(\hat{x} - \lambda\nabla_x\phi^s(\hat{x})\big)\|\right] + \frac{\lambda\epsilon}{2} \tag{53}$$

$$\le 2\mathbb{E}\left[\|\tilde{x} - \hat{x}\|\right] + \lambda\mathbb{E}\left[\|\nabla_x\Phi(\tilde{x}, y_*(\tilde{x})) - \nabla_x\Phi(\hat{x}, y_*(\hat{x}))\|\right] + \frac{\lambda\epsilon}{2}.$$

According to [7, Proposition 1], $y_*(\cdot)$ is Lipschitz with constant $\kappa_{yx} = \frac{L_{yx}}{\mu_y}$. Therefore, we get

$$\|\nabla_x\Phi(\tilde{x}, y_*(\tilde{x})) - \nabla_x\Phi(\hat{x}, y_*(\hat{x}))\| \le L_{xx}\|\tilde{x} - \hat{x}\| + L_{xy}\|y_*(\tilde{x}) - y_*(\hat{x})\| \le \big(L_{xx} + L_{xy}\kappa_{yx}\big)\|\tilde{x} - \hat{x}\|,$$

which together with eq. (53) implies that

$$\frac{1}{\lambda}\mathbb{E}\left[\|\tilde{x} - \mathbf{prox}_{\lambda f}\big(\tilde{x} - \lambda\nabla_x\phi^s(\tilde{x})\big)\|\right] \le \big(\frac{2}{\lambda} + L_{xx} + L_{xy}\kappa_{yx}\big)\mathbb{E}\left[\|\tilde{x} - \hat{x}\|\right] + \frac{\epsilon}{2}. \tag{54}$$

Let $\lambda^{-1} = 2\gamma$, and $C \triangleq (4\gamma + L_{xx} + L_{xy}\kappa_{yx})^{-1}/2$. Thus, for any $\tilde{x} \in \mathbf{dom}\, f$ such that $\mathbb{E}\left[\|\tilde{x} - \hat{x}\|\right] \le C\epsilon$, we have

$$\mathbb{E}\left[\frac{1}{\lambda}\|\tilde{x} - \mathbf{prox}_{\lambda f}\big(\tilde{x} - \lambda\nabla_x\phi^s(\tilde{x})\big)\|\right] \le \epsilon.$$

Indeed, when $f(x) = 0$ for all $x \in \mathcal{X}$, we get $\phi(x) = \phi^s(x)$ and the above inequality implies that

$$\mathbb{E}\left[\|\nabla\phi(\tilde{x})\|\right] \le \epsilon.$$

The rest directly follows from invoking Lemma 10 with $\hat{\epsilon} = C\epsilon$, and $x_0 = x_\epsilon$. □

## C Proofs of Lemma 2 and Lemma 7

We first discuss the proof of Lemma 7 and later establish Lemma 2 through specializing some parts of this proof. Indeed recall that Lemma 7 is stated for a generic SAPD+ subproblem of the form (2). In Lemma 12 below, we restate Lemma 7 and rather than using a generic subproblem, we state it for the specific subproblems as in (4), which arise while implementing SAPD+. It is crucial to remind that the matrix inequality (MI) we establish in Lemma 7, i.e., eq. (32), helps us describe the admissible set of algorithm parameters that guarantee the *linear* convergence of inner loop iterates generated by SAPD, i.e., $\left\{ \mathbb{E}\left[ \|x_k^t - x_*^t\|^2 + \|y_k^t - y_*^t\|^2 \right] \right\}_{k \geq 0}$, for any $t \geq 0$.

**Lemma 12.** *Suppose Assumptions 1, 2, 3 hold. For any given $\mu_x > 0$ and $t \in \mathbb{Z}_+$, consider solving the SCSC subproblem in* (4) *using SAPD, displayed in Algorithm 1. Let $(x_*^t, y_*^t)$ denote the unique saddle point of* (4)*, and let $\{x_k^t, y_k^t\}_{k \geq 0}$ be the iterate sequence when initialized from $(x_0^t, y_0^t) \in \mathbf{dom}\, f \times \mathbf{dom}\, g$ and using $\tau, \sigma, \theta$ that satisfy* (32) *for some $\alpha \in [0, \frac{1}{\sigma})$ and $\rho \in (0, 1)$, where $L_{xx}' \triangleq L_{xx} + \mu_x + \gamma$. Then for all $N \geq \mathbb{Z}_+$, it holds that*

$$
\begin{aligned}
&\mathbb{E}\left[ \left( \frac{1}{\tau} - \mu_x \right) \|x_N^t - x_*^t\|^2 + \left( \frac{1}{\sigma} - \alpha \right) \|y_N^t - y_*^t\|^2 \right] \\
&\leq \rho^N \mathbb{E}\left[ \frac{1}{\tau} \|x_0^t - x_*^t\|^2 + \frac{1}{\sigma} \|y_0^t - y_*^t\|^2 \right] + \frac{\rho}{1 - \rho}\left( \tau \Xi_{\tau,\sigma,\theta}^x \delta_x^2 + \sigma \Xi_{\tau,\sigma,\theta}^y \delta_y^2 \right),
\end{aligned}
\tag{55}
$$

*where $\Xi_{\tau,\sigma,\theta}^x$ and $\Xi_{\tau,\sigma,\theta}^y$ are defined in* (18a) *and* (18b)*, respectively.*

*Proof.* The proof is provided in appendix C.2. □

Clearly, it is sufficient to prove Lemma 12 in order to establish Lemma 7. Moreover, as stated earlier, we will exploit the techniques used in the proof of Lemma 12 when showing Lemma 2 –this is why we restated Lemma 7.

### C.1 Preliminary technical results

Recall that given some $x_0^t \in \mathbf{dom}\, f$ and $\mu_x > 0$, we define

$$
\mathcal{L}^t(x, y) \triangleq f(x) + \Phi^t(x, y) - g(y),
\tag{56a}
$$

$$
\Phi^t(x, y) \triangleq \Phi(x, y) + \frac{\mu_x + \gamma}{2} \|x - x_0^t\|^2,
\tag{56b}
$$

where $\gamma > 0$ is the weak-convexity constant of $\Phi(\cdot, y)$ for any $y \in \mathbf{dom}\, g$. It follows from Assumption 2 that $\nabla_y \Phi^t$ and $\nabla_x \Phi^t$ are Lipschitz such that

$$
\|\nabla_y \Phi^t(x, y) - \nabla_y \Phi^t(x', y')\| \leq L_{yx} \|x - x'\| + L_{yy} \|y - y'\|,
\tag{57}
$$

$$
\|\nabla_x \Phi^t(x, y) - \nabla_x \Phi^t(x', y')\| \leq L_{xx}' \|x - x'\| + L_{xy} \|y - y'\|
\tag{58}
$$

such that $L_{xx}' \triangleq L_{xx} + \mu_x + \gamma$. Furthermore, (56b) implies that for any $y \in \mathbf{dom}\, g$, $\Phi^t(\cdot, y)$ is strongly convex with modulus $\mu_x > 0$.

We will derive some key inequalities below for SAPD iterates $\{x_k^t, y_k^t\}_{k \geq 0}$ generated by Algorithm 1 to solve $\min_x \max_y \mathcal{L}^t(x, y)$. Let $x_{-1}^t = x_0^t$, $y_{-1}^t = y_0^t$, and for $k \geq 0$, define

$$
q_k^t \triangleq \nabla_y \Phi^t(x_k^t, y_k^t) - \nabla_y \Phi^t(x_{k-1}^t, y_{k-1}^t), \qquad s_k^t \triangleq \nabla_y \Phi^t(x_k^t, y_k^t) + \theta q_k^t.
\tag{59}
$$

Thus $q_0^t = \mathbf{0}$; and for $k \geq 0$, Assumption 2 implies that

$$
\|q_{k+1}^t\| \leq L_{yx} \|x_{k+1}^t - x_k^t\| + L_{yy} \|y_{k+1}^t - y_k^t\|.
\tag{60}
$$

**Lemma 13.** *Suppose Assumptions 1, 2, 3 hold. Let $\{x_k^t, y_k^t\}_{k \geq 0}$ be SAPD iterates generated according to Algorithm 1 for solving $\min_x \max_y \mathcal{L}^t(x, y)$. Then for all $x \in \mathbf{dom}\, f \subset \mathcal{X}$, $y \in \mathbf{dom}\, g \subset \mathcal{Y}$, and $k \geq 0$,*

$$
\begin{aligned}
&\mathcal{L}^t(x_{k+1}^t, y) - \mathcal{L}^t(x, y_{k+1}^t) \\
&\leq -\langle q_{k+1}^t, y_{k+1}^t - y \rangle + \theta \langle q_k^t, y_k^t - y \rangle + \Lambda_k^t(x, y) - \Sigma_{k+1}^t(x, y) + \Gamma_{k+1}^t + \varepsilon_k^{t,x}(x) + \varepsilon_k^{t,y}(y),
\end{aligned}
\tag{61}
$$

*where*

$$\varepsilon_k^{t,x}(x) \triangleq \langle \tilde{\nabla}_x \Phi^t(x_k^t, y_{k+1}^t; \omega_k^x) - \nabla_x \Phi^t(x_k^t, y_{k+1}^t), \ x - x_{k+1}^t \rangle, \quad \varepsilon_k^{t,y}(y) \triangleq \langle \tilde{s}_k^t - s_k^t, y_{k+1}^t - y \rangle,$$

$q_k^t$ *and* $s_k^t$ *are defined as in* (59)*, and*

$$\Lambda_k^t(x, y) \triangleq (\frac{1}{2\tau} - \frac{\mu_x}{2}) \|x - x_k^t\|^2 + \frac{1}{2\sigma} \|y - y_k^t\|^2,$$

$$\Sigma_{k+1}^t(x, y) \triangleq \frac{1}{2\tau} \|x - x_{k+1}^t\|^2 + (\frac{1}{2\sigma} + \frac{\mu_y}{2}) \|y - y_{k+1}^t\|^2,$$

$$\Gamma_{k+1}^t \triangleq (\frac{L'_{xx}}{2} - \frac{1}{2\tau}) \|x_{k+1}^t - x_k^t\|^2 - \frac{1}{2\sigma} \|y_{k+1}^t - y_k^t\|^2$$
$$+ \theta L_{yx} \|x_k^t - x_{k-1}^t\| \|y_{k+1}^t - y_k^t\| + \theta L_{yy} \|y_k^t - y_{k-1}^t\| \|y_{k+1}^t - y_k^t\|.$$

*Proof.* Fix $x \in \mathbf{dom} f$ and $y \in \mathbf{dom} g$. Using Lemma 7.1 from [15] for the $y-$ and $x-$subproblems in Algorithm 1, we get

$$f(x_{k+1}^t) + \langle \tilde{\nabla}_x \Phi^t(x_k^t, y_{k+1}^t; \omega_k^x), x_{k+1}^t - x \rangle \leq f(x) + \frac{1}{2\tau} \left[ \|x - x_k^t\|^2 - \|x - x_{k+1}^t\|^2 - \|x_{k+1}^t - x_k^t\|^2 \right],$$

$$g(y_{k+1}^t) - \langle \tilde{s}_k^t, y_{k+1}^t - y \rangle \leq g(y) + \frac{1}{2\sigma} \left[ \|y - y_k^t\|^2 - \|y - y_{k+1}^t\|^2 - \|y_{k+1}^t - y_k^t\|^2 \right].$$

Thus, by adding and subtracting we further get

$$f(x_{k+1}^t) + \langle \nabla_x \Phi^t(x_k^t, y_{k+1}^t), \ x_{k+1}^t - x \rangle$$
$$\leq f(x) + \frac{1}{2\tau} (\|x - x_k^t\|^2 - \|x - x_{k+1}^t\|^2 - \|x_{k+1}^t - x_k^t\|^2) + \varepsilon_k^{t,x}(x), \tag{62a}$$

$$g(y_{k+1}^t) - \langle s_k^t, y_{k+1}^t - y \rangle$$
$$\leq g(y) + \frac{1}{2\sigma} (\|y - y_k^t\|^2 - \|y - y_{k+1}^t\|^2 - \|y_{k+1}^t - y_k^t\|^2) + \varepsilon_k^{t,y}(y). \tag{62b}$$

Rearranging the terms in (62b), we get

$$-g(y) + g(y_{k+1}^t)$$
$$\leq \langle s_k^t, y_{k+1}^t - y \rangle + \frac{1}{2\sigma} \left[ \|y - y_k^t\|^2 - \|y - y_{k+1}^t\|^2 - \|y_{k+1}^t - y_k^t\|^2 \right] + \varepsilon_k^{t,y}(y). \tag{63}$$

Since $y_{k+1}^t \in \mathbf{dom} g$, the inner product in (62a) can be lower bounded using convexity of $\Phi^t(\cdot, y_{k+1}^t)$ as follows (see Assumption 2):

$$\langle \nabla_x \Phi^t(x_k^t, y_{k+1}^t), x_{k+1}^t - x \rangle = \langle \nabla_x \Phi^t(x_k^t, y_{k+1}^t), x_k^t - x \rangle + \langle \nabla_x \Phi^t(x_k^t, y_{k+1}^t), x_{k+1}^t - x_k^t \rangle$$
$$\geq \Phi^t(x_k^t, y_{k+1}^t) - \Phi^t(x, y_{k+1}^t) + \frac{\mu_x}{2} \|x - x_k^t\|^2 + \langle \nabla_x \Phi^t(x_k^t, y_{k+1}^t), \ x_{k+1}^t - x_k^t \rangle.$$

Using this inequality after adding $\Phi^t(x_{k+1}^t, y_{k+1}^t)$ to both sides of (62a), we get

$$\Phi^t(x_{k+1}^t, y_{k+1}^t) + f(x_{k+1}^t)$$
$$\leq \Phi^t(x, y_{k+1}^t) + f(x) + \Phi^t(x_{k+1}^t, y_{k+1}^t) - \Phi^t(x_k^t, y_{k+1}^t) - \langle \nabla_x \Phi^t(x_k^t, y_{k+1}^t), x_{k+1}^t - x_k^t \rangle$$
$$+ \frac{1}{2\tau} \left[ \|x - x_k^t\|^2 - \|x - x_{k+1}^t\|^2 - \|x_{k+1}^t - x_k^t\|^2 \right] - \frac{\mu_x}{2} \|x - x_k^t\|^2 + \varepsilon_k^{t,x}(x)$$
$$\leq \Phi^t(x, y_{k+1}^t) + f(x) + \frac{L'_{xx}}{2} \|x_{k+1}^t - x_k^t\|^2$$
$$+ \frac{1}{2\tau} \left[ \|x - x_k^t\|^2 - \|x - x_{k+1}^t\|^2 - \|x_{k+1}^t - x_k^t\|^2 \right] - \frac{\mu_x}{2} \|x - x_k^t\|^2 + \varepsilon_k^{t,x}(x), \tag{64}$$

where the last step uses Assumption 2. Rearranging the terms gives us

$$f(x_{k+1}^t) - f(x) - \Phi^t(x, y_{k+1}^t) \leq -\Phi^t(x_{k+1}^t, y_{k+1}^t) + \frac{L'_{xx}}{2} \|x_{k+1}^t - x_k^t\|^2$$
$$+ \frac{1}{2\tau} \left[ \|x - x_k^t\|^2 - \|x - x_{k+1}^t\|^2 - \|x_{k+1}^t - x_k^t\|^2 \right] - \frac{\mu_x}{2} \|x - x_k^t\|^2 + \varepsilon_k^{t,x}(x). \tag{65}$$

Then, for $k \geq 0$, by summing (63) and (65), we obtain

$$\mathcal{L}(x_{k+1}^t, y) - \mathcal{L}(x, y_{k+1}^t) = f(x_{k+1}^t) + \Phi^t(x_{k+1}^t, y) - g(y) - f(x) - \Phi^t(x, y_{k+1}^t) + g(y_{k+1}^t)$$

$$\leq \Phi^t(x_{k+1}^t, y) - \Phi^t(x_{k+1}^t, y_{k+1}) + \langle s_k^t, y_{k+1} - y \rangle + \frac{L'_{xx}}{2} \|x_{k+1}^t - x_k^t\|^2$$

$$+ \frac{1}{2\sigma} \left[ \|y - y_k^t\|^2 - \|y - y_{k+1}^t\|^2 - \|y_{k+1}^t - y_k^t\|^2 \right] + \varepsilon_k^{t,y}(y)$$

$$+ \frac{1}{2\tau} \left[ \|x - x_k^t\|^2 - \|x - x_{k+1}^t\|^2 - \|x_{k+1}^t - x_k^t\|^2 \right] - \frac{\mu_x}{2} \|x - x_k^t\|^2 + \varepsilon_k^{t,x}(x).$$

(66)

From Assumption 2, the $\mu_y$-strongly concavity of $\Phi^t(x, \cdot)$ for fixed $x \in \mathbf{dom}\, f \subset \mathcal{X}$ implies

$$\Phi^t(x_{k+1}^t, y) - \Phi^t(x_{k+1}^t, y_{k+1}^t) + \langle s_k^t, y_{k+1}^t - y \rangle$$

$$\leq \langle \nabla_y \Phi^t(x_{k+1}^t, y_{k+1}^t), y - y_{k+1}^t \rangle - \frac{\mu_y}{2} \|y - y_{k+1}^t\|^2 + \langle \nabla_y \Phi^t(x_k^t, y_k^t) + \theta q_k^t, y_{k+1}^t - y \rangle$$

$$= - \langle q_{k+1}^t, y_{k+1}^t - y \rangle - \frac{\mu_y}{2} \|y - y_{k+1}^t\|^2 + \theta \langle q_k^t, y_k^t - y \rangle + \theta \langle q_k^t, y_{k+1}^t - y_k^t \rangle.$$

Thus, using the above inequality within (66), we get

$$\mathcal{L}^t(x_{k+1}^t, y) - \mathcal{L}^t(x, y_{k+1}^t) \leq -\langle q_{k+1}^t, y_{k+1}^t - y \rangle + \theta \langle q_k^t, y_k^t - y \rangle + \theta \langle q_k^t, y_{k+1}^t - y_k^t \rangle$$

$$+ \frac{L'_{xx}}{2} \|x_{k+1}^t - x_k^t\|^2 + \frac{1}{2\sigma} \left[ \|y - y_k^t\|^2 - \|y - y_{k+1}^t\|^2 - \|y_{k+1}^t - y_k^t\|^2 \right] - \frac{\mu_y}{2} \|y - y_{k+1}^t\|^2$$

$$+ \frac{1}{2\tau} \left[ \|x - x_k^t\|^2 - \|x - x_{k+1}^t\|^2 - \|x_{k+1}^t - x_k^t\|^2 \right] - \frac{\mu_x}{2} \|x - x_k^t\|^2 + \varepsilon_k^{t,x}(x) + \varepsilon_k^{t,y}(y).$$

Finally, (61) follows from using Cauchy-Schwarz for $\langle q_k^t, y_{k+1}^t - y_k^t \rangle$ and (60). $\qquad \square$

**Lemma 14.** *[3, Theorem 6.42] Let $f$ be proper, closed and convex function. Then for any $x, x' \in \mathcal{X}$, we get $\|\mathbf{prox}_f(x) - \mathbf{prox}_f(x')\| \leq \|x - x'\|$.*

Next, based on the above inequality, we prove an intermediate result, which we use later to bound the variance of the SAPD iterate sequence.

**Lemma 15.** *Suppose Assumptions 1, 2, 3 hold. Let $\{x_k^t, y_k^t\}_{k \geq 0}$ be SAPD iterates generated as in Algorithm 1 for solving $\min_x \max_y \mathcal{L}^t(x, y)$. For $k \geq 0$, let $q_k^t$ and $s_k^t$ be defined as in (59), and let*

$$\hat{x}_{k+1}^t \triangleq \mathbf{prox}_{\tau f}\left(x_k^t - \tau \nabla_x \Phi^t(x_k^t, y_{k+1}^t)\right), \quad \hat{\hat{x}}_{k+1}^t \triangleq \mathbf{prox}_{\tau f}\left(x_k^t - \tau \nabla_x \Phi^t(x_k^t, \hat{y}_{k+1}^t)\right),$$

$$\hat{y}_{k+1}^t \triangleq \mathbf{prox}_{\sigma g}\left(y_k^t + \sigma s_k^t\right), \quad \hat{\hat{y}}_{k+1}^t \triangleq \mathbf{prox}_{\sigma g}\left(\hat{y}_k^t + \sigma(1+\theta)\nabla_y \Phi^t(\hat{x}_k^t, \hat{y}_k^t) - \sigma\theta\nabla_y \Phi^t(x_{k-1}^t, y_{k-1}^t)\right),$$

*then the following inequalities hold for $k \geq 0$:*

$$\|x_{k+1}^t - \hat{x}_{k+1}^t\| \leq \tau \|\Delta_k^{t,x}\|, \qquad \|y_{k+1}^t - \hat{y}_{k+1}^t\| \leq \sigma\left((1+\theta)\|\Delta_k^{t,y}\| + \theta\|\Delta_{k-1}^{t,y}\|\right), \qquad (67a)$$

$$\|y_{k+1}^t - \hat{\hat{y}}_{k+1}^t\| \leq \sigma\left((1+\theta)\|\Delta_k^{t,y}\| + \theta\|\Delta_{k-1}^{t,y}\| + \tau(1+\theta)L_{yx}\|\Delta_{k-1}^{t,x}\|\right) \qquad (67b)$$

$$+ \sigma\left(1 + \sigma(1+\theta)L_{yy} + \tau\sigma(1+\theta)L_{yx}L_{xy}\right)\left((1+\theta)\|\Delta_{k-1}^{t,y}\| + \theta\|\Delta_{k-2}^{t,y}\|\right),$$

*where $\Delta_k^{t,x} \triangleq \tilde{\nabla}_x \Phi^t(x_k^t, y_{k+1}^t; \omega_k^x) - \nabla_x \Phi^t(x_k^t, y_{k+1}^t)$, and $\Delta_k^{t,y} \triangleq \tilde{\nabla}_y \Phi^t(x_k^t, y_k^t; \omega_k^y) - \nabla_y \Phi^t(x_k^t, y_k^t)$.*

*Proof.* The first inequality in eq. (67a) is from Lemma 14; for the second, we have

$$\|y_{k+1}^t - \hat{y}_{k+1}^t\| \leq \sigma\|\tilde{s}_k^t - s_k^t\| \leq \sigma\left((1+\theta)\|\Delta_k^{t,y}\| + \theta\|\Delta_{k-1}^{t,y}\|\right),$$

which follows from Lemma 14 and the triangle inequality. To show eq. (67b), we bound $\|y_{k+1}^t - \hat{y}_{k+1}^t\|$ and $\|\hat{y}_{k+1}^t - \hat{\hat{y}}_{k+1}^t\|$ separately. It follows from Lemma 14 that $\|x_{k+1}^t - \hat{\hat{x}}_{k+1}^t\| \leq \tau\|\tilde{\nabla}_x \Phi^t(x_k^t, y_{k+1}^t; \omega_k^x) - \nabla_x \Phi^t(x_k^t, \hat{y}_{k+1}^t)\|$. After adding and subtracting $\nabla_x \Phi^t(x_k^t, y_{k+1}^t)$, Assumption 2 implies that

$$\|x_{k+1}^t - \hat{\hat{x}}_{k+1}^t\| \leq \tau\left(\|\Delta_k^{t,x}\| + L_{xy}\|y_{k+1}^t - \hat{y}_{k+1}^t\|\right). \qquad (68)$$

We will use this relation to bound $\|\hat{y}_{k+1}^t - \hat{\hat{y}}_{k+1}^t\|$. Indeed, using Lemma 14, we have

$$
\begin{aligned}
\|\hat{y}_{k+1}^t - \hat{\hat{y}}_{k+1}^t\| \leq & \|y_k^t - \hat{y}_k^t + \sigma(1+\theta)\left(\nabla_y \Phi^t(x_k^t, y_k^t) - \nabla_y \Phi^t(\hat{x}_k^t, \hat{y}_k^t)\right)\| \\
\leq & (1 + \sigma(1+\theta)L_{yy})\|y_k^t - \hat{y}_k^t\| + \sigma(1+\theta)L_{yx}\|x_k^t - \hat{x}_k^t\| \\
\leq & \left(1 + \sigma(1+\theta)L_{yy} + \tau\sigma(1+\theta)L_{yx}L_{xy}\right)\|y_k^t - \hat{y}_k^t\| + \tau\sigma(1+\theta)L_{yx}\|\Delta_{k-1}^{t,x}\| \\
\leq & \sigma\left(1 + \sigma(1+\theta)L_{yy} + \tau\sigma(1+\theta)L_{yx}L_{xy}\right) \cdot \left((1+\theta)\|\Delta_{k-1}^{t,y}\| + \theta\|\Delta_{k-2}^{t,y}\|\right) \\
& + \sigma\tau(1+\theta)L_{yx}\|\Delta_{k-1}^{t,x}\|,
\end{aligned}
$$

where the second, third and fourth inequalities follow from Assumption 2, eq. (68) and the second inequality in eq. (67a), respectively. Combining this with $\|y_{k+1}^t - \hat{\hat{y}}_{k+1}^t\| \leq \|y_{k+1}^t - \hat{y}_{k+1}^t\| + \|\hat{y}_{k+1}^t - \hat{\hat{y}}_{k+1}^t\|$, and the second inequality in eq. (67a) give us the desired bound. $\qquad\square$

Next, we provide some inequalities to bound the SAPD variance term later in our analysis.

**Lemma 16.** *Suppose Assumptions 1, 2, 3 hold. Let $\{x_k^t, y_k^t\}_{k\geq 0}$ be SAPD iterates generated according to Algorithm 1 for solving $\min_x \max_y \mathcal{L}^t(x, y)$. The following inequality holds for all $k \geq 0$:*

$$
\mathbb{E}\left[\langle \Delta_k^{t,x}, \hat{x}_{k+1}^t - x_{k+1}^t\rangle\right] \leq \tau\delta_x^2, \qquad \mathbb{E}\left[\langle \Delta_k^{t,y}, y_{k+1}^t - \hat{y}_{k+1}^t\rangle\right] \leq \sigma(1+2\theta)\delta_y^2,
$$

$$
\mathbb{E}\left[\langle \Delta_{k-1}^{t,y}, \hat{y}_{k+1}^t - y_{k+1}^t\rangle\right]
$$

$$
\leq \sigma\left[\left((2 + \sigma(1+\theta)L_{yy} + \tau\sigma(1+\theta)L_{yx}L_{xy}) \cdot (1+2\theta) + \frac{\tau(1+\theta)L_{yx}}{2}\right)\delta_y^2 + \frac{\tau(1+\theta)L_{yx}}{2}\delta_x^2\right],
$$

*where $\Delta_k^{t,x}$ and $\Delta_k^{t,y}$ are defined in Lemma 15.*

*Proof.* With the convention that $y_{-2}^t = y_{-1}^t = y_0^t$, and $x_{-2}^t = x_{-1}^t = x_0^t$, Lemma 15 and Cauchy-Schwarz inequality imply for all $k \geq 0$ that

$$
\langle \Delta_k^{t,x}, x_{k+1}^t - \hat{x}_{k+1}^t\rangle \leq \tau\|\Delta_k^{t,x}\|^2,
$$

$$
\langle \Delta_k^{t,y}, y_{k+1}^t - \hat{y}_{k+1}^t\rangle \leq \sigma\left((1+\theta)\|\Delta_k^{t,y}\|^2 + \theta\|\Delta_{k-1}^{t,y}\|\|\Delta_k^{t,y}\|\right),
$$

$$
\langle \Delta_{k-1}^{t,y}, y_{k+1}^t - \hat{y}_{k+1}^t\rangle \leq \sigma\Bigg((1+\theta)\|\Delta_k^{t,y}\|\|\Delta_{k-1}^{t,y}\| + \theta\|\Delta_{k-1}^{t,y}\|^2 + \tau(1+\theta)L_{yx}\|\Delta_{k-1}^{t,x}\|\|\Delta_{k-1}^{t,y}\|
$$

$$
+ \left(1 + \sigma(1+\theta)L_{yy} + \tau\sigma(1+\theta)L_{yx}L_{xy}\right) \cdot \left((1+\theta)\|\Delta_{k-1}^{t,y}\|^2 + \theta\|\Delta_{k-2}^{t,y}\|\|\Delta_{k-1}^{t,y}\|\right)\Bigg).
$$

Next, using Assumption 3 and $\|a\|\|b\| \leq \frac{1}{2}\|a\|^2 + \frac{1}{2}\|b\|^2$, which holds for $a, b \in \mathbb{R}^n$, and taking the expectation leads to the desired result. $\qquad\square$

Before we move on to prove our intermediate result in Lemma 12, we give two technical lemmas that help us simplify the SAPD parameter selection rule and lead to the matrix inequality in eq. (32).

**Lemma 17.** *Given $\tau, \sigma > 0$, $\theta, \alpha \geq 0$, and $\rho \in (0, 1)$, let*

$$
G' \triangleq \begin{pmatrix}
\frac{1}{\tau}(1 - \frac{1}{\rho}) + \frac{\mu_x}{\rho} & 0 & 0 & 0 & 0 \\
0 & \frac{1}{\sigma}(1 - \frac{1}{\rho}) + \mu_y & -|1 - \frac{\theta}{\rho}|L_{yx} & -|1 - \frac{\theta}{\rho}|L_{yy} & 0 \\
0 & -|1 - \frac{\theta}{\rho}|L_{yx} & \frac{1}{\tau} - L'_{xx} & 0 & -\frac{\theta}{\rho}L_{yx} \\
0 & -|1 - \frac{\theta}{\rho}|L_{yy} & 0 & \frac{1}{\sigma} - \alpha & -\frac{\theta}{\rho}L_{yy} \\
0 & 0 & -\frac{\theta}{\rho}L_{yx} & -\frac{\theta}{\rho}L_{yy} & \frac{\alpha}{\rho}
\end{pmatrix}, \tag{69}
$$

*then $G \succeq 0$ if and only if $G' \succeq 0$, where $G$ is defined in eq. (32).*

*Proof.* $\forall \mathbf{y} = (y_1, y_2, y_3, y_4, y_5)^\top \in \mathbb{R}^5$, letting $\tilde{\mathbf{y}} = (y_1, -y_2, y_3, y_4, y_5)^\top$, we have

$$
\mathbf{y}^\top G'\mathbf{y} = \begin{cases} \mathbf{y}^\top G\mathbf{y} & \text{if } \theta \leq \rho, \\ \tilde{\mathbf{y}}^\top G\tilde{\mathbf{y}} & \text{else;} \end{cases} \qquad \mathbf{y}^\top G\mathbf{y} = \begin{cases} \mathbf{y}^\top G'\mathbf{y} & \text{if } \theta \leq \rho, \\ \tilde{\mathbf{y}}^\top G'\tilde{\mathbf{y}} & \text{else.} \end{cases}
$$

Thus, $G \succeq 0$ is equivalent to $G' \succeq 0$. $\qquad\square$

**Lemma 18.** *Given $\tau, \sigma > 0$, $\theta, \alpha \geq 0$, and $\rho \in (0, 1)$, consider $G$ defined in eq. (32). If $G \succeq 0$, then $G'' \succeq 0$, where*

$$G'' \triangleq \begin{pmatrix} \frac{1}{\sigma}(1 - \frac{1}{\rho}) + \mu_y + \frac{\alpha}{\rho} & (-|1 - \frac{\theta}{\rho}| - \frac{\theta}{\rho})L_{yx} & (-|1 - \frac{\theta}{\rho}| - \frac{\theta}{\rho})L_{yy} \\ (-|1 - \frac{\theta}{\rho}| - \frac{\theta}{\rho})L_{yx} & \frac{1}{\tau} - L'_{xx} & 0 \\ (-|1 - \frac{\theta}{\rho}| - \frac{\theta}{\rho})L_{yy} & 0 & \frac{1}{\sigma} - \alpha \end{pmatrix} \succeq 0. \quad (70)$$

*Proof.* Note that $\mathbf{x}^\top G'' \mathbf{x} = {\mathbf{x}'}^\top G' \mathbf{x}' \geq 0$ for all $\mathbf{x} = [x_1 \ x_2 \ x_3]^\top \in \mathbb{R}^3$, where $\mathbf{x}' = [0 \ x_1 \ x_2 \ x_3 \ x_1]^\top$ and $G'$ is defined in (69). Then the desired result follows from Lemma 17. $\qquad\square$

Finally, with the following observation, we will be ready to proceed to the proof of Lemma 12. Let $\{\mathcal{F}_k^{t,x}\}$ and $\{\mathcal{F}_k^{t,y}\}$ be the filtrations such that $\mathcal{F}_k^{t,x} \triangleq \mathcal{F}(\{x_i^t\}_{i=0}^k, \{y_i^t\}_{i=0}^{k+1})$ and $\mathcal{F}_k^{t,y} \triangleq \mathcal{F}(\{x_i^t\}_{i=0}^k, \{y_i^t\}_{i=0}^k)$ denote the $\sigma$-algebras generated by the random variables in their arguments. A consequence of Assumption 3 is that for $\mathcal{F}_k^{t,x}$-measurable random variable $v$, i.e., $v \in \mathcal{F}_k^{t,x}$, we have that $\mathbb{E}\left[\langle \tilde{\nabla}\Phi_x(x_k^t, y_{k+1}^t; \omega_k^x) - \nabla\Phi_x(x_k^t, y_{k+1}^t), v\rangle\right] = 0$; similarly, for $v \in \mathcal{F}_k^{t,y}$, it holds that $\mathbb{E}\left[\langle \tilde{\nabla}\Phi_y(x_k^t, y_k^t; \omega_k^y) - \nabla\Phi_y(x_k^t, y_k^t), v\rangle\right] = 0$.

## C.2 Proof of Lemma 12

*Proof.* Fix arbitrary $(x, y) \in \mathbf{dom}\, f \times \mathbf{dom}\, g$. Since $(x_{k+1}^t, y_{k+1}^t) \in \mathbf{dom}\, f \times \mathbf{dom}\, g$, using the concavity of $\mathcal{L}^t(x_{k+1}^t, \cdot)$ and the convexity of $\mathcal{L}^t(\cdot, y_{k+1}^t)$, Jensen's lemma immediately implies that

$$K_N(\rho)\left(\mathcal{L}^t(\bar{x}_N^t, y) - \mathcal{L}^t(x, \bar{y}_N^t)\right) \leq \sum_{k=0}^{N-1} \rho^{-k}\left(\mathcal{L}^t(x_{k+1}^t, y) - \mathcal{L}^t(x, y_{k+1}^t)\right), \ \forall \rho \in (0, 1], \quad (71)$$

where $\bar{x}_N^t = \frac{1}{K_N(\rho)}\sum_{k=0}^{N-1}\rho^{-k}x_{k+1}^t$, $\bar{y}_N^t = \frac{1}{K_N(\rho)}\sum_{k=0}^{N-1}\rho^{-k}y_{k+1}^t$, $K_N(\rho) = \sum_{k=0}^{N-1}\rho^{-k+1}$. Thus, if we multiply both sides of (61) by $\rho^{-k}$ and sum the resulting inequality from $k = 0$ to $N - 1$, then using (71) we get

$$K_N(\rho)\left(\mathcal{L}^t(\bar{x}_N^t, y) - \mathcal{L}(x, \bar{y}_N^t)\right)$$
$$\leq \sum_{k=0}^{N-1} \rho^{-k}\Big( \underbrace{-\langle q_{k+1}^t, y_{k+1}^t - y\rangle + \theta\langle q_k^t, y_k^t - y\rangle + \Lambda_k^t(x, y) - \Sigma_{k+1}^t(x, y) + \Gamma_{k+1}^t}_{\textbf{part 1}}$$
$$\underbrace{-\langle \tilde{\nabla}_x\Phi^t(x_k^t, y_{k+1}^t; \omega_k^x) - \nabla_x\Phi^t(x_k^t, y_{k+1}^t), x_{k+1}^t - x\rangle}_{\textbf{part 2}} + \underbrace{\langle \tilde{s}_k^t - s_k^t, y_{k+1}^t - y\rangle}_{\textbf{part 3}} \Big). \quad (72)$$

Using Cauchy–Schwarz inequality and (60) leads to

$$|\langle q_{k+1}^t, y_{k+1}^t - y\rangle| \leq S_{k+1}^t(x, y) \triangleq L_{yx}\|x_{k+1}^t - x_k^t\|\|y_{k+1}^t - y\| + L_{yy}\|y_{k+1}^t - y_k^t\|\|y_{k+1}^t - y\| \quad (73)$$

for $k \geq -1$. Recall $x_{-1}^t = x_0^t$, $y_{-1}^t = y_0^t$, thus $q_0 = \mathbf{0}$; therefore, for **part 1**,

$$\sum_{k=0}^{N-1} \rho^{-k}(\theta\langle q_k^t, y_k^t - y\rangle - \langle q_{k+1}^t, y_{k+1}^t - y\rangle) = \sum_{k=0}^{N-2} \rho^{-k}\left(\frac{\theta}{\rho} - 1\right)\langle q_{k+1}^t, y_{k+1}^t - y\rangle - \rho^{-N+1}\langle q_N^t, y_N^t - y\rangle \quad (74)$$

$$\leq \sum_{k=0}^{N-2} \rho^{-k}|1 - \frac{\theta}{\rho}| S_{k+1}^t(x, y) + \rho^{-N+1}S_N^t(x, y) = \sum_{k=0}^{N-1} \rho^{-k}|1 - \frac{\theta}{\rho}| S_{k+1}^t(x, y) + \rho^{-N+1}\frac{\theta}{\rho}S_N^t(x, y),$$

where the first inequality follows from eq. (73).

Next, letting $\Delta_k^{t,x}$ and $\hat{x}_{k+1}$ be defined as in Lemma 15, we equivalently write **part 2** as

$$\sum_{k=0}^{N-1} -\rho^{-k}\langle \Delta_k^{t,x}, x_{k+1}^t - x\rangle = \sum_{k=0}^{N-1} \rho^{-k}\left(\langle \Delta_k^{t,x}, \hat{x}_{k+1}^t - x_{k+1}^t\rangle - \langle \Delta_k^{t,x}, \hat{x}_{k+1}^t - x\rangle\right). \quad (75)$$

Moreover, for $\Delta_k^{t,y}$, $\hat{y}_{k+1}^t$ and $\hat{\hat{y}}_{k+1}^t$ defined as in Lemma 15, we also equivalently write **part 3** as

$$\sum_{k=0}^{N-1} \rho^{-k} \langle \tilde{s}_k^t - s_k, y_{k+1}^t - y \rangle$$

$$= \sum_{k=0}^{N-1} \rho^{-k} \Big[ (1+\theta)\langle \Delta_k^{t,y}, y_{k+1}^t - \hat{y}_{k+1}^t + \hat{y}_{k+1}^t - y \rangle - \theta \langle \Delta_{k-1}^{t,y}, y_{k+1}^t - \hat{\hat{y}}_{k+1}^t + \hat{\hat{y}}_{k+1}^t - y \rangle \Big]. \tag{76}$$

Adding $\rho^{-N+1} D_N^t(x,y)$ to both sides of (72), then using (74), (75) and (76), for any fixed $(x,y) \in \mathcal{X} \times \mathcal{Y}$, we get

$$K_N(\rho) \left( \mathcal{L}^t(\bar{x}_N^t, y) - \mathcal{L}^t(x, \bar{y}_N^t) \right) + \rho^{-N+1} D_N^t(x,y) \le U_N^t(x,y) + \sum_{k=0}^{N-1} \rho^{-k} (P_k^t(x,y) + Q_k^t), \tag{77}$$

where $U_N^t(x,y)$, $D_N^t(x,y)$ are defined as

$$U_N^t(x,y) \triangleq \sum_{k=0}^{N-1} \rho^{-k} \left( \Gamma_{k+1}^t + \Lambda_k^t(x,y) - \Sigma_{k+1}(x,y) + |1 - \frac{\theta}{\rho}| S_{k+1}^t(x,y) \right)$$

$$- \rho^{-N+1} \left( -D_N^t(x,y) - \frac{\theta}{\rho} S_N^t(x,y) \right), \tag{78a}$$

$$D_N^t(x,y) \triangleq \frac{1}{2\rho} \left( \frac{1}{\tau} - \mu_x \right) \|x_N^t - x\|^2 + \frac{1}{2\rho} \left( \frac{1}{\sigma} - \alpha \right) \|y_N^t - y\|^2, \tag{78b}$$

and $P_k^t(x,y)$, $Q_k^t$ for $k = 0, \cdots, N-1$ are defined as

$$P_k^t(x,y) \triangleq - \langle \Delta_k^{t,x}, \hat{x}_{k+1}^t - x \rangle + (1+\theta)\langle \Delta_k^{t,y}, \hat{y}_{k+1}^t - y \rangle - \theta \langle \Delta_{k-1}^{t,y}, \hat{\hat{y}}_{k+1}^t - y \rangle, \tag{79a}$$

$$Q_k^t \triangleq \langle \Delta_k^{t,x}, \hat{x}_{k+1}^t - x_{k+1}^t \rangle + (1+\theta)\langle \Delta_k^{t,y}, y_{k+1}^t - \hat{y}_{k+1}^t \rangle - \theta \langle \Delta_{k-1}^{t,y}, y_{k+1}^t - \hat{\hat{y}}_{k+1}^t \rangle. \tag{79b}$$

For any fixed $(x,y) \in \mathcal{X} \times \mathcal{Y}$, we first analyze $U_N^t(x,y)$. After adding and subtracting $\frac{\alpha}{2}\|y_{k+1}^t - y_k^t\|^2$, and rearranging the terms, we get

$$U_N^t(x,y) = \frac{1}{2} \sum_{k=0}^{N-1} \rho^{-k} \left( \xi_k^\top A \xi_k - \xi_{k+1}^\top B \xi_{k+1} \right) - \rho^{-N+1} (-D_N^t(x,y) - \frac{\theta}{\rho} S_N^t(x,y))$$

$$= \frac{1}{2} \xi_0^\top A \xi_0 - \frac{1}{2} \sum_{k=1}^{N-1} \rho^{-k+1} [\xi_k^\top (B - \frac{1}{\rho} A) \xi_k] - \rho^{-N+1} \left( \frac{1}{2} \xi_N^\top B \xi_N - D_N^t(x,y) - \frac{\theta}{\rho} S_N^t(x,y) \right), \tag{80}$$

where $A, B \in \mathbb{R}^{5\times5}$ and $\xi_k \in \mathbb{R}^5$ are defined for $k \ge 0$ as follows: $\xi_k \triangleq \begin{pmatrix} \|x_k^t - x\| \\ \|y_k^t - y\| \\ \|x_k^t - x_{k-1}^t\| \\ \|y_k^t - y_{k-1}^t\| \\ \|y_{k+1}^t - y_k^t\| \end{pmatrix}$ such

that $x_{-1}^t = x_0^t$, $y_{-1}^t = y_0^t$, and

$$A \triangleq \begin{pmatrix} \frac{1}{\tau} - \mu_x & 0 & 0 & 0 & 0 \\ 0 & \frac{1}{\sigma} & 0 & 0 & 0 \\ 0 & 0 & 0 & 0 & \theta L_{yx} \\ 0 & 0 & 0 & 0 & \theta L_{yy} \\ 0 & 0 & \theta L_{yx} & \theta L_{yy} & -\alpha \end{pmatrix}, \quad B \triangleq \begin{pmatrix} \frac{1}{\tau} & 0 & 0 & 0 & 0 \\ 0 & \frac{1}{\sigma} + \mu_y & -|1 - \frac{\theta}{\rho}| L_{yx} & -|1 - \frac{\theta}{\rho}| L_{yy} & 0 \\ 0 & -|1 - \frac{\theta}{\rho}| L_{yx} & \frac{1}{\tau} - L_{xx}' & 0 & 0 \\ 0 & -|1 - \frac{\theta}{\rho}| L_{yy} & 0 & \frac{1}{\sigma} - \alpha & 0 \\ 0 & 0 & 0 & 0 & 0 \end{pmatrix}.$$

In Lemma 17 we show that eq. (32) is equivalent to $B - \frac{1}{\rho} A \succeq 0$; therefore, it follows from (80) that for any given $(x,y) \in \mathcal{X} \times \mathcal{Y}$,

$$U_N^t(x,y) \le \frac{1}{2} \xi_0^\top A \xi_0 - \rho^{-N+1} (\frac{1}{2} \xi_N^\top B \xi_N - D_N^t(x,y) - \frac{\theta}{\rho} S_N^t(x,y)), \text{ holds w.p. 1.}$$

Furthermore, we have

$$\frac{1}{2}\xi_N^\top B\xi_N - D_N^t(x,y) - \frac{\theta}{\rho}S_N^t(x,y) = \frac{1}{2}\xi_N^\top \begin{pmatrix} \frac{1}{\tau}(1-\frac{1}{\rho}) + \frac{\mu_x}{\rho} & \mathbf{0}_{1\times 3} & 0 \\ \mathbf{0}_{3\times 1} & G'' & \mathbf{0}_{3\times 1} \\ 0 & \mathbf{0}_{1\times 3} & 0 \end{pmatrix}\xi_N \geq 0,$$

which follows from eq. (32) and Lemma 18, where $G''$ is defined in eq. (70). Finally,

$$\frac{1}{2}\xi_0^\top A\xi_0 \leq \frac{1}{2\tau}\|x - x_0^t\|^2 + \frac{1}{2\sigma}\|y - y_0^t\|^2.$$

Thus, for any $(x,y) \in \mathcal{X} \times \mathcal{Y}$,

$$U_N^t(x,y) \leq \frac{1}{2\tau}\|x - x_0^t\|^2 + \frac{1}{2\sigma}\|y - y_0^t\|^2, \quad \text{w.p. } 1. \tag{81}$$

Now, we are ready to show eq. (55). It follows from eq. (77) and eq. (81) that, for any $(x,y) \in \mathcal{X} \times \mathcal{Y}$,

$$K_N(\rho)\Big(\mathcal{L}^t(\bar{x}_N^t, y) - \mathcal{L}^t(x, \bar{y}_N^t)\Big) + \rho^{-N+1}D_N^t(x,y)$$
$$\leq \frac{1}{2\tau}\|x - x_0^t\|^2 + \frac{1}{2\sigma}\|y - y_0^t\|^2 + \sum_{k=0}^{N-1}\rho^{-k}(P_k^t(x,y) + Q_k^t). \tag{82}$$

Let $(x_*^t, y_*^t)$ be the unique saddle point of $\mathcal{L}^t$. If we substitute $(x,y) = (x_*^t, y_*^t)$ into eq. (82) and use the fact $\mathcal{L}^t(\bar{x}_N^t, y_*^t) - \mathcal{L}^t(x_*^t, \bar{y}_N^t) \geq 0$, we obtain that

$$\rho^{-N+1}D_N^t(x_*^t, y_*^t) \leq \frac{1}{2\tau}\|x_*^t - x_0^t\|^2 + \frac{1}{2\sigma}\|y_*^t - y_0^t\|^2 + \sum_{k=0}^{N-1}\rho^{-k}(P_k^t(x_*^t, y_*^t) + Q_k^t). \tag{83}$$

From Assumption 3, for $k \geq -1$, we have

$$\mathbb{E}\left[\langle \Delta_k^{t,x}, \hat{x}_{k+1}^t - x_*^t\rangle\right] = \mathbb{E}\left[\langle \Delta_k^{t,y}, \hat{y}_{k+1}^t - y_*^t\rangle\right] = \mathbb{E}\left[\langle \Delta_{k-1}^{t,y}, \hat{y}_{k+1}^t - y_*^t\rangle\right] = 0.$$

Thus,

$$\mathbb{E}[P_k^t(x_*^t, y_*^t)] = 0.$$

Moreover, from Assumption 3, for $k \geq -1$, we have

$$\mathbb{E}\left[\|\Delta_k^{t,x}\|^2\right] \leq \delta_x^2, \quad \mathbb{E}\left[\|\Delta_k^{t,y}\|^2\right] \leq \delta_y^2.$$

Therefore, we uniformly upper bound $\mathbb{E}[Q_k^t]$ for $k \geq 0$ using Lemma 16, i.e.,

$$\mathbb{E}[\sum_{k=0}^{N-1}\rho^{-k}Q_k^t] \leq \Big(\tau\Xi_{\tau,\sigma,\theta}^x\delta_x^2 + \sigma\Xi_{\tau,\sigma,\theta}^y\delta_y^2\Big)\sum_{k=0}^{N-1}\rho^{-k},$$

where $\Xi_{\tau,\sigma,\theta}^x$ and $\Xi_{\tau,\sigma,\theta}^y$ are defined in (18a) and (18b). Therefore, combining this result with $\mathbb{E}[P_k^t(x_*^t, y_*^t)] = 0$ for any $k \in \{0, \ldots, N-1\}$, we get

$$\mathbb{E}[\sum_{k=0}^{N-1}\rho^{-k}(P_k^t(x_*^t, y_*^t) + Q_k^t)] \leq \sum_{k=0}^{N-1}\rho^{-k}\Big(\tau\Xi_{\tau,\sigma,\theta}^x\delta_x^2 + \sigma\Xi_{\tau,\sigma,\theta}^y\delta_y^2\Big). \tag{84}$$

Then, using the definition of $D_N^t(x_*^t, y_*^t)$ in eq. (78b) and the fact

$$\sum_{k=0}^{N-1}\rho^{-k} = \rho^{-N+1}\frac{1-\rho^N}{1-\rho} \leq \rho^{-N+1}\frac{1}{1-\rho},$$

for any $\rho \in (0,1)$, the desired inequality in (55) follows from (83) and (84). $\qquad\square$

## C.3 Proof of Lemma 2

Throughout this proof, our analysis is based on the proof of Lemma 12. To analyze the expected gap in Lemma 2, we consider the setting with $\rho = 1$, which implies that $K_N(\rho) = N$ and $\bar{x}_N^t = \frac{1}{N} \sum_{k=0}^{N-1} x_{k+1}^t$, $\bar{y}_N^t = \frac{1}{N} \sum_{k=0}^{N-1} y_{k+1}^t$. The proof of Lemma 2 is different than that of Lemma 12 in the way we analyze the variance terms. To be precise, we construct the auxiliary sequences –see $\tilde{x}_k$, $\tilde{y}_k^+$, $\tilde{y}_k^-$ defined in eq. (88) and eq. (91) –for the analysis of **part 2** and **part 3** in eq. (72) to provide guarantees on the expected gap function.

*Proof.* Fix arbitrary $(x, y) \in \mathbf{dom}\, f \times \mathbf{dom}\, g$. Since $(x_{k+1}^t, y_{k+1}^t) \in \mathbf{dom}\, f \times \mathbf{dom}\, g$, using the concavity of $\mathcal{L}^t(x_{k+1}^t, \cdot)$ and the convexity of $\mathcal{L}^t(\cdot, y_{k+1}^t)$, Jensen's lemma immediately implies that

$$N\left(\mathcal{L}^t(\bar{x}_N^t, y) - \mathcal{L}^t(x, \bar{y}_N^t)\right) \leq \sum_{k=0}^{N-1} \left(\mathcal{L}^t(x_{k+1}^t, y) - \mathcal{L}^t(x, y_{k+1}^t)\right), \tag{85}$$

where $\bar{x}_N^t = \frac{1}{N} \sum_{k=0}^{N-1} x_{k+1}^t$, $\bar{y}_N^t = \frac{1}{N} \sum_{k=0}^{N-1} y_{k+1}^t$. Summing eq. (61) from $k = 0$ to $N-1$ and using (85), we get

$$N\left(\mathcal{L}^t(\bar{x}_N^t, y) - \mathcal{L}(x, \bar{y}_N^t)\right)$$

$$\leq \sum_{k=0}^{N-1} \underbrace{-\langle q_{k+1}^t, y_{k+1}^t - y\rangle + \theta\langle q_k^t, y_k^t - y\rangle}_{\textbf{part 1}} + \Lambda_k^t(x, y) - \Sigma_{k+1}^t(x, y) + \Gamma_{k+1}^t$$

$$\underbrace{-\langle \tilde{\nabla}_x \Phi^t(x_k^t, y_{k+1}^t; \omega_k^x) - \nabla_x \Phi^t(x_k^t, y_{k+1}^t), x_{k+1}^t - x\rangle}_{\textbf{part 2}} + \underbrace{\langle \tilde{s}_k^t - s_k^t, y_{k+1}^t - y\rangle}_{\textbf{part 3}}. \tag{86}$$

The bound on **Part 1** immediately follows from eq. (74) with $\rho = 1$, i.e.,

$$\sum_{k=0}^{N-1} \theta\langle q_k^t, y_k^t - y\rangle - \langle q_{k+1}^t, y_{k+1}^t - y\rangle \leq \sum_{k=0}^{N-1} |1-\theta|\, S_{k+1}^t(x, y) + \theta S_N^t(x, y). \tag{87}$$

Recall that $x_{-1}^t = x_0^t$ and $y_{-1}^t = y_0^t$; thus, $q_0^t = \mathbf{0}$.

Next we consider **part 2**, let $\Delta_k^{t,x}$ be defined as in Lemma 15. For some arbitrary $\eta_x > 0$, define $\{\tilde{x}_k\}$ sequence as follows:

$$\tilde{x}_0 \triangleq x_0^t, \quad \tilde{x}_{k+1} \triangleq \underset{x' \in \mathcal{X}}{\mathrm{argmin}} -\langle \Delta_k^{t,x}, x'\rangle + \frac{\eta_x}{2}\|x' - \tilde{x}_k\|^2, \quad \forall\, k \geq 0. \tag{88}$$

Then by [29, Lemma 2.1], for all $k \geq 0$ and $x \in \mathcal{X}$, we have that

$$\langle \Delta_k^{t,x}, x - \tilde{x}_k\rangle \leq \frac{\eta_x}{2}\|x - \tilde{x}_k\|^2 - \frac{\eta_x}{2}\|x - \tilde{x}_{k+1}\|^2 + \frac{1}{2\eta_x}\|\Delta_k^{t,x}\|^2.$$

Thus, using $\tilde{x}_0 = x_0^t$ we get

$$\sum_{k=0}^{N-1} \langle \Delta_k^{t,x}, x - \tilde{x}_k\rangle \leq \sum_{k=0}^{N-1} \left(\frac{\eta_x}{2}\|x - \tilde{x}_k\|^2 - \frac{\eta_x}{2}\|x - \tilde{x}_{k+1}\|^2 + \frac{1}{2\eta_x}\|\Delta_k^{t,x}\|^2\right)$$

$$= \frac{\eta_x}{2}(\|x - x_0^t\|^2 - \|x - \tilde{x}_N\|^2) + \sum_{k=0}^{N-1} \frac{1}{2\eta_x}\|\Delta_k^{t,x}\|^2 \leq \frac{\eta_x}{2}\|x - x_0^t\|^2 + \frac{1}{2\eta_x}\sum_{k=0}^{N-1}\|\Delta_k^{t,x}\|^2; \tag{89}$$

hence, **part 2** becomes

$$\sum_{k=0}^{N-1} \langle \Delta_k^{t,x}, x - x_{k+1}^t\rangle$$

$$= \sum_{k=0}^{N-1} \langle \Delta_k^{t,x}, \hat{x}_{k+1}^t - x_{k+1}^t\rangle - \langle \Delta_k^{t,x}, \hat{x}_{k+1} - \tilde{x}_k\rangle + \langle \Delta_k^{t,x}, x - \tilde{x}_k\rangle \tag{90}$$

$$\leq \frac{\eta_x}{2}\|x - x_0^t\|^2 + \sum_{k=0}^{N-1} \langle \Delta_k^{t,x}, \hat{x}_{k+1}^t - x_{k+1}^t\rangle - \langle \Delta_k^{t,x}, \hat{x}_{k+1} - \tilde{x}_k\rangle + \frac{1}{2\eta_x}\|\Delta_k^{t,x}\|^2,$$

which follows from eq. (89), and $\hat{x}_{k+1}$ is defined in Lemma 15.[6]

---

[6]When $\delta_x = 0$, clearly $\Delta_k^{t,x} = \mathbf{0}$; thus, **part 2** is equal to 0 and we can set $\eta_x = 0$ for which (90) becomes $0 \leq 0$.

Next, we consider **part 3**, let $\Delta_k^{t,y}$ be defined as in Lemma 15. For some arbitrary $\eta_y > 0$, we construct two auxiliary sequences: let $\tilde{y}_0^+ = \tilde{y}_0^- = y_0^t$, and for $k \geq 0$, we define

$$\tilde{y}_{k+1}^+ \triangleq \operatorname*{argmin}_{y' \in \mathcal{Y}} \langle \Delta_k^{t,y}, y' \rangle + \frac{\eta_y}{2} \|y' - \tilde{y}_k^+\|^2, \quad \tilde{y}_{k+1}^- \triangleq \operatorname*{argmin}_{y' \in \mathcal{Y}} -\langle \Delta_k^{t,y}, y' \rangle + \frac{\eta_y}{2} \|y' - \tilde{y}_k^-\|^2. \quad (91)$$

Thus, it follows from [29, Lemma 2.1] that for $y \in \mathcal{Y}$,

$$\langle \Delta_k^{t,y}, \tilde{y}_k^+ - y \rangle \leq \frac{\eta_y}{2} \|y - \tilde{y}_k^+\|^2 - \frac{\eta_y}{2} \|y - \tilde{y}_{k+1}^+\|^2 + \frac{1}{2\eta_y} \|\Delta_k^{t,y}\|^2,$$

$$\langle \Delta_k^{t,y}, y - \tilde{y}_k^- \rangle \leq \frac{\eta_y}{2} \|y - \tilde{y}_k^-\|^2 - \frac{\eta_y}{2} \|y - \tilde{y}_{k+1}^-\|^2 + \frac{1}{2\eta_y} \|\Delta_k^{t,y}\|^2.$$

Therefore, as in (89), we get[7]

$$\sum_{k=0}^{N-1} (1+\theta) \langle \Delta_k^{t,y}, \tilde{y}_k^+ - y \rangle + \theta \langle \Delta_{k-1}^{t,y}, y - \tilde{y}_{k-1}^- \rangle$$
$$\leq \frac{\eta_y}{2}(1+2\theta)\|y - y_0^t\|^2 + \frac{1}{2\eta_y} \sum_{k=0}^{N-1} \left( (1+\theta)\|\Delta_k^{t,y}\|^2 + \theta\|\Delta_{k-1}^{t,y}\|^2 \right). \quad (92)$$

Next, using eq. (92), we can bound **part 3** as follows:

$$\sum_{k=0}^{N-1} \langle \tilde{s}_k^t - s_k^t, y_{k+1}^t - y \rangle$$
$$= \sum_{k=0}^{N-1} (1+\theta) \langle \Delta_k^{t,y}, y_{k+1}^t - \hat{y}_{k+1}^t + \hat{y}_{k+1}^t - \tilde{y}_k^+ + \tilde{y}_k^+ - y \rangle - \theta \langle \Delta_{k-1}^{t,y}, y_{k+1}^t - \hat{y}_{k+1}^t + \hat{y}_{k+1}^t - \tilde{y}_{k-1}^- + \tilde{y}_{k-1}^- - y \rangle$$
$$\leq \sum_{k=0}^{N-1} (1+\theta) \langle \Delta_k^{t,y}, y_{k+1}^t - \hat{y}_{k+1}^t + \hat{y}_{k+1}^t - \tilde{y}_k^+ \rangle - \theta \langle \Delta_{k-1}^{t,y}, y_{k+1}^t - \hat{y}_{k+1}^t + \hat{y}_{k+1}^t - \tilde{y}_{k-1}^- \rangle$$
$$+ \frac{1}{2\eta_y} \sum_{k=0}^{N-1} \left( (1+\theta)\|\Delta_k^{t,y}\|^2 + \theta\|\Delta_{k-1}^{t,y}\|^2 \right) + \frac{\eta_y}{2}(1+2\theta)\|y - y_0^t\|^2, \quad (93)$$

where $\hat{y}_{k+1}^t$ and $\hat{\hat{y}}_{k+1}^t$ are defined in Lemma 15.

For any fixed $(x, y) \in \mathbf{dom}\, f \times \mathbf{dom}\, g$, we use (87), (90) and (93) to get

$$N\left( \mathcal{L}^t(\bar{x}_N^t, y) - \mathcal{L}^t(x, \bar{y}_N^t) \right)$$
$$\leq \tilde{U}_N^t(x,y) + \frac{\eta_x}{2}\|x - x_0^t\|^2 + \frac{\eta_y}{2}(1+2\theta)\|y - y_0^t\|^2 + \sum_{k=0}^{N-1} (\tilde{P}_k^t + \tilde{Q}_k^t), \quad (94)$$

where $\tilde{U}_N^t(x,y)$ and $\tilde{P}_k^t, \tilde{Q}_k^t$ for $k = 0, \dots, N-1$ are defined as follows:

$$\tilde{U}_N^t(x,y) \triangleq \sum_{k=0}^{N-1} \left( \Gamma_{k+1}^t + \Lambda_k^t(x,y) - \Sigma_{k+1}^t(x,y) + |1-\theta|\, S_{k+1}^t(x,y) \right) + \theta S_N^t(x,y), \quad (95a)$$

$$\tilde{P}_k^t \triangleq -\langle \Delta_k^{t,x}, \hat{x}_{k+1}^t - \tilde{x}_k \rangle + (1+\theta)\langle \Delta_k^{t,y}, \hat{y}_{k+1}^t - \tilde{y}_k^+ \rangle - \theta\langle \Delta_{k-1}^{t,y}, \hat{\hat{y}}_{k+1}^t - \tilde{y}_{k-1}^- \rangle, \quad (95b)$$

$$\tilde{Q}_k^t \triangleq \langle \Delta_k^{t,x}, \hat{x}_{k+1}^t - x_{k+1}^t \rangle + (1+\theta)\langle \Delta_k^{t,y}, y_{k+1}^t - \hat{y}_{k+1}^t \rangle - \theta\langle \Delta_{k-1}^{t,y}, y_{k+1}^t - \hat{\hat{y}}_{k+1}^t \rangle$$
$$+ \frac{1}{2\eta_x}\|\Delta_k^{t,x}\|^2 + \frac{1+\theta}{2\eta_y}\|\Delta_k^{t,y}\|^2 + \frac{\theta}{2\eta_y}\|\Delta_{k-1}^{t,y}\|^2. \quad (95c)$$

The remaining part of the analysis directly follows from the arguments we used in the proof of Lemma 12. For any fixed $(x, y) \in \mathcal{X} \times \mathcal{Y}$, we first analyze $\tilde{U}_N^t(x,y)$. For some given $\alpha > 0$, after

---

[7]As in **part 2**, when $\delta_y = 0$, we can set $\eta_y = 0$.

adding and subtracting $\frac{\alpha}{2}\|y_{k+1}^t - y_k^t\|^2$, and rearranging the terms, we get

$$\tilde{U}_N^t(x,y) = \frac{1}{2}\sum_{k=0}^{N-1}\left(\xi_k^\top \tilde{A}\xi_k - \xi_{k+1}^\top \tilde{B}\xi_{k+1}\right) + \theta S_N^t(x,y)$$

$$= \frac{1}{2}\xi_0^\top \tilde{A}\xi_0 - \frac{1}{2}\sum_{k=1}^{N-1}[\xi_k^\top(\tilde{B}-\tilde{A})\xi_k] - \left(\frac{1}{2}\xi_N^\top \tilde{B}\xi_N - \theta S_N^t(x,y)\right), \tag{96}$$

where $A, B \in \mathbb{R}^{5\times5}$ and $\xi_k \in \mathbb{R}^5$ are defined for $k \geq 0$ as follows:

$$\xi_k \triangleq \begin{pmatrix} \|x_k^t - x\| \\ \|y_k^t - y\| \\ \|x_k^t - x_{k-1}^t\| \\ \|y_k^t - y_{k-1}^t\| \\ \|y_{k+1}^t - y_k^t\| \end{pmatrix}, \qquad \tilde{A} \triangleq \begin{pmatrix} \frac{1}{\tau} - \mu_x & 0 & 0 & 0 & 0 \\ 0 & \frac{1}{\sigma} & 0 & 0 & 0 \\ 0 & 0 & 0 & 0 & \theta L_{yx} \\ 0 & 0 & 0 & 0 & \theta L_{yy} \\ 0 & 0 & \theta L_{yx} & \theta L_{yy} & -\alpha \end{pmatrix},$$

and

$$\tilde{B} \triangleq \begin{pmatrix} \frac{1}{\tau} & 0 & 0 & 0 & 0 \\ 0 & \frac{1}{\sigma} + \mu_y & -|1-\theta|\,L_{yx} & -|1-\theta|\,L_{yy} & 0 \\ 0 & -|1-\theta|\,L_{yx} & \frac{1}{\tau} - L'_{xx} & 0 & 0 \\ 0 & -|1-\theta|\,L_{yy} & 0 & \frac{1}{\sigma} - \alpha & 0 \\ 0 & 0 & 0 & 0 & 0 \end{pmatrix}$$

such that $x_{-1}^t = x_0^t$, $y_{-1}^t = y_0^t$. Lemma 17 together with $\rho = 1$ implies that eq. (16) is equivalent to $\tilde{B} - \tilde{A} \succeq 0$; therefore, it follows from (96) that, for any given $(x,y) \in \mathcal{X} \times \mathcal{Y}$, we have

$$U_N^t(x,y) \leq \frac{1}{2}\xi_0^\top \tilde{A}\xi_0 - \left(\frac{1}{2}\xi_N^\top \tilde{B}\xi_N - \theta S_N^t(x,y)\right), \quad \text{w.p. 1.}$$

Furthermore, we also have

$$\frac{1}{2}\xi_N^\top \tilde{B}\xi_N - \theta S_N^t(x,y) \geq \frac{1}{2}\xi_N^\top \begin{pmatrix} \mu_x & \mathbf{0}_{1\times3} & 0 \\ \mathbf{0}_{3\times1} & G'' & \mathbf{0}_{3\times1} \\ 0 & \mathbf{0}_{1\times3} & 0 \end{pmatrix}\xi_N \geq 0,$$

which follows from eq. (16) and Lemma 18 with $\rho = 1$, where $G''$ is defined in eq. (70). Finally,

$$\frac{1}{2}\xi_0^\top \tilde{A}\xi_0 \leq \frac{1}{2\tau}\|x - x_0^t\|^2 + \frac{1}{2\sigma}\|y - y_0^t\|^2.$$

Thus, the above three inequalities imply that, for any $(x,y) \in \mathcal{X} \times \mathcal{Y}$,

$$\tilde{U}_N^t(x,y) \leq \frac{1}{2\tau}\|x - x_0^t\|^2 + \frac{1}{2\sigma}\|y - y_0^t\|^2, \quad \text{w.p. 1.} \tag{97}$$

Now, we are ready to show eq. (17). It follows from eq. (94) and eq. (97) that

$$N \sup_{(x,y)\in\mathcal{X}\times\mathcal{Y}}\left\{\mathcal{L}^t(\bar{x}_N^t, y) - \mathcal{L}^t(x, \bar{y}_N^t)\right\}$$

$$\leq \left(\frac{1}{2\tau} + \frac{\eta_x}{2}\right)\|x_*^t(\bar{y}_N^t) - x_0^t\|^2 + \left(\frac{1}{2\sigma} + \frac{\eta_y(1+2\theta)}{2}\right)\|y_*(\bar{x}_N^t) - y_0^t\|^2 + \sum_{k=0}^{N-1}(\tilde{P}_k^t + \tilde{Q}_k^t), \tag{98}$$

where $(x_*^t(\bar{y}_N^t), y_*(\bar{x}_N^t))$ is the point achieving the supremum on the left hand side. Indeed, to derive the above inequality, we substitute $(x,y) = (x_*^t(\bar{y}_N^t), y_*(\bar{x}_N^t))$ into the eq. (94) and use the fact that

$$\sup_{(x,y)\in\mathcal{X}\times\mathcal{Y}}\left\{\mathcal{L}^t(\bar{x}_N^t, y) - \mathcal{L}^t(x, \bar{y}_N^t)\right\} = \mathcal{L}^t(\bar{x}_N^t, y_*(\bar{x}_N^t)) - \mathcal{L}^t(x_*^t(\bar{y}_N^t), \bar{y}_N^t).$$

From Assumption 3, for $k \geq -1$, we have

$$\mathbb{E}\left[\langle \Delta_k^{t,x}, \hat{x}_{k+1}^t - \tilde{x}_k\rangle\right] = \mathbb{E}\left[\langle \Delta_k^{t,y}, \hat{y}_{k+1}^t - \tilde{y}_k^\pm\rangle\right] = \mathbb{E}\left[\langle \Delta_{k-1}^{t,y}, \hat{y}_{k+1}^t - \tilde{y}_{k-1}^-\rangle\right] = 0.$$

Thus, $\mathbb{E}[\tilde{P}_k^t] = 0$. Moreover, for $k \geq -1$, from Assumption 3 we also have

$$\mathbb{E}\left[\|\Delta_k^{t,x}\|^2\right] \leq \delta_x^2, \qquad \mathbb{E}\left[\|\Delta_k^{t,y}\|^2\right] \leq \delta_y^2.$$

Next, we uniformly upper bound $\mathbb{E}\left[\tilde{Q}_k^t\right]$ for $k \geq 0$ using Lemma 16, i.e.,

$$\mathbb{E}[\sum_{k=0}^{N-1} \tilde{Q}_k^t] \leq N\left[\left(\tau \Xi_{\tau,\sigma,\theta}^x + \frac{1}{2\eta_x}\right)\delta_x^2 + \left(\sigma \Xi_{\tau,\sigma,\theta}^y + \frac{1+2\theta}{2\eta_y}\right)\delta_y^2\right].$$

Therefore, combining this result with $\mathbb{E}[\tilde{P}_k^t] = 0$ for any $k \in \{0, \ldots, N-1\}$, we get

$$\mathbb{E}[\sum_{k=0}^{N-1}(\tilde{P}_k^t + \tilde{Q}_k^t)] \leq N\,\Xi_{\tau,\sigma,\theta}. \tag{99}$$

Finally, setting $\eta_x = \frac{1}{\tau}$, $\eta_y = \frac{1}{\sigma}$, $x_0^{t+1} = \bar{x}_N^t$, and $y_0^{t+1} = \bar{y}_N^t$, the desired result in (17) follows from (98) and (99). $\qquad\square$

# D  Computation of $\epsilon$-stationary point in practice

In this section, we discuss how to compute a point $x_\epsilon$ such that $\mathbb{E}[\|\nabla\phi_\lambda(x_\epsilon)\|] \leq \epsilon$ –as the $t_*$ in remark 5 can not be computed in practice. This result is shown in Theorem 7, which directly follows from Theorem 1 and Lemma 10. Below, for the sake of completeness, we state a known technical result that we need for the proof of Theorem 7.

**Lemma 19.** *Suppose Assumptions 1 and 2 hold. Then $\phi_\lambda(\cdot)$ is $\frac{1}{\lambda}$-smooth for $\lambda \in (0, \gamma^{-1})$, where $\phi_\lambda(\cdot)$ is defined in definition 3 for $\phi(\cdot) = \max_{y \in \mathcal{Y}} \mathcal{L}(\cdot, y)$.*

*Proof.* Let $R(x) \triangleq x - \mathbf{prox}_{\lambda\phi}(x)$ for $\lambda \in (0, \gamma^{-1})$ and $x \in \mathbf{dom}\, f$. Indeed, by definition 3, we know $R(x) = \lambda\nabla\phi_\lambda(x)$. Then by the optimality condition of $\mathbf{prox}_{\lambda\phi}(x)$, we obtain that

$$R(x) \in \partial f(\mathbf{prox}_{\lambda\phi}(x))$$

holds for $x \in \mathbf{dom}\, f$. Hence, for $x_1, x_2 \in \mathbf{dom}\, f$, we have that

$$\langle R(x_1) - R(x_2), \mathbf{prox}_{\lambda\phi}(x_1) - \mathbf{prox}_{\lambda\phi}(x_2)\rangle \geq 0,$$

which further implies that

$$\begin{aligned}
\|x_1 - x_2\|^2 &= \|R(x_1) - \mathbf{prox}_{\lambda\phi}(x_1) - R(x_2) + \mathbf{prox}_{\lambda\phi}(x_2)\|^2 \\
&\geq \|R(x_1) - R(x_2)\|^2 + \|\mathbf{prox}_{\lambda\phi}(x_1) - \mathbf{prox}_{\lambda\phi}(x_2)\|^2 \\
&\geq \|R(x_1) - R(x_2)\|^2.
\end{aligned}$$

Then using the fact $R(x) = \lambda\nabla\phi_\lambda(x)$ completes the proof. $\qquad\square$

**Theorem 7.** *Consider $\mathcal{L}$ defined in (1). Suppose Assumptions 1, 2, 3 hold. Under the premise of Theorem 1, for any $\epsilon > 0$, SAPD+ can generate an point $x_\epsilon$ such that $\mathbb{E}[\|\nabla\phi_\lambda(x_\epsilon)\|] \leq \epsilon$ within $\mathcal{O}\left(\frac{L\kappa_y \mathcal{G}(x_0^0, y_0^0)}{\epsilon^2}\ln(1/\epsilon) + \frac{L\kappa_y \delta^2 \mathcal{G}(x_0^0, y_0^0)}{\epsilon^4}\ln(1/\epsilon)\right)$ stochastic first-order oracle calls.*

*Proof.* Under the premise of Theorem 1, given $\epsilon > 0$, SAPD+ generates $\{x_0^t\}_{t=0}^T$ such that $\min_{t=0,\ldots,T} \mathbb{E}[\|\nabla\phi_\lambda(x_0^t)\|] \leq \epsilon/4$, for $T \geq 96\mathcal{G}(x_0^0, y_0^0) \cdot \frac{16\gamma}{\epsilon^2} + 1$. Therefore, for each $x_0^t$, if we let $\hat{x}_0^t = \mathbf{prox}_{\lambda\phi}(x_0^t)$, then Lemma 10 ensures that we can generate a point $\tilde{x}_*^t$ such that

$$\mathbb{E}[\|\tilde{x}_*^t - \hat{x}_0^t\|] \leq \hat{\epsilon}$$

within $N_{\hat{\epsilon}}$ many iterations, where

$$N_{\hat{\epsilon}} = \mathcal{O}\left(\frac{\max\{L_{xx}, L_{yx}\}}{\gamma} + \frac{\max\{L_{yx}, L_{xy}\}}{\sqrt{\gamma\mu_y}} + \frac{\max\{L_{yy}, L_{yx}\}}{\mu_y} + \left(\frac{\delta_x^2}{\gamma} + \frac{\delta_y^2}{\mu_y}\right)\frac{1}{\gamma\hat{\epsilon}^2}\right)\cdot\ln\left(\frac{\max\{1, \mu_y/\gamma\}}{\hat{\epsilon}}\right)$$

Moreover, if we compute the GNME of $\phi(\cdot)$ at $\tilde{x}_*^t$, it follows that

$$
\begin{aligned}
\|\nabla\phi_\lambda(\tilde{x}_*^t)\| &\leq \|\nabla\phi_\lambda(\tilde{x}_*^t) - \nabla\phi_\lambda(\hat{x}_0^t)\| + \|\nabla\phi_\lambda(\hat{x}_*^t) - \nabla\phi_\lambda(x_0^t)\| + \|\nabla\phi_\lambda(x_0^t)\| \\
&\leq \frac{2}{\lambda}\|\tilde{x}_*^t - \hat{x}_0^t\| + \frac{1}{\lambda}\|\hat{x}_*^t - x_0^t\| + \|\nabla\phi_\lambda(x_0^t)\| \\
&= \frac{2}{\lambda}\|\tilde{x}_*^t - \hat{x}_0^t\| + 2\|\nabla\phi_\lambda(x_0^t)\| \\
&\leq \frac{2}{\lambda}\hat{\epsilon} + 2\|\nabla\phi_\lambda(x_0^t)\|.
\end{aligned}
$$

where the second inequality is by Lemma 19; the first equality is by definition 3 and the fact $\hat{x}_0^t = \mathbf{prox}_{\lambda\phi}(x_0^t)$. Furthermore, because $\min_{t=0,\ldots,T} \mathbb{E}\left[\|\nabla\phi_\lambda(x_0^t)\|\right] \leq \epsilon/4$, then we have

$$
\min_{t=0,\ldots,T} \mathbb{E}[\|\nabla\phi_\lambda(\tilde{x}_*^t)\|] \leq \frac{2}{\lambda}\hat{\epsilon} + \frac{\epsilon}{2}
$$

and we let $x_\epsilon = \tilde{x}_*^{\tilde{t}}$, where $\tilde{t}_* \triangleq \operatorname{argmin}_{\{t=0,..,T\}} \mathbb{E}[\|\nabla\phi_\lambda(\tilde{x}_*^t)\|]$. Therefore, setting $\lambda = \frac{1}{2\gamma}$ and $\hat{\epsilon} = \frac{1}{8\gamma}$, Lemma 10 implies that calling SAPD $T$ times, each with $\tilde{N}$ iterations, one can generate $x_\epsilon$ such that

$$
\mathbb{E}\left[\|\nabla\phi_\lambda(x_\epsilon)\|\right] \leq \epsilon,
$$

where $T$ is given in Theorem 1 and

$$
\tilde{N} = \mathcal{O}\left(\frac{\max\{L_{xx}, L_{yx}\}}{\gamma} + \frac{\max\{L_{yx}, L_{xy}\}}{\sqrt{\gamma\mu_y}} + \frac{\max\{L_{yy}, L_{yx}\}}{\mu_y} + \left(\frac{\delta_x^2}{\gamma} + \frac{\delta_y^2}{\mu_y}\right)\frac{\gamma}{\epsilon^2}\right)\cdot\ln\left(\frac{\max\{\gamma, \mu_y\}}{\epsilon}\right)
$$

Thus, considering the setting in (8), one can compute $x_\epsilon$ in practice requiring $T\tilde{N} = \mathcal{O}\left(\frac{L\kappa_{yy}\mathcal{G}(x_0^0,y_0^0)}{\epsilon^2}\ln(1/\epsilon) + \frac{L\kappa_{yy}\delta^2\mathcal{G}(x_0^0,y_0^0)}{\epsilon^4}\ln(1/\epsilon)\right)$ oracle calls; furthermore, $\ln(1/\epsilon)$ can be removed by employing a restarting strategy as in [43]. $\qquad\square$

## E  Proof of Theorem 3

For completeness, we provide a technical lemma below establishing Lipschitz continuity of the best response functions (see also [43, Lemma 2.5] and [24, Lemma B.2(a)]).

**Lemma 20.** *[7, Proposition 1] Suppose Assumptions 1 and 2 hold. For any given $y \in \mathbf{dom}\, g$, let $x_*^t(y) \triangleq \operatorname{argmin}_{x\in\mathcal{X}} \mathcal{L}^t(x,y)$; and for any given $x \in \mathbf{dom}\, f$, let $y_*(x) \triangleq \operatorname{argmax}_{y\in\mathcal{Y}} \mathcal{L}^t(x,y) = \operatorname{argmax}_{y\in\mathcal{Y}} \mathcal{L}(x,y)$. Then $x_*^t(\cdot)$ and $y_*(\cdot)$ are Lipschitz maps on $\mathbf{dom}\, g$ and $\mathbf{dom}\, f$, with constants $\kappa_{xy}$ and $\kappa_{yx}$, respectively, where $\kappa_{xy} \triangleq L_{xy}/\mu_x$ and $\kappa_{yx} \triangleq L_{yx}/\mu_y$.*

**Lemma 21.** *For any $t \geq 0$, let $z_*^t \triangleq (x_*^t, y_*^t)$ be the unique saddle point of $\mathcal{L}^t$ defined in eq. (4), and let $\{z_k^t\}_{k=0}^{N_t}$ be generated by running SAPD on $\min_{x\in\mathcal{X}}\max_{y\in\mathcal{Y}} \mathcal{L}^t(x,y)$ for $N_t \in \mathbb{Z}_+$ iterations, where $z_k^t \triangleq (x_k^t, y_k^t)$; and define $z_0^{t+1} \triangleq \frac{1}{N_t}\sum_{i=0}^{N_t-1} z_{i+1}^t$. Under the setting of Lemma 2,*

$$
\max\left\{\mathbb{E}\left[\mathcal{G}^t(z_0^{t+1})\right], \mathbb{E}\left[\|z_0^{t+1} - z_*^t\|^2\right]\right\} \leq \frac{1}{N_t}C_{\tau,\sigma,\theta}\mathbb{E}[\|z_0^t - z_*^t\|^2] + C'_{\tau,\sigma,\theta} \tag{100}
$$

*holds for all $t \geq 0$ and $N_t \geq 1$, for some positive constants $C_{\tau,\sigma,\theta}$ and $C'_{\tau,\sigma,\theta}$.*

*Proof.* For simplicity we assume $N_t = N$ for all $t \geq 0$ –the proof still holds for arbitrary $\{N_t\}_{t\geq 0} \subset \mathbb{Z}_+$. The proof mainly follows the proof of Lemma 2. We first show a bound for $\mathbb{E}\left[\|z_0^{t+1} - z_*^t\|^2\right]$ that is in the form of the rhs of eq. (100); then, we show it for $\mathbb{E}\left[\mathcal{G}^t(z_0^{t+1})\right]$. In addition, given $\{z_k^t\}_{k=0}^{N_t}$, we let $\bar{z}_{N_t}^t = (\bar{x}_{N_t}^t, \bar{y}_{N_t}^t)$, and $\bar{z}_{N_t}^t = z_0^{t+1} = \frac{1}{N_t}\sum_{i=0}^{N_t-1} z_{i+1}^t$ for all $t \geq 0$ and $N_t \geq 1$.

Now, we start with analyzing $\mathbb{E}\left[\|z_0^{t+1} - z_*^t\|^2\right]$. The analysis below mainly relies on the proof of Lemma 2. Indeed, given $z_0^t = (x_0^t, y_0^t)$ for $t \geq 0$, substituting $x = x_*^t$ and $y = y_*^t$ within (94) and then using eq. (97), we obtain that

$$
\begin{aligned}
&N\mathbb{E}\left[\mathcal{L}^t(\bar{x}_N^t, y_*^t) - \mathcal{L}^t(x_*^t, \bar{y}_N^t)\right] \\
&\leq \mathbb{E}\left[\left(\frac{1}{2\tau} + \frac{\eta_x}{2}\right)\|x_*^t - x_0^t\|^2 + \left(\frac{1}{2\sigma} + \frac{\eta_y(1+2\theta)}{2}\right)\|y_*^t - y_0^t\|^2 + \sum_{k=0}^{N-1}\rho^{-k}(\tilde{P}_k^t + \tilde{Q}_k^t)\right].
\end{aligned} \tag{101}
$$

Moreover, since $\mathcal{L}^t(\cdot, y_*^t)$ is $\mu_x$-strongly convex and $\mathcal{L}^t(x_*^t, \cdot)$ is $\mu_y$-strongly concave, and $(x_*^t, y_*^t)$ is the unique saddle point of $\mathcal{L}^t$, we have that

$$\frac{\mu_x}{2}\|\bar{x}_N^t - x_*^t\|^2 + \frac{\mu_y}{2}\|\bar{y}_N^t - y_*^t\|^2 \leq \mathcal{L}^t(\bar{x}_N^t, y_*^t) - \mathcal{L}^t(x_*^t, \bar{y}_N^t). \tag{102}$$

If we let $\eta_x = \frac{1}{\tau}$ and $\eta_y = \frac{1}{\sigma}$, then it follows from eqs. (101, 102,99) and the fact that $\bar{z}_N^t = z_0^{t+1}$ that

$$N\mathbb{E}\left[\frac{\mu_x}{2}\|x_0^{t+1} - x_*^t\|^2 + \frac{\mu_y}{2}\|y_0^{t+1} - y_*^t\|^2\right] \leq \mathbb{E}\left[\overline{U}^t(x_*^t, y_*^t)\right] + N\Xi_{\tau,\sigma,\theta}, \tag{103}$$

where $\Xi_{\tau,\sigma,\theta}$ is defined in Lemma 2 and for any $(x, y) \in \mathcal{X} \times \mathcal{Y}$, we define

$$\overline{U}^t(x, y) \triangleq \frac{1}{\tau}\|x - x_0^t\|^2 + \frac{1+\theta}{\sigma}\|y - y_0^t\|^2. \tag{104}$$

Therefore, we conclude that

$$\mathbb{E}\left[\|z_0^{t+1} - z_*^t\|^2\right] \leq \frac{1}{N}\overline{C}_{\tau,\sigma,\theta}\mathbb{E}\left[\|z_0^t - z_*^t\|^2\right] + \overline{C}'_{\tau,\sigma,\theta}, \tag{105}$$

where

$$\overline{C}_{\tau,\sigma,\theta} \triangleq \frac{2\max\{\frac{1}{\tau}, \frac{1+\theta}{\sigma}\}}{\min\{\mu_x, \mu_y\}}, \qquad \overline{C}'_{\tau,\sigma,\theta} \triangleq \frac{2}{\min\{\mu_x, \mu_y\}}\Xi_{\tau,\sigma,\theta}.$$

This completes the first part of the proof. Next, we will bound $\mathbb{E}[\mathcal{G}^t(z_0^{t+1})]$ using the bound on $\mathbb{E}[\|z_0^{t+1} - z_*^t\|^2]$ we derived in the first part.

Given $z_0^t$, using eq. (98) and eq. (99) in the proof Lemma 2 for $\eta_x = \frac{1}{\tau}$ and $\eta_y = \frac{1}{\sigma}$ as above, we obtain that

$$\mathbb{E}\left[\mathcal{G}^t(z_0^{t+1})\right] \leq \frac{1}{N}\mathbb{E}\left[\overline{U}^t\left(x_*^t(y_0^{t+1}), y_*(x_0^{t+1})\right)\right] + \Xi_{\tau,\sigma,\theta}, \tag{106}$$

where $\overline{U}^t(x, y)$ is defined in (104) and $\Xi_{\tau,\sigma,\theta}$ is defined in Lemma 2; furhermore, $x_*^t(\cdot)$ and $y_*(\cdot)$ are defined in eq. (6). Next, we will use eq. (105) to derive an upper bound for the right hand side of eq. (106).

Since $z_*^t$ is the unique saddle point for $\mathcal{L}^t$, we have $x_*^t(y_*^t) = x_*^t$ and $y_*(x_*^t) = y_*^t$. Moreover, according to Lemma 20, $x_*^t(\cdot), y_*(\cdot)$ is Lipschitz with constants $\kappa_{xy} = \frac{L_{xy}}{\mu_x}$ and $\kappa_{yx} = \frac{L_{yx}}{\mu_y}$, respectively. Therefore, Lemma 20 and

$$\overline{U}^t(x_*^t(y_0^{t+1}), y_*(x_0^{t+1})) \leq \frac{2}{\tau}\|x_*^t - x_*^t(y_0^{t+1})\|^2 + \frac{2+2\theta}{\sigma}\|y_*^t - y_*(x_0^{t+1})\|^2 + 2\overline{U}^t(x_*^t, y_*^t), \quad \text{w.p. 1,}$$

together imply that

$$\mathbb{E}\left[\overline{U}^t(x_*^t(y_0^{t+1}), y_*(x_0^{t+1}))\right]$$

$$\leq \mathbb{E}\left[\frac{2\kappa_{xy}^2}{\tau}\|y_*^t - y_0^{t+1}\|^2 + \frac{(2+2\theta)\kappa_{yx}^2}{\sigma}\|x_*^t - x_0^{t+1}\|^2 + 2\overline{U}^t(x_*^t, y_*^t)\right]$$

$$\leq \mathbb{E}\left[\max\left\{\frac{2}{\tau}, \frac{(2+2\theta)}{\sigma}\right\}\left(\max\{\kappa_{xy}^2, \kappa_{yx}^2\}\|z_0^{t+1} - z_*^t\|^2 + \|z_0^t - z_*^t\|^2\right)\right]$$

$$\leq \mathbb{E}\left[\max\left\{\frac{2}{\tau}, \frac{(2+2\theta)}{\sigma}\right\}\max\{1, \kappa_{xy}^2, \kappa_{yx}^2\}\left(\left(\frac{1}{N}+1\right)\overline{C}_{\tau,\sigma,\theta}\|z_0^t - z_*^t\|^2 + \overline{C}'_{\tau,\sigma,\theta}\right)\right],$$

where we use eq. (105) for the last inequality. Then, if we use the above inequality within eq. (106), it follows that

$$\mathbb{E}\left[\mathcal{G}^t(z_0^{t+1})\right] \leq \frac{1}{N}\overline{\overline{C}}_{\tau,\sigma,\theta}\mathbb{E}\left[\|z_0^t - z_*^t\|^2\right] + \frac{1}{N}\overline{\overline{C}}'_{\tau,\sigma,\theta} + \Xi_{\tau,\sigma,\theta},$$

where

$$\overline{\overline{C}}_{\tau,\sigma,\theta} \triangleq 4\max\left\{\frac{1}{\tau}, \frac{1+\theta}{\sigma}\right\}\max\{1, \kappa_{xy}^2, \kappa_{yx}^2\}\overline{C}_{\tau,\sigma,\theta},$$

$$\overline{\overline{C}}'_{\tau,\sigma,\theta} \triangleq 2\max\left\{\frac{1}{\tau}, \frac{1+\theta}{\sigma}\right\}\max\{1, \kappa_{xy}^2, \kappa_{yx}^2\}\overline{C}'_{\tau,\sigma,\theta}.$$

Thus, for $C_{\tau,\sigma,\theta} \triangleq \max\{\overline{C}_{\tau,\sigma,\theta}, \overline{\overline{C}}_{\tau,\sigma,\theta}\}$ and $C'_{\tau,\sigma,\theta} \triangleq \max\{\overline{C}'_{\tau,\sigma,\theta}, \frac{1}{N}\overline{\overline{C}}'_{\tau,\sigma,\theta} + \Xi_{\tau,\sigma,\theta}\}$, we get the desired result in (100). $\qquad\square$

**Lemma 22.** *Under the premise of Lemma 2, $\mathbb{E}[\|z_0^t - z_*^t\|^2]$, $\mathbb{E}\left[\mathcal{G}^t(z_0^t)\right]$ and $\mathbb{E}\left[\mathcal{G}^t(z_0^{t+1})\right]$ are finite for any $t \geq 0$ when either Assumption 4 or Assumption 5 holds.*

*Proof.* In Lemma 21, we show that

$$\max\left\{\mathbb{E}\left[\mathcal{G}^t(z_0^{t+1})\right],\ \mathbb{E}\left[\|z_0^{t+1} - z_*^t\|^2\right]\right\} \leq \frac{1}{N_t} C_{\tau,\sigma,\theta}\mathbb{E}[\|z_0^t - z_*^t\|^2] + C'_{\tau,\sigma,\theta}, \tag{107}$$

for some $C_{\tau,\sigma,\theta}, C'_{\tau,\sigma,\theta} \in \mathbb{R}_+$ constants, dependent on the SAPD parameters. Next, we show that $\{\mathbb{E}\left[\|z_0^t - z_*^t\|^2\right]\} < \infty$ for all $t \geq 0$ by induction. This is trivially true for $t = 0$, i.e., $\mathbb{E}\left[\|z_0^0 - z_*^0\|^2\right] = \|z_0^0 - z_*^0\|^2 < \infty$. Next, for some $t \geq 0$, suppose $\mathbb{E}\left[\|z_0^t - z_*^t\|^2\right] < \infty$, (107) implies that

$$\mathbb{E}\left[\|z_0^{t+1} - z_*^t\|^2\right] < \infty. \tag{108}$$

The inductive assumption $\mathbb{E}\left[\|z_0^t - z_*^t\|^2\right] < \infty$ and (108) imply that

$$\mathbb{E}\left[\|z_0^{t+1} - z_0^t\|^2\right] \leq 2\mathbb{E}\left[\|z_0^{t+1} - z_*^t\|^2\right] + 2\mathbb{E}\left[\|z_0^t - z_*^t\|^2\right] < \infty. \tag{109}$$

For any $\mu_x > 0$, fix $\lambda = (\mu_x + \gamma)^{-1}$; since we have $x_*^\ell = \mathbf{prox}_{\lambda\phi}(x_0^\ell)$ for $\ell = t, t+1$ and $\mathbf{prox}_{\lambda\phi}(\cdot)$ is non-expansive, we have $\mathbb{E}[\|x_*^{t+1} - x_*^t\|^2] \leq \mathbb{E}[\|x_0^{t+1} - x_0^t\|^2]$. Moreover, Lemma 20 implies that $\mathbb{E}[\|y_*^{t+1} - y_*^t\|^2] \leq \kappa_{yx}^2\mathbb{E}[\|x_*^{t+1} - x_*^t\|^2]$ for $\kappa_{yx} = \frac{L_{yx}}{\mu_y}$; thus, using (109), we get

$$\mathbb{E}[\|z_*^{t+1} - z_*^t\|^2] \leq (\kappa_{yx}^2 + 1)\mathbb{E}[\|x_0^{t+1} - x_0^t\|^2] \leq (\kappa_{yx}^2 + 1)\mathbb{E}[\|z_0^{t+1} - z_0^t\|^2] < \infty. \tag{110}$$

Therefore, we can conclude that $\mathbb{E}[\|z_0^{t+1} - z_*^{t+1}\|^2] \leq 2\mathbb{E}[\|z_0^{t+1} - z_*^t\|^2] + 2\mathbb{E}[\|z_*^{t+1} - z_*^t\|^2] < \infty$, which follows from (108) and (110). This completes induction, providing us with $\mathbb{E}[\|z_0^t - z_*^t\|^2] < \infty$ for all $t \geq 0$. Note that using this result together with the definition of $\mathcal{G}^t$ and (107) implies that $0 \leq \mathbb{E}[\mathcal{G}^t(z_0^{t+1})] < \infty$ for $t \geq 0$.

Next, we will argue that $\mathbb{E}[\mathcal{G}^t(z_0^t)] < \infty$ for all $t \geq 0$ as well. Recall that $\mathcal{G}^t(z_0^t) = \sup_{y\in\mathcal{Y}} \mathcal{L}^t(x_0^t, y) - \inf_{x\in\mathcal{X}} \mathcal{L}^t(x, y_0^t)$; furthermore, note that $\mathcal{L}^t(x_0^t, y) = \mathcal{L}(x_0^t, y)$ for all $y \in \mathcal{Y}$, and given $z_0^t$, we have $\mathcal{L}^t(\cdot, y_0^t)$ strongly convex with modulus $\mu_x$ and $\mathcal{L}(x_0^t, \cdot)$ strongly concave with modulus $\mu_y$. Therefore, we have

$$\mathcal{L}(x_0^t, y) \leq \mathcal{L}(x_0^t, y_0^t) + \left\langle \nabla_y\Phi(x_0^t, y_0^t) - s_g(y_0^t),\ y - y_0^t\right\rangle - \frac{\mu_y}{2}\|y - y_0^t\|^2$$

$$\leq \mathcal{L}(x_0^t, y_0^t) + \frac{1}{2\mu_y}\|\nabla_y\Phi(x_0^t, y_0^t) - s_g(y_0^t)\|^2, \tag{111}$$

$$\mathcal{L}^t(x, y_0^t) \geq \mathcal{L}(x_0^t, y_0^t) + \left\langle \nabla_x\Phi(x_0^t, y_0^t) + s_f(x_0^t),\ x - x_0^t\right\rangle + \frac{\mu_x}{2}\|x - x_0^t\|^2$$

$$\geq \mathcal{L}(x_0^t, y_0^t) - \frac{1}{2\mu_x}\|\nabla_x\Phi(x_0^t, y_0^t) + s_f(x_0^t)\|^2, \tag{112}$$

where $s_f(x_0^t) \in \partial f(x_0^t)$ and $s_g(y_0^t) \in \partial g(y_0^t)$ such that $\|s_f(x_0^t)\| \leq B_f$ and $\|s_g(y_0^t)\| \leq B_g$ –see Assumption 5; moreover, we have used the fact that $\mathcal{L}^t(x_0^t, y_0^t) = \mathcal{L}(x_0^t, y_0^t)$ and $\partial_x\mathcal{L}^t(x_0^t, y_0^t) = \nabla_x\Phi(x_0^t, y_0^t) + \partial f(x_0^t)$. Thus, (111) and (112) imply that

$$\mathcal{G}^t(z_0^t) = \sup_{x\in\mathcal{X}, y\in\mathcal{Y}}\{\mathcal{L}(x_0^t, y) - \mathcal{L}^t(x, y_0^t)\} \leq \left(\|\nabla\Phi(z_0^t)\|^2 + \|s_f(x_0^t)\|^2 + \|s_g(y_0^t)\|^2\right)/\min\{\mu_x, \mu_y\}$$

$$\leq \frac{1}{\mu}\left(\|\nabla\Phi(z_0^t) - \nabla\Phi(z_0^0)\|^2 + \|\nabla\Phi(z_0^0)\|^2 + B_f^2 + B_g^2\right)$$

$$\leq \frac{L}{\mu}\|z_0^t - z_0^0\|^2 + \frac{1}{\mu}\left(\|\nabla\Phi(z_0^0)\|^2 + B_f^2 + B_g^2\right),$$

where $L = \max\{L_{xx}, L_{yy}, L_{yx}, L_{xy}\}$ and $\mu = \min\{\mu_x, \mu_y\}$. Finally, (109) implies that $\mathbb{E}[\|z_0^t - z_0^0\|^2] < \infty$; therefore, we can conclude that $\mathbb{E}[\mathcal{G}^t(z_0^t)] < \infty$ for all $t \geq 0$. $\qquad\square$

Thus, Lemma 22 implies that the analysis given in appendix A.1 directly goes through if we replace Assumption 4 with Assumption 5, which does not require compactness of the problem domain.

# F   Proof of Theorem 4 and preliminary technical results

The general proof structure of Theorem 4 is the same with Theorem 1's. The main difference is the way we bound the variance, which is given in Lemma 24.

## F.1   Construction for the iteration complexity result

**Lemma 23.** *Suppose Assumptions 1, 3, 6 and 7 hold. Given $\{N_t\}_{t\geq 0} \subset \mathbb{Z}_+$, let $\{x_0^t, y_0^t\}_{t\geq 0}$ be generated by SAPD+, stated in Algorithm 2, when VR-flag=**true**, initialized from $(x_0^0, y_0^0) \in$* **dom** $f \times$ **dom** $g$ *and using $\tau, \sigma, \theta, \mu_x > 0$ that satisfy*

$$G - \operatorname{diag}(g) \succeq 0, \tag{113}$$

*for some $\alpha \in [0, \frac{1}{\sigma})$, $\rho \in (0, 1]$ and $\pi_x, \pi_y > 0$, where $G$ is defined in (32), $g \triangleq [\pi_x, \pi_y, L'_x, L'_y, 0]^\top$ and*

$$L'_x \triangleq c(\rho)\, \big(\frac{{L'_{xx}}^2}{\pi_x b'_x} + \frac{2(1 + 2\theta + 2\theta^2)\rho^{-1}L_{yx}^2}{\pi_y b'_y}\big), \quad L'_y \triangleq c(\rho)\, \big(\frac{\rho L_{xy}^2}{\pi_x b'_x} + \frac{2(1 + 2\theta + 2\theta^2)\rho^{-1}L_{yy}^2}{\pi_y b'_y}\big),$$

*such that $c(\rho) = \frac{2}{1-\rho}(\rho^{-q+1} - 1)$ for $\rho \in (0, 1)$ and $c(\rho) = 2(q - 1)$ for $\rho = 1$, where $L'_{xx} \triangleq L_{xx} + \mu_x + \gamma$. Then for all $t \geq 0$, it holds that*

$$\mathbb{E}\left[\mathcal{G}^t(x_0^{t+1}, y_0^{t+1})\right] \leq \frac{M^{VR}}{K_{N_t}(\rho)}\left(\frac{\mu_x}{4}\mathbb{E}\left[\|x_*^t(y_0^{t+1}) - x_0^t\|^2\right] + \frac{\mu_y}{4}\mathbb{E}\left[\|y_*(x_0^{t+1}) - y_0^t\|^2\right]\right) + \Xi^{VR}, \tag{114}$$

*where $K_{N_t}(\rho) = \sum_{k=0}^{N_t - 1}\rho^{-k}$, $\Xi^{VR} \triangleq \frac{\delta_x^2}{2\pi_x b} + (1 + 2\theta + 2\theta^2)\frac{\delta_y^2}{\pi_y b}$ and $M^{VR} \triangleq \max\{\frac{2}{\mu_x}(\frac{1}{\tau} - \mu_x), \frac{2}{\mu_y \sigma}\}$.*

*Proof.* For easier readability, we provide the proof in a separate subsection, see appendix F.2.   □

**Theorem 8.** *Under the premise of Lemma 23, given an arbitrary $\zeta > 0$ and $T \in \mathbb{Z}_+$, suppose $N_t = N$ for all $t = 0, \ldots T$ for some $N \in \mathbb{Z}_+$ such that $N \geq (1 + \zeta)M^{VR}$, and (21) has a solution for some $\beta_1, \beta_2 \in (0, 1)$ and $p_1, p_2, p_3 > 0$ such that $p_1 + p_2 + p_3 = 1$. If either Assumption 4 or Assumption 5 holds, then (22) holds with $\Xi_{\tau,\sigma,\theta} = \Xi^{VR}$ for $\lambda = (\gamma + \mu_x)^{-1}$ and for all $T \geq 1$.*

*Proof.* The proof is omitted as it is essentially the same with the proof of Theorem 6.   □

## F.2   Proof of Lemma 23 and preliminary technical results

In this section we prove Lemma 23. We first state a technical lemma that will be used in our analysis.

**Lemma 24.** *Suppose Assumptions 1, 3, 6 and 7 hold. Let $\{x_k^t, y_k^t\}_{k\geq 0}$ be VR-SAPD iterates generated according to Algorithm 3 for solving $\min_x \max_y \mathcal{L}^t(x, y)$. Then,*

$$\mathbb{E}\left[\|v_k^t - \nabla_x \Phi^t(x_k^t, y_{k+1}^t)\|^2\right] \leq \frac{\delta_x^2}{b} + \sum_{i=(n_k-1)q+1}^{k}\frac{2{L'_{xx}}^2}{b'_x}\mathbb{E}\left[\|x_i^t - x_{i-1}^t\|^2\right] + \frac{2L_{xy}^2}{b'_x}\mathbb{E}\left[\|y_{i+1}^t - y_i^t\|^2\right], \tag{115a}$$

$$\mathbb{E}\left[\|w_k^t - \nabla_y \Phi^t(x_k^t, y_k^t)\|^2\right] \leq \frac{\delta_y^2}{b} + \sum_{i=(n_k-1)q+1}^{k}\frac{2L_{yx}^2}{b'_y}\mathbb{E}\left[\|x_i^t - x_{i-1}^t\|^2\right] + \frac{2L_{yy}^2}{b'_y}\mathbb{E}\left[\|y_i^t - y_{i-1}^t\|^2\right], \tag{115b}$$

*for all $k \geq 0$ such that $\operatorname{mod}(k, q) \neq 0$, where $n_k \triangleq \lceil k/q \rceil$. Moreover, if $\operatorname{mod}(k, q) = 0$, then*

$$\mathbb{E}\left[\|v_k^t - \nabla_x \Phi^t(x_k^t, y_{k+1}^t)\|^2\right] \leq \frac{\delta_x^2}{b}, \quad \mathbb{E}\left[\|w_k^t - \nabla_y \Phi^t(x_k^t, y_k^t)\|^2\right] \leq \frac{\delta_y^2}{b}. \tag{116}$$

*Proof.* Recall that $\tilde{\nabla}_x \Phi^t_{I_k^x}(x_k^t, y_{k+1}^t) \triangleq \frac{1}{|I_k^x|}\sum_{\omega_k^{x,i} \in I_k^x}\tilde{\nabla}_x \Phi^t(x_k^t, y_{k+1}^t; \omega^{x,i})$, where $I_k^x = \{\omega_k^{x,i}\}_{i=1}^{b'_x}$ is a randomly generated batch with $|I_k^x| = b'_x$ independent elements which are also independent of $(x_{k-1}^t, y_k^t)$ and $(x_k^t, y_{k+1}^t)$. According to the definition of $v_k$ in Algorithm 3, for $\operatorname{mod}(k, q) > 0$,

$$v_k^t = v_{k-1}^t + \tilde{\nabla}_x \Phi^t_{I_k^x}(x_k^t, y_{k+1}^t) - \tilde{\nabla}_x \Phi^t_{I_k^x}(x_{k-1}^t, y_k^t). \tag{117}$$

Therefore,

$$\mathbb{E}\left[\|v_k^t - \nabla_x \Phi^t(x_k^t, y_{k+1}^t)\|^2\right]$$

$$= \mathbb{E}\left[\|v_{k-1}^t + \tilde{\nabla}_x \Phi_{I_k^x}^t(x_k^t, y_{k+1}^t) - \tilde{\nabla}_x \Phi_{I_k^x}^t(x_{k-1}^t, y_k^t) - \nabla_x \Phi^t(x_k^t, y_{k+1}^t)\|^2\right]$$

$$= \mathbb{E}\left[\|v_{k-1}^t - \nabla_x \Phi^t(x_{k-1}^t, y_k^t) + \nabla_x \Phi^t(x_{k-1}^t, y_k^t) - \tilde{\nabla}_x \Phi_{I_k^x}^t(x_{k-1}^t, y_k^t) + \tilde{\nabla}_x \Phi_{I_k^x}^t(x_k^t, y_{k+1}^t) - \nabla_x \Phi^t(x_k^t, y_{k+1}^t)\|^2\right]$$

$$= \mathbb{E}\left[\|v_{k-1}^t - \nabla_x \Phi^t(x_{k-1}^t, y_k^t)\|^2\right]$$
$$+ \mathbb{E}\left[\|\nabla_x \Phi^t(x_{k-1}^t, y_k^t) - \tilde{\nabla}_x \Phi_{I_k^x}^t(x_{k-1}^t, y_k^t) + \tilde{\nabla}_x \Phi_{I_k^x}^t(x_k^t, y_{k+1}^t) - \nabla_x \Phi^t(x_k^t, y_{k+1}^t)\|^2\right],$$

(118)

where for the last equality we used

$$\mathbb{E}\left[\nabla_x \Phi^t(x_{k-1}^t, y_k^t) - \tilde{\nabla}_x \Phi_{I_k^x}^t(x_{k-1}^t, y_k^t) + \tilde{\nabla}_x \Phi_{I_k^x}^t(x_k^t, y_{k+1}^t) - \nabla_x \Phi^t(x_k^t, y_{k+1}^t)\right] = 0.$$

Next, we bound the second expectation on the rhs of (118). It follows that

$$\mathbb{E}\left[\|\nabla_x \Phi^t(x_{k-1}^t, y_k^t) - \tilde{\nabla}_x \Phi_{I_k^x}^t(x_{k-1}^t, y_k^t) + \tilde{\nabla}_x \Phi_{I_k^x}^t(x_k^t, y_{k+1}^t) - \nabla_x \Phi^t(x_k^t, y_{k+1}^t)\|^2\right]$$

$$= \frac{1}{b_x'^2} \mathbb{E}\left[\|\sum_{i=1}^{b_x'} \left(\tilde{\nabla}_x \Phi^t(x_k^t, y_{k+1}^t; \omega_k^{x,i}) - \tilde{\nabla}_x \Phi^t(x_{k-1}^t, y_k^t; \omega_k^{x,i}) - \nabla_x \Phi^t(x_k^t, y_{k+1}^t) + \nabla_x \Phi^t(x_{k-1}^t, y_k^t)\right)\|^2\right]$$

$$= \frac{1}{b_x'^2} \sum_{i=1}^{b_x'} \mathbb{E}\left[\|\tilde{\nabla}_x \Phi^t(x_k^t, y_{k+1}^t; \omega_k^{x,i}) - \tilde{\nabla}_x \Phi^t(x_{k-1}^t, y_k^t; \omega_k^{x,i}) - \nabla_x \Phi^t(x_k^t, y_{k+1}^t) + \nabla_x \Phi^t(x_{k-1}^t, y_k^t)\|^2\right]$$

$$\leq \frac{1}{b_x'^2} \sum_{i=1}^{b_x'} \left(\mathbb{E}\left[\|\tilde{\nabla}_x \Phi^t(x_k^t, y_{k+1}^t; \omega_k^{x,i}) - \tilde{\nabla}_x \Phi^t(x_{k-1}^t, y_k^t; \omega_k^{x,i})\|^2\right]\right)$$

$$\leq \frac{2L_{xx}'^2}{b_x'} \mathbb{E}\left[\|x_k^t - x_{k-1}^t\|^2\right] + \frac{2L_{xy}^2}{b_x'} \mathbb{E}\left[\|y_{k+1}^t - y_k^t\|^2\right],$$

(119)

where the second equality follows from the stochastic oracle being unbiased –see Assumption 3, which implies

$$\mathbb{E}\left[\nabla_x \Phi^t(x_{k-1}^t, y_k^t) - \tilde{\nabla}_x \Phi^t(x_{k-1}^t, y_k^t; \omega_k^{x,i}) + \tilde{\nabla}_x \Phi^t(x_k^t, y_{k+1}^t; \omega_k^{x,i}) - \nabla_x \Phi^t(x_k^t, y_{k+1}^t)\right] = 0,$$

for all $i = 1, \ldots, b_x'$ and $\{\omega_i^k\}_{i=1}^{b_x'}$ being independent; the first inequality is because $\mathbb{E}\left[\|\zeta - \mathbb{E}[\zeta]\|^2\right] \leq \mathbb{E}\left[\|\zeta\|^2\right]$ for any given random variable $\zeta$ with finite second order moment –we invoke this inequality for $\zeta = \tilde{\nabla}_x \Phi^t(x_k^t, y_{k+1}^t; \omega_k^{x,i}) - \tilde{\nabla}_x \Phi^t(x_{k-1}^t, y_k^t; \omega_k^{x,i})$; and finally, the last inequality follows from Assumption 6 and the inequality $(a+b)^2 \leq 2a^2 + 2b^2$ for any $a, b \in \mathbb{R}$. Next, if we combine eq. (118) and eq. (119), we get

$$\mathbb{E}\left[\|v_k^t - \nabla_x \Phi^t(x_k^t, y_{k+1}^t)\|^2\right]$$
$$\leq \mathbb{E}\left[\|v_{k-1}^t - \nabla_x \Phi^t(x_{k-1}^t, y_k^t)\|^2\right] + \frac{2L_{xx}'^2}{b_x'} \mathbb{E}\left[\|x_k^t - x_{k-1}^t\|^2\right] + \frac{2L_{xy}^2}{b_x'} \mathbb{E}\left[\|y_{k+1}^t - y_k^t\|^2\right].$$

Hence, if we sum the above inequality from $(n_k - 1)q + 1$ to $k$, we get a telescoping sum:

$$\mathbb{E}\left[\|v_k^t - \nabla_x \Phi^t(x_k^t, y_{k+1}^t)\|^2\right]$$

$$\leq \sum_{i=(n_k-1)q+1}^{k} \frac{2L_{xx}'^2}{b_x'} \mathbb{E}\left[\|x_i^t - x_{i-1}^t\|^2\right] + \sum_{i=(n_k-1)q+1}^{k} \frac{2L_{xy}^2}{b_x'} \mathbb{E}\left[\|y_{i+1}^t - y_i^t\|^2\right]$$

$$+ \mathbb{E}\left[\|v_{(n_k-1)q} - \nabla_x \Phi^t(x_{(n_k-1)q}^t, y_{(n_k-1)q+1}^t)\|^2\right]$$

(120)

$$\leq \sum_{i=(n_k-1)q+1}^{k} \frac{2L_{xx}'^2}{b_x'} \mathbb{E}\left[\|x_i^t - x_{i-1}^t\|^2\right] + \sum_{i=(n_k-1)q+1}^{k} \frac{2L_{xy}^2}{b_x'} \mathbb{E}\left[\|y_{i+1}^t - y_i^t\|^2\right] + \frac{\delta_x^2}{b},$$

where the last inequality follows from Assumption 3 since $\text{mod}((n_k - 1)q, q) = 0$ and for $\ell \in \mathbb{Z}_+$ such that $\text{mod}(\ell, q) = 0$, we have $v_\ell = \tilde{\nabla}_x \Phi_{\mathcal{B}_\ell^x}^t(x_\ell^t, y_{\ell+1}^t) = \frac{1}{|\mathcal{B}_\ell^x|} \sum_{\omega_\ell^{x,i} \in \mathcal{B}_\ell^x} \tilde{\nabla}_x \Phi^t(x_\ell^t, y_{\ell+1}^t; \omega_\ell^{x,i})$,

where $\mathcal{B}_\ell^x = \{\omega_\ell^{x,i}\}$ is a randomly generated batch with $|\mathcal{B}_\ell^x| = b$ independent elements which are also independent of $(x_\ell^t, y_{\ell+1}^t)$. This completes the proof of the case for $k$ such that $\mathrm{mod}(k,q) > 0$.

When $\mathrm{mod}(k,q) = 0$, it follows from Algorithm 3 that $v_k = \tilde{\nabla}_x \Phi_{\mathcal{B}_k}^t(x_k^t, y_{k+1}^t)$. Hence, above discussion yields

$$\mathbb{E}\left[\|v_k^t - \nabla_x \Phi^t(x_k^t, y_{k+1}^t)\|^2\right] \le \frac{\delta_x^2}{b}. \tag{121}$$

Finally, the second inequality in (115b) can be shown similarly. $\qquad\square$

Next, we will modify Lemma 13 for VR-SAPD, stated in Algorithm 3. Specifically, instead of using the stochastic oracles $\tilde{\nabla}_x \Phi^t(x_k^t, y_{k+1}^t; \omega_k^x)$ and $\tilde{\nabla}_y \Phi^t(x_k^t, y_k^t; \omega_k^y)$ as in Lemma 13, we adopt $v_k^t$ and $w_k^t$ to estimate $\nabla_x \Phi^t(x_k^t, y_{k+1}^t)$ and $\nabla_y \Phi^t(x_k^t, y_k^t)$, respectively.

**Lemma 25.** *Suppose Assumptions 1, 3, and 6 hold. Let $\{x_k^t, y_k^t\}_{k \ge 0}$ be VR-SAPD iterates generated according to Algorithm 3 for solving $\min_x \max_y \mathcal{L}^t(x,y)$. Then for all $x \in \mathbf{dom}\, f \subset \mathcal{X}$, $y \in \mathbf{dom}\, g \subset \mathcal{Y}$, and $k \ge 0$,*

$$\mathcal{L}^t(x_{k+1}^t, y) - \mathcal{L}^t(x, y_{k+1}^t) \tag{122}$$
$$\le -\langle q_{k+1}^t, y_{k+1}^t - y\rangle + \theta\langle q_k^t, y_k^t - y\rangle + \Lambda_k^t(x,y) - \Sigma_{k+1}^t(x,y) + \Gamma_{k+1}^t + \varepsilon_k^{t,x}(x) + \varepsilon_k^{t,y}(y),$$

*where $\varepsilon_k^{t,x}(x) \triangleq \langle v_k^t - \nabla_x \Phi^t(x_k^t, y_{k+1}^t),\ x - x_{k+1}^t\rangle$ and $\varepsilon_k^{t,y}(y) \triangleq \langle \tilde{s}_k^t - s_k^t, y_{k+1}^t - y\rangle$ for $\tilde{s}_k^t = (1+\theta)w_k^t - \theta w_{k-1}^t$ as defined in Algorithm 3, $q_k^t$ and $s_k^t$ are defined as in (59), and $\Lambda_k^t(x,y)$, $\Sigma_{k+1}^t(x,y)$, $\Gamma_{k+1}^t$ are the same with those in Lemma 13.*

*Proof.* The proof uses the same arguments as the proof of Lemma 13. One only needs to replace $\tilde{\nabla}_x \Phi^t(x_k^t, y_{k+1}^t)$ and $\tilde{\nabla}_y \Phi^t(x_k^t, y_k^t)$ in the proof of Lemma 13 with $v_k^t, w_k^t$, respectively. $\qquad\square$

### F.3 Proof of Lemma 23

*Proof.* For simplifying the notation, let $N_t = N$ for some $N \in \mathbb{Z}_+$. For arbitrary $(x,y) \in \mathbf{dom}\, f \times \mathbf{dom}\, g$, since $(x_{k+1}^t, y_{k+1}^t) \in \mathbf{dom}\, f \times \mathbf{dom}\, g$, using the concavity of $\mathcal{L}^t(x_{k+1}^t, \cdot)$ and the convexity of $\mathcal{L}^t(\cdot, y_{k+1}^t)$, Lemma 25 and Jensen's lemma immediately implies that

$$K_N(\rho)\left(\mathcal{L}^t(\bar{x}_N^t, y) - \mathcal{L}^t(x, \bar{y}_N^t)\right) \le \sum_{k=0}^{N-1} \rho^{-k}\left(\mathcal{L}^t(x_{k+1}^t, y) - \mathcal{L}^t(x, y_{k+1}^t)\right),\ \forall \rho \in (0,1], \tag{123}$$

where $\bar{x}_N^t = \frac{1}{K_N(\rho)}\sum_{k=0}^{N-1}\rho^{-k}x_{k+1}^t$, $\bar{y}_N^t = \frac{1}{K_N(\rho)}\sum_{k=0}^{N-1}\rho^{-k}y_{k+1}^t$, and $K_N(\rho) = \sum_{i=0}^{N-1}\rho^{-k}$. Thus, if we multiply both sides of (122) by $\rho^{-k}$ and sum the resulting inequality from $k = 0$ to $N-1$, then using (123) we get

$$K_N(\rho)\left(\mathcal{L}^t(\bar{x}_N^t, x) - \mathcal{L}(x, \bar{y}_N^t)\right)$$

$$\le \sum_{k=0}^{N-1} \rho^{-k}\Big( -\langle q_{k+1}^t, y_{k+1}^t - x\rangle + \theta\langle q_k^t, y_k^t - x\rangle + \Lambda_k^t(x,y) - \Sigma_{k+1}^t(x,y) + \Gamma_{k+1}^t$$
$$\qquad - \langle v_k^t - \nabla_x \Phi^t(x_k^t, y_{k+1}^t), x_{k+1}^t - x\rangle + \langle \tilde{s}_k^t - s_k^t, y_{k+1}^t - x\rangle\Big)$$

$$\le \sum_{k=0}^{N-1} \rho^{-k}\Big( \underbrace{-\langle q_{k+1}^t, y_{k+1}^t - x\rangle + \theta\langle q_k^t, y_k^t - x\rangle}_{\textbf{part 1}} + \Lambda_k^t(x,y) - \Sigma_{k+1}^t(x,y) + \Gamma_{k+1}^t$$
$$\qquad + \frac{\pi_x}{2}\|x_{k+1}^t - x\|^2 + \frac{\pi_y}{2}\|y_{k+1}^t - x\|^2 + \underbrace{\frac{1}{2\pi_x}\|v_k^t - \nabla_x \Phi^t(x_k^t, y_{k+1}^t)\|^2 + \frac{1}{2\pi_y}\|\tilde{s}_k^t - s_k^t\|^2}_{\textbf{part 2}}\Big).$$
$$\tag{124}$$

The second inequality follows from Young's inequality for some constants $\pi_x, \pi_y > 0$.

The following bound for **part1** can be obtained from (74). Indeed, for any $k \ge -1$, we get

$$\sum_{k=0}^{N-1} \rho^{-k}\left(\theta\langle q_k^t, y_k^t - y_*^t\rangle - \langle q_{k+1}^t, y_{k+1}^t - y_*^t\rangle\right) \le \sum_{k=0}^{N-1} \rho^{-k}|1 - \frac{\theta}{\rho}|\, S_{k+1}^t(x,y) + \rho^{-N+1}\frac{\theta}{\rho}S_N^t(x,y). \tag{125}$$

where $S_{k+1}^t(x, y)$ is defined in (73).

Next we consider **part 2**, recall that $n_k = \lceil k/q \rceil$ such that $(n_k - 1)q + 1 \leq k \leq n_k q$, it follows from Lemma 24 that

$$\sum_{k=0}^{N-1} \rho^{-k} \mathbb{E}\left[\|v_k^t - \nabla_x \Phi^t(x_k^t, y_{k+1}^t)\|^2\right]$$

$$\leq \sum_{\substack{k \in \{1, \ldots, N-1\} \\ \text{s.t. } \mathrm{mod}(k,q) \neq 0}} \rho^{-k} \sum_{i=(n_k-1)q+1}^{k} \left(\frac{2L_{xx}'^2}{b_x'} \mathbb{E}\left[\|x_i^t - x_{i-1}^t\|^2\right] + \frac{2L_{xy}^2}{b_x'} \mathbb{E}\left[\|y_{i+1}^t - y_i^t\|^2\right]\right) + \frac{\delta_x^2}{b} \sum_{k=0}^{N-1} \rho^{-k}$$

$$= \sum_{k=1}^{N-1} \rho^{-k} \left(\frac{2L_{xx}'^2}{b_x'} \mathbb{E}\left[\|x_k^t - x_{k-1}^t\|^2\right] + \frac{2L_{xy}^2}{b_x'} \mathbb{E}\left[\|y_{k+1}^t - y_k^t\|^2\right]\right) \sum_{i=0}^{n_k q - k - 1} \rho^{-i} + \frac{\delta_x^2}{b} \sum_{k=0}^{N-1} \rho^{-k}$$

$$= \sum_{k=0}^{N-1} \rho^{-k} \cdot \frac{\rho}{1-\rho} (\rho^{-n_k q + k} - 1) \left(\frac{2L_{xx}'^2}{b_x'} \mathbb{E}\left[\|x_k^t - x_{k-1}^t\|^2\right] + \frac{2L_{xy}^2}{b_x'} \mathbb{E}\left[\|y_{k+1}^t - y_k^t\|^2\right]\right) + \frac{\delta_x^2}{b} \sum_{k=0}^{N-1} \rho^{-k}$$

$$\leq \sum_{k=0}^{N-1} \rho^{-k} \cdot \frac{\rho}{1-\rho} (\rho^{-q+1} - 1) \left(\frac{2L_{xx}'^2}{b_x'} \mathbb{E}\left[\|x_k^t - x_{k-1}^t\|^2\right] + \frac{2L_{xy}^2}{b_x'} \mathbb{E}\left[\|y_{k+1}^t - y_k^t\|^2\right]\right) + \frac{\delta_x^2}{b} \sum_{k=0}^{N-1} \rho^{-k}$$

$$\tag{126}$$

where the first inequality follows from Lemma 24, the following equality is by rearranging terms, and for the last inequality we used the following bound: $n_k = \lceil k/q \rceil \leq k/q + (q-1)/q$; hence, $-n_k q + k \geq -q + 1$. To bound **part 2** in eq. (124), we next consider $\|\tilde{s}_k^t - s_k^t\|^2$. For $k > 0$,

$$\|\tilde{s}_k^t - s_k^t\|^2 = \|(1+\theta)w_k^t - (1+\theta)\nabla_y \Phi^t(x_k^t, y_k^t) - \theta w_{k-1}^t + \theta \nabla_y \Phi^t(x_{k-1}^t, y_{k-1}^t)\|^2$$
$$\leq 2(1+\theta)^2 \|w_k^t - \nabla_y \Phi^t(x_k^t, y_k^t)\|^2 + 2\theta^2 \|w_{k-1}^t - \nabla_y \Phi^t(x_{k-1}^t, y_{k-1}^t)\|^2. \tag{127}$$

First, $x_{-1}^t = x_0^t$, $y_{-1}^t = y_0^t$ and (59) imply that $s_0^t = \nabla_y \Phi^t(x_0^t, y_0^t)$, and recall that in Algorithm 3, we set $\tilde{s}_0^t = w_0^t$; hence,

$$\|\tilde{s}_0^t - s_0^t\|^2 = \|w_0^t - \nabla_y \Phi^t(x_0^t, y_0^t)\|^2,$$

and eq. (127) holds for $k \geq 0$ with $w_{-1}^t \triangleq \nabla_y \Phi^t(x_0^t, y_0^t)$. Then, Lemma 24 implies that

$$\sum_{k=0}^{N-1} \rho^{-k} \mathbb{E}\left[\|\tilde{s}_k^t - s_k^t\|^2\right]$$

$$\leq 2(1+\theta)^2 \sum_{\substack{k \in \{1, \ldots, N-1\} \\ \text{s.t. } \mathrm{mod}(k,q) \neq 0}} \rho^{-k} \sum_{i=(n_k-1)q+1}^{k} \left(\frac{2L_{yx}^2}{b_y'} \mathbb{E}\left[\|x_i^t - x_{i-1}^t\|^2\right] + \frac{2L_{yy}^2}{b_y'} \mathbb{E}\left[\|y_i^t - y_{i-1}^t\|^2\right]\right)$$

$$+ \frac{2\theta^2}{\rho} \sum_{\substack{k \in \{1, \ldots, N-2\} \\ \text{s.t. } \mathrm{mod}(k,q) \neq 0}} \rho^{-k} \sum_{i=(n_k-1)q+1}^{k} \left(\frac{2L_{yx}^2}{b_y'} \mathbb{E}\left[\|x_i^t - x_{i-1}^t\|^2\right] + \frac{2L_{yy}^2}{b_y'} \mathbb{E}\left[\|y_i^t - y_{i-1}^t\|^2\right]\right)$$

$$+ 2(1 + 2\theta + 2\theta^2) \frac{\delta_y^2}{b} \sum_{k=0}^{N-1} \rho^{-k}$$

$$\leq \frac{2}{\rho}(1 + 2\theta + 2\theta^2) \sum_{\substack{k \in \{1, \ldots, N-1\} \\ \text{s.t. } \mathrm{mod}(k,q) \neq 0}} \rho^{-k} \sum_{i=(n_k-1)q+1}^{k} \left(\frac{2L_{yx}^2}{b_y'} \mathbb{E}\left[\|x_i^t - x_{i-1}^t\|^2\right] + \frac{2L_{yy}^2}{b_y'} \mathbb{E}\left[\|y_i^t - y_{i-1}^t\|^2\right]\right)$$

$$+ 2(1 + 2\theta + 2\theta^2) \frac{\delta_y^2}{b} \sum_{k=0}^{N-1} \rho^{-k},$$

$$\tag{128}$$

where the first inequality follows from Lemma 24; in the last inequality we used $\rho \le 1$ and combined the two sums. Next, as in eq. (126), we can further obtain that

$$\sum_{k=0}^{N-1} \rho^{-k} \mathbb{E}\left[\|\tilde{s}_k^t - s_k^t\|^2\right]$$

$$\le 2(1 + 2\theta + 2\theta^2) \sum_{k=0}^{N-1} \rho^{-k} \cdot \frac{1}{1-\rho}(\rho^{-q+1} - 1)\left(\frac{2L_{yx}^2}{b_y'} \mathbb{E}\left[\|x_k^t - x_{k-1}^t\|^2\right] + \frac{2L_{yy}^2}{b_y'} \mathbb{E}\left[\|y_k^t - y_{k-1}^t\|^2\right]\right)$$

$$+ 2(1 + 2\theta + 2\theta^2)\frac{\delta_y^2}{b} \sum_{k=0}^{N-1} \rho^{-k}.$$

$$(129)$$

Now we can bound **part 2** in eq. (124) using eq. (126) and eq. (129). In addition, Given $(\bar{x}_N^t, \bar{y}_N^t)$, the point $(x_*^t(\bar{y}_N^t), y_*(\bar{x}_N^t)) \triangleq \arg\max_{(x,y)\in\mathcal{X}\times\mathcal{Y}} \mathcal{L}^t(\bar{x}_N^t, y) - \mathcal{L}^t(x, \bar{y}_N^t)$ uniquely exists. We will use the fact that

$$\mathcal{G}^t(\bar{x}_N^t, \bar{y}_N^t) = \sup_{(x,y)\in\mathcal{X}\times\mathcal{Y}} \mathcal{L}^t(\bar{x}_N^t, y) - \mathcal{L}^t(x, \bar{y}_N^t) = \mathcal{L}^t(\bar{x}_N^t, y_*(\bar{x}_N^t)) - \mathcal{L}^t(x_*^t(\bar{y}_N^t), \bar{y}_N^t)$$

to complete the proof.

Recall that we defined $D_N^t(x,y) = \frac{1}{2\rho}(\frac{1}{\tau} - \mu_x)\|x_N^t - x\|^2 + \frac{1}{2}\left(\frac{1}{\rho\sigma} - \alpha\right)\|y_N^t - y\|^2$; first, we substitute $(x,y) = (x_*^t(\bar{y}_N^t), y_*(\bar{x}_N^t))$ into (124), and then add $\rho^{-N+1}D_N^t(x_*^t(\bar{y}_N^t), y_*(\bar{x}_N^t))$ to both sides of (124). Finally, taking the expectation of the new inequality, and then using eq. (125), eq. (126) and eq. (129) to bound **part 1** and **part 2**, we obtain

$$\mathbb{E}\left[K_N(\rho)\mathcal{G}^t(\bar{x}_N^t, \bar{y}_N^t) + \rho^{-N+1}D_N^t(x_*^t(\bar{y}_N^t), y_*(\bar{x}_N^t))\right]$$

$$\le \mathbb{E}\left[\hat{U}_N^t(x_*^t(\bar{y}_N^t), y_*(\bar{x}_N^t))\right] + \left(\frac{\delta_x^2}{2\pi_x b} + (1 + 2\theta + 2\theta^2)\frac{\delta_y^2}{\pi_y b}\right)\sum_{k=0}^{N-1} \rho^{-k}, \qquad (130)$$

where $\hat{U}_N^t(x,y)$ is defined as

$$\hat{U}_N^t(x,y) \triangleq \sum_{k=0}^{N-1} \rho^{-k}\left(\Gamma_{k+1}^t + \Lambda_k^t(x,y) - \Sigma_{k+1}^t(x,y)\right.$$

$$+ |1 - \frac{\theta}{\rho}| S_{k+1}^t(x,y) + \frac{\pi_x}{2}\|x_{k+1}^t - x\|^2 + \frac{\pi_y}{2}\|y_{k+1}^t - y\|^2\bigg)$$

$$+ \sum_{k=0}^{N-1} \frac{\rho^{-k+1}}{1-\rho}(\rho^{-q+1} - 1)\left(\left(\frac{L_{xx}'^2}{\pi_x b_x'} + \frac{2(1+2\theta+2\theta^2)\rho^{-1}L_{yx}^2}{\pi_y b_y'}\right)\|x_k^t - x_{k-1}^t\|^2 + \frac{L_{xy}^2}{\pi_x b_x'}\|y_{k+1}^t - y_k^t\|^2\right)$$

$$+ \sum_{k=0}^{N-1} \frac{\rho^{-k+1}}{1-\rho}(\rho^{-q+1} - 1)\frac{2(1+2\theta+2\theta^2)\rho^{-1}L_{yy}^2}{\pi_y b_y'}\|y_k^t - y_{k-1}^t\|^2 - \rho^{-N+1}(-D_N^t(x,y) - \frac{\theta}{\rho}S_N^t(x,y)).$$

$$(131)$$

The remaining part of the analysis directly follows from the arguments we used in the proof of Lemma 12. We can analyze $\hat{U}_N^t(x_*^t(\bar{y}_N^t), y_*(\bar{x}_N^t))$ through writing it as a telescoping sum. After adding and subtracting $\frac{\alpha}{2}\|y_{k+1}^t - y_k^t\|^2$, and rearranging the terms, we get

$$\hat{U}_N^t(x_*^t(\bar{y}_N^t), y_*(\bar{x}_N^t)) = \frac{1}{2}\sum_{k=0}^{N-1} \rho^{-k}\left(\xi_k^{*\top}\hat{A}\xi_k^* - \xi_{k+1}^{*\top}\hat{B}\xi_{k+1}^*\right)$$

$$- \rho^{-N+1}(-D_N^t(x_*^t(\bar{y}_N^t), y_*(\bar{x}_N^t)) - \frac{\theta}{\rho}S_N^t(x_*^t(\bar{y}_N^t), y_*(\bar{x}_N^t)))$$

$$= \frac{1}{2}\xi_0^{*\top}\hat{A}\xi_0^* - \frac{1}{2}\sum_{k=1}^{N-1} \rho^{-k+1}[\xi_k^{*\top}(\hat{B} - \frac{1}{\rho}\hat{A})\xi_k^*]$$

$$- \rho^{-N+1}(\frac{1}{2}\xi_N^{*\top}\hat{B}\xi_N^* - D_N^t(x_*^t(\bar{y}_N^t), y_*(\bar{x}_N^t)) - \frac{\theta}{\rho}S_N^t(x_*^t(\bar{y}_N^t), y_*(\bar{x}_N^t))),$$

$$(132)$$

where $\xi_k^* \in \mathbb{R}^5$ is defined for $k \geq 0$ as follows: $\xi_k^* \triangleq \begin{pmatrix} \|x_k^t - x_*^t(\bar{y}_N^t)\| \\ \|y_k^t - y_*(\bar{y}_N^t)\| \\ \|x_k^t - x_{k-1}^t\| \\ \|y_k^t - y_{k-1}^t\| \\ \|y_{k+1}^t - y_k^t\| \end{pmatrix}$ such that $x_{-1}^t = x_0^t$,

$y_{-1}^t = y_0^t$; and $\hat{A}, \hat{B} \in \mathbb{R}^{5 \times 5}$ are defined as:

$$\hat{A} \triangleq \begin{pmatrix} \frac{1}{\tau} - \mu_x & 0 & 0 & 0 & 0 \\ 0 & \frac{1}{\sigma} & 0 & 0 & 0 \\ 0 & 0 & \rho L_x' & 0 & \theta L_{yx} \\ 0 & 0 & 0 & \rho L_y^+ & \theta L_{yy} \\ 0 & 0 & \theta L_{yx} & \theta L_{yy} & -\alpha \end{pmatrix},$$

$$\hat{B} \triangleq \begin{pmatrix} \frac{1}{\tau} - \pi_x & 0 & 0 & 0 & 0 \\ 0 & \frac{1}{\sigma} + \mu_y - \pi_y & -|1 - \frac{\theta}{\rho}| L_{yx} & -|1 - \frac{\theta}{\rho}| L_{yy} & 0 \\ 0 & -|1 - \frac{\theta}{\rho}| L_{yx} & \frac{1}{\tau} - L_{xx}' & 0 & 0 \\ 0 & -|1 - \frac{\theta}{\rho}| L_{yy} & 0 & \frac{1}{\sigma} - \alpha - L_y^- & 0 \\ 0 & 0 & 0 & 0 & 0 \end{pmatrix},$$

with

$$L_x' \triangleq \frac{2}{1 - \rho}(\rho^{-q+1} - 1)\Big(\frac{{L_{xx}'}^2}{\pi_x b_x'} + \frac{2(1 + 2\theta + 2\theta^2)\rho^{-1} L_{yx}^2}{\pi_y b_y'}\Big),$$

$$L_y^+ \triangleq \frac{2}{1 - \rho}(\rho^{-q+1} - 1)\frac{2(1 + 2\theta + 2\theta^2)\rho^{-1} L_{yy}^2}{\pi_y b_y'},$$

$$L_y^- \triangleq \frac{2\rho}{1 - \rho}(\rho^{-q+1} - 1)\frac{L_{xy}^2}{\pi_x b_x'}.$$

Using the same argument as in the proof of Lemma 17, and noticing that $L_y'$ in eq. (113) can be written as $L_y' = L_y^+ + L_y^-$, one can show that eq. (113) holds if and only if $\hat{B} - \frac{1}{\rho}\hat{A} \succeq 0$. Therefore, it follows from (132) that

$$\hat{U}_N^t(x_*^t(\bar{y}_N^t), y_*(\bar{x}_N^t)) \leq \frac{1}{2}{\xi_0^*}^\top \hat{A} \xi_0^*$$
$$- \rho^{-N+1}\Big(\frac{1}{2}{\xi_N^*}^\top \hat{B} \xi_N^* - D_N^t(x_*^t(\bar{y}_N^t), y_*(\bar{x}_N^t)) - \frac{\theta}{\rho}S_N^t(x_*^t(\bar{y}_N^t), y_*(\bar{x}_N^t))\Big),$$

holds w.p. 1. Furthermore, define

$$G''' \triangleq \begin{pmatrix} \frac{1}{\sigma}(1 - \frac{1}{\rho}) + \mu_y - \pi_y + \alpha & (-|1 - \frac{\theta}{\rho}| - \frac{\theta}{\rho})L_{yx} & (-|1 - \frac{\theta}{\rho}| - \frac{\theta}{\rho})L_{yy} \\ (-|1 - \frac{\theta}{\rho}| - \frac{\theta}{\rho})L_{yx} & \frac{1}{\tau} - L_{xx}' & 0 \\ (-|1 - \frac{\theta}{\rho}| - \frac{\theta}{\rho})L_{yy} & 0 & \frac{1}{\sigma} - \alpha - L_y^- \end{pmatrix},$$

and recall that $D_N^t(x, y) = \frac{1}{2\rho}(\frac{1}{\tau} - \mu_x)\|x_N^t - x\|^2 + \frac{1}{2}\Big(\frac{1}{\rho\sigma} - \alpha\Big)\|y_N^t - y\|^2$. Using a similar argument as in the proof of Lemma 18, we can show that eq. (113) implies

$$\frac{1}{2}{\xi_N^*}^\top \hat{B} \xi_N^* - D_N^t\Big(x_*^t(\bar{y}_N^t), y_*(\bar{x}_N^t)\Big) - \frac{\theta}{\rho}S_N^t\Big(x_*^t(\bar{y}_N^t), y_*(\bar{x}_N^t)\Big)$$

$$= \frac{1}{2}{\xi_N^*}^\top \begin{pmatrix} \frac{1}{\tau}(1 - \frac{1}{\rho}) + \frac{\mu_x}{\rho} - \pi_x & \mathbf{0}_{1\times 3} & 0 \\ \mathbf{0}_{3\times 1} & G''' & \mathbf{0}_{3\times 1} \\ 0 & \mathbf{0}_{1\times 3} & 0 \end{pmatrix} \xi_N^* \geq 0.$$

Finally, since $x_{-1}^t = x_0^t$, $y_{-1}^t = y_0^t$, we have

$$\frac{1}{2}{\xi_0^*}^\top \hat{A} \xi_0^* \leq \Big(\frac{1}{2\tau} - \frac{\mu_x}{2}\Big)\|x_*^t(\bar{y}_N^t) - x_0^t\|^2 + \frac{1}{2\sigma}\|y_*(\bar{x}_N^t) - y_0^t\|^2.$$

Therefore, we obtain that

$$\hat{U}_N^t(x_*^t(\bar{y}_N^t), y_*(\bar{x}_N^t)) \leq \left(\frac{1}{2\tau} - \frac{\mu_x}{2}\right)\|x_*^t(\bar{y}_N^t) - x_0^t\|^2 + \frac{1}{2\sigma}\|y_*(\bar{x}_N^t) - y_0^t\|^2, \text{ holds w.p. 1.}$$

Now, we are ready to show the desired result of Lemma 23. Since $D_N^t(x_*^t(\bar{y}_N^t), y_*(\bar{x}_N^t)) \geq 0$, it follows from (130) that

$$K_N(\rho)\mathbb{E}[\mathcal{G}^t(\bar{x}_N^t, \bar{y}_N^t)] \leq \mathbb{E}\Big[\left(\frac{1}{2\tau} - \frac{\mu_x}{2}\right)\|x_*^t(\bar{y}_N^t) - x_0^t\|^2 + \frac{1}{2\sigma}\|y_*(\bar{x}_N^t) - y_0^t\|^2\Big]$$
$$+ K_N(\rho)\Big(\frac{\delta_x^2}{2\pi_x b} + (1 + 2\theta + 2\theta^2)\frac{\delta_y^2}{\pi_y b}\Big).$$

Then dividing both side by $K_N(\rho)$ completes the proof. $\qquad\square$

## F.4 A particular parameter choice

We employ the matrix inequality (MI) in eq. (113) to describe the admissible set of `VR-SAPD` parameters that guarantee convergence. Next, in Lemma 26, we compute a particular solution to it by exploiting its structure.

**Lemma 26.** *For any* $\mu_x > 0$, *let* $L_{xx}' = L_{xx} + \gamma + \mu_x$. *Let* $\theta \in (0, 1]$ *and* $\tau, \sigma > 0$ *be chosen as*

$$\theta = 1, \quad \tau = \frac{1}{L_{yx} + L_{xx}' + L_x'}, \quad \sigma = \frac{1}{2L_{yy} + L_{yx} + L_y'}, \tag{133}$$

*where* $L_x'$ *and* $L_y'$ *are defined in Lemma 23. Then* $\{\tau, \sigma, \theta, \alpha, \rho, \pi_x, \pi_y\}$ *is a solution to* (113) *for* $\rho = 1$, $\pi_x = \mu_x$, $\pi_y = \mu_y$ *and* $\alpha = L_{yx} + L_{yy}$.

*Proof.* Define $M_1 \triangleq \begin{pmatrix} \frac{1}{\tau} - L_{xx}' - L_x' & 0 & -L_{yx} \\ 0 & 0 & 0 \\ -L_{yx} & 0 & L_{yx} \end{pmatrix}$ and $M_2 \triangleq \begin{pmatrix} 0 & 0 & 0 \\ 0 & \frac{1}{\sigma} - \alpha - L_y' & -L_{yy} \\ 0 & -L_{yy} & L_{yy} \end{pmatrix}$. Our choice of $\{\rho, \pi_x, \pi_y, \alpha\}$ implies that (113) holds whenever

$$M_1 + M_2 = \begin{pmatrix} \frac{1}{\tau} - L_{xx}' - L_x' & 0 & -L_{yx} \\ 0 & \frac{1}{\sigma} - \alpha - L_y' & -L_{yy} \\ -L_{yx} & -L_{yy} & L_{yx} + L_{yy} \end{pmatrix} \succeq \mathbf{0}.$$

Our choice of $\alpha = L_{yx} + L_{yy}$, and $\tau, \sigma > 0$ as in (133) implies that $M_1 \succeq 0$ and $M_2 \succeq 0$. Thus, $M_1 + M_2 \succeq 0$. $\qquad\square$

Next, based on this lemma, we will give an explicit parameter choice for Algorithm 3.

## F.5 Proof of Theorem 4

*Proof.* Lemma 26 implies that our choice of $\{\tau, \sigma, \theta, \alpha, \rho, \pi_x, \pi_y\}$ ensures that eq. (113) holds. For the outer loop, if we set N as in (11) and

$$p_1 = \frac{1}{16}, \; p_2 = \frac{19}{32}, \; p_3 = \frac{11}{32}, \; \beta_1 = \frac{4}{5}, \; \beta_2 = \frac{1}{2}, \; \zeta = 32, \tag{134}$$

all assumptions of Theorem 8 are satisfied, i.e., both the inequality system in (21) and $N \geq (1 + \zeta)M^{\text{VR}}$ hold. Specifically, $M^{\text{VR}} = 2\max\{\frac{1}{\gamma\tau} - 1, \frac{1}{\mu_y\sigma}\}$; thus, $N \geq (1 + \zeta)M^{\text{VR}}$ trivially holds by our choice of $N$ in (11). The proof of eq. (21) holding for parameters in (134) follows directly from the proof of Theorem 1.

Since all assumptions of Theorem 8 are satisfied for $\mu_x = \gamma$, $\{\tau, \sigma, \theta\}$ as in (133), $N$ and $b$ as in eq. (11) and other parameters chosen as in eq. (134), if we substitute $\mu_x = \gamma$ and the specific parameter values given in eq. (134) into eq. (22) with $\Xi_{\tau,\sigma,\theta} = \Xi^{\text{VR}} = \frac{\delta_x^2}{2\gamma b} + 5\frac{\delta_y^2}{\mu_y b}$, it follows that

$$\frac{1}{T + 1}\sum_{t=0}^{T}\mathbb{E}\left[\|\nabla\phi_\lambda(x_0^t)\|^2\right] \leq 48\gamma\left(\frac{1}{T + 1}\mathcal{G}(x_0^0, y_0^0) + \frac{\delta_x^2}{2\gamma b} + \frac{5\delta_y^2}{\mu_y b}\right). \tag{135}$$

Thus, for any $\epsilon > 0$, the right side of the above inequality can be bounded by $\epsilon^2$ when

$$\frac{48\gamma}{T+1}\mathcal{G}(x_0^0, y_0^0) \leq \frac{\epsilon^2}{6}, \quad \frac{24\delta_x^2}{b} \leq \frac{\epsilon^2}{6}, \quad \frac{240\gamma\delta_y^2}{\mu_y b} \leq \frac{2\epsilon^2}{3}. \tag{136}$$

Our choice of $b$ in (11) and $T \geq 288\mathcal{G}(x_0^0, y_0^0)\frac{\gamma}{\epsilon^2}$ ensures that all the inequalities in (136) hold. Moreover, our choice of $N$ and $\{\tau, \sigma, \theta\}$ in (11) and $\rho = 1$ together with the definitions of $L_x'$ and $L_y'$ in Lemma 23 implies (12). Furthermore, it follows from the statement of Algorithm 3 that the total computation complexity is $T(Nb/q + N(b_x' + b_y'))$, which completes the proof. $\qquad\square$

# G  Proof of Theorem 5 and preliminary technical results

Recall the definition of $\hat{\mathcal{L}}$ given in eq. (13). For any $x \in \mathcal{X}$, define $\phi(x) \triangleq \max_{y \in \mathcal{Y}} \mathcal{L}(x, y)$ and $\hat{\phi}(x) \triangleq \max_{y \in \mathcal{Y}} \hat{\mathcal{L}}(x, y)$; moreover, let $\phi_\lambda(\cdot)$ and $\hat{\phi}_\lambda(\cdot)$ be respective Moreau envelopes for some $\lambda \in (0, \gamma^{-1})$.

We first show that one can obtain an $\epsilon$-stationary point for the WCMC problem in the form of (1) such that $f(\cdot) = 0$, $\mu_y = 0$ and $\mathcal{D}_y < \infty$ by computing an $\epsilon$-stationary point for eq. (13) with $\hat{\mu}_y = \Theta(\epsilon^2/(\gamma\mathcal{D}_y^2))$. Indeed, in Lemma 27 below, we extend [24, Corollary A.8] from $g$ being an indicator function of a closed convex set to a closed convex function.

**Lemma 27.** *Under the premise of Theorem 5, for some fixed $\hat{\mu}_y = \Theta(\epsilon^2/(\gamma\mathcal{D}_y^2))$, let $x_\epsilon \in \mathcal{X}$ be such that $\|\nabla\hat{\phi}(x_\epsilon)\| \leq \epsilon/(2\sqrt{6})$, where $\hat{\phi}(x) \triangleq \max_{y \in \mathcal{Y}} \hat{\mathcal{L}}(x, y)$. Then, $x_\epsilon$ is an $\epsilon$-stationary point of $\phi(\cdot)$, i.e., $\|\nabla\phi_\lambda(x_\epsilon)\| \leq \epsilon$ for $\lambda \in (0, \gamma^{-1})$, where $\phi(x) \triangleq \max_{y \in \mathcal{Y}} \mathcal{L}(x, y)$.*

*Proof.* Below We state some useful relations that will be used later in the proof. Since $f(\cdot) = 0$, eq. (13) implies that for all $(x, y) \in \mathcal{X} \times \mathbf{dom}\, g$,

$$\nabla_x \mathcal{L}(x, y) = \nabla_x \hat{\mathcal{L}}(x, y), \quad \|\nabla_y \Phi(x, y) - \nabla_y \hat{\Phi}(x, y)\| \leq \hat{\mu}_y \mathcal{D}_y. \tag{137}$$

We define $\hat{y}_*(\cdot) \triangleq \text{argmax}_{y \in \mathcal{Y}} \hat{\mathcal{L}}(\cdot, y)$. It follows that from Lemma 11 that

$$\hat{y}_*(x_\epsilon) = \mathbf{prox}_{\alpha g}\big(\hat{y}_*(x_\epsilon) + \alpha\nabla_y \hat{\Phi}(x_\epsilon, \hat{y}_*(x_\epsilon))\big). \tag{138}$$

for any $\alpha > 0$. We are now ready for the proof of Lemma 27.

Let $y^+ \triangleq \mathbf{prox}_{\alpha g}\big(\hat{y}_*(x_\epsilon) + \alpha\nabla_y \Phi(x_\epsilon, \hat{y}_*(x_\epsilon))\big)$, then we have

$$\|y^+ - \hat{y}_*(x_\epsilon)\|$$
$$= \|\mathbf{prox}_{\alpha g}\big(\hat{y}_*(x_\epsilon) + \alpha\nabla_y \Phi(x_\epsilon, \hat{y}_*(x_\epsilon))\big) - \mathbf{prox}_{\alpha g}\big(\hat{y}_*(x_\epsilon) + \alpha\nabla_y \hat{\Phi}(x_\epsilon, \hat{y}_*(x_\epsilon))\big)\| \tag{139}$$
$$\leq \alpha\|\nabla_y \Phi(x_\epsilon, \hat{y}_*(x_\epsilon)) - \nabla_y \hat{\Phi}(x_\epsilon, \hat{y}_*(x_\epsilon))\| \leq \alpha\hat{\mu}_y \mathcal{D}_y$$

where the first equality is by eq. (138); the second inequality is by $\|\mathbf{prox}_{\alpha g}(y_1) - \mathbf{prox}_{\alpha g}(y_2)\| \leq \|y_1 - y_2\|$ for all $y_1, y_2 \in \mathcal{Y}$ and eq. (137). Moreover, using Assumption 2 and the above inequalities, we have

$$\|\nabla_x \mathcal{L}(x_\epsilon, y^+)\| \leq \|\nabla_x \mathcal{L}(x_\epsilon, y^+) - \nabla\hat{\phi}(x_\epsilon)\| + \|\nabla\hat{\phi}(x_\epsilon)\|$$
$$\leq \|\nabla_x \mathcal{L}(x_\epsilon, y^+) - \nabla_x \mathcal{L}(x_\epsilon, \hat{y}_*(x_\epsilon))\| + \frac{\epsilon}{2\sqrt{6}} \leq L_{xy}\alpha\hat{\mu}_y D_y + \frac{\epsilon}{2\sqrt{6}},$$

where the second inequality follows from Danskin's theorem and the fact that $\|\nabla\hat{\phi}(x_\epsilon)\| \leq \epsilon/(2\sqrt{6})$; finally, the last inequality use Assumption 2 and (139). Thus, using $(a + b)^2 \leq 2(a^2 + b^2)$ for any $a, b \in \mathbb{R}$, we get

$$\|\nabla_x \mathcal{L}(x_\epsilon, y^+)\|^2 \leq \frac{\epsilon^2}{12} + 2L_{xy}^2\alpha^2\hat{\mu}_y^2 D_y^2. \tag{140}$$

Later in the proof, eq. (139) and eq. (140) will be useful when we further analyze $y^+$.

Recall that our ultimate goal is to show that $\|\nabla\phi_\lambda(x_\epsilon)\| \leq \epsilon$. Now, for some arbitrary $\mu_x > 0$, consider $\mathbf{prox}_{\lambda\phi}(x_\epsilon) = \text{argmin}_{v \in \mathcal{X}} \phi(v) + \frac{1}{2\lambda}\|v - x_\epsilon\|^2$, where $\lambda = (\mu_x + \gamma)^{-1}$. It follows from Lemma 1 that

$$\|\nabla\phi_\lambda(x_\epsilon)\|^2 = \frac{1}{\lambda^2}\|x_\epsilon - \mathbf{prox}_{\lambda\phi}(x_\epsilon)\|^2.$$

Since $\lambda = (\mu_x + \gamma)^{-1}$, $\phi(\cdot) + \frac{1}{2\lambda}\|\cdot - x_\epsilon\|^2$ is $\mu_x$-strongly convex with the unique minimizer $\mathbf{prox}_{\lambda\phi}(x_\epsilon)$; therefore,

$$
\begin{aligned}
&\max_{y\in\mathcal{Y}}\mathcal{L}(x_\epsilon, y) - \max_{y\in\mathcal{Y}}\mathcal{L}(\mathbf{prox}_{\lambda\phi}(x_\epsilon), y) - \frac{1}{2\lambda}\|\mathbf{prox}_{\lambda\phi}(x_\epsilon) - x_\epsilon\|^2 \\
&= \phi(x_\epsilon) - \phi(\mathbf{prox}_{\lambda\phi}(x_\epsilon)) - \frac{1}{2\lambda}\|\mathbf{prox}_{\lambda\phi}(x_\epsilon) - x_\epsilon\|^2 \qquad (141) \\
&\geq \frac{\mu_x}{2}\|x_\epsilon - \mathbf{prox}_{\lambda\phi}(x_\epsilon)\|^2 = \lambda^2\frac{\mu_x}{2}\|\nabla\phi_\lambda(x_\epsilon)\|^2.
\end{aligned}
$$

In the following analysis, we will continue to polish the upper bound on $\|\nabla\phi_\lambda(x_\epsilon)\|^2$ on the left hand side of eq. (141). Indeed,

$$
\begin{aligned}
&\max_{y\in\mathcal{Y}}\mathcal{L}(x_\epsilon, y) - \max_{y\in\mathcal{Y}}\mathcal{L}(\mathbf{prox}_{\lambda\phi}(x_\epsilon), y) - \frac{1}{2\lambda}\|\mathbf{prox}_{\lambda\phi}(x_\epsilon) - x_\epsilon\|^2 \\
&= \max_{y\in\mathcal{Y}}\mathcal{L}(x_\epsilon, y) - \mathcal{L}(x_\epsilon, y^+) + \mathcal{L}(x_\epsilon, y^+) - \max_{y\in\mathcal{Y}}\mathcal{L}(\mathbf{prox}_{\lambda\phi}(x_\epsilon), y) - \frac{1}{2\lambda}\|\mathbf{prox}_{\lambda\phi}(x_\epsilon) - x_\epsilon\|^2 \\
&\leq \max_{y\in\mathcal{Y}}\mathcal{L}(x_\epsilon, y) - \mathcal{L}(x_\epsilon, y^+) + \mathcal{L}(x_\epsilon, y^+) - \mathcal{L}(\mathbf{prox}_{\lambda\phi}(x_\epsilon), y^+) - \frac{1}{2\lambda}\|\mathbf{prox}_{\lambda\phi}(x_\epsilon) - x_\epsilon\|^2 \\
&\leq \max_{y\in\mathcal{Y}}\mathcal{L}(x_\epsilon, y) - \mathcal{L}(x_\epsilon, y^+) + \|x_\epsilon - \mathbf{prox}_{\lambda\phi}(x_\epsilon)\|\|\nabla_x\mathcal{L}(x_\epsilon, y^+)\| - \frac{\mu_x}{2}\|x_\epsilon - \mathbf{prox}_{\lambda\phi}(x_\epsilon)\|^2 \\
&\leq \max_{y\in\mathcal{Y}}\mathcal{L}(x_\epsilon, y) - \mathcal{L}(x_\epsilon, y^+) + \frac{\|\nabla_x\mathcal{L}(x_\epsilon, y^+)\|^2}{2\mu_x},
\end{aligned}
$$
$$(142)$$

where the second inequality follows from the $\mu_x$-strongly convexity of $\mathcal{L}(\cdot, y^+) + \frac{1}{2\lambda}\|\cdot - x_\epsilon\|^2$ and Cauchy-Schwarz inequality. Next, we continue to derive an appropriate upper bound on $\max_{y\in\mathcal{Y}}\mathcal{L}(x_\epsilon, y) - \mathcal{L}(x_\epsilon, y^+)$. Recall that $y^+ = \mathbf{prox}_{\alpha g}(\hat{y}_*(x_\epsilon) + \alpha\nabla_y\Phi(x_\epsilon, \hat{y}_*(x_\epsilon)))$; hence, the first-order optimality condition yields that

$$
-\frac{1}{\alpha}\left(y^+ - \hat{y}_*(x_\epsilon) - \alpha\nabla_y\Phi(x_\epsilon, \hat{y}_*(x_\epsilon))\right) \in \partial g(y^+).
$$

Therefore, for any $y \in \mathcal{Y}$, we have that

$$
g(y) - g(y^+) \geq \left\langle y - y^+, -\frac{1}{\alpha}\left(y^+ - \hat{y}_*(x_\epsilon) - \alpha\nabla_y\Phi(x_\epsilon, \hat{y}_*(x_\epsilon))\right)\right\rangle,
$$

which is equivalent to

$$
g(y^+) - g(y) \leq \frac{1}{\alpha}\langle y - y^+, y^+ - \hat{y}_*(x_\epsilon)\rangle - \langle\nabla_y\Phi(x_\epsilon, \hat{y}_*(x_\epsilon)), y - y^+\rangle. \qquad (143)
$$

Now, we ready to provide a useful upper bound on $\max_{y\in\mathcal{Y}}\mathcal{L}(x_\epsilon, y) - \mathcal{L}(x_\epsilon, y^+)$. Indeed, given any $\tilde{y} \in \mathrm{argmax}_{y\in\mathcal{Y}}\mathcal{L}(x_\epsilon, y)$, we have

$$
\begin{aligned}
&\max_{y\in\mathcal{Y}}\mathcal{L}(x_\epsilon, y) - \mathcal{L}(x_\epsilon, y^+) = \mathcal{L}(x_\epsilon, \tilde{y}) - \mathcal{L}(x_\epsilon, \hat{y}_*(x_\epsilon)) + \mathcal{L}(x_\epsilon, \hat{y}_*(x_\epsilon)) - \mathcal{L}(x_\epsilon, y^+) \\
&= \underbrace{\Phi(x_\epsilon, \tilde{y}) - \Phi(x_\epsilon, \hat{y}_*(x_\epsilon))}_{\textbf{part 1}} - g(\tilde{y}) + g(\hat{y}_*(x_\epsilon)) + \underbrace{\Phi(x_\epsilon, \hat{y}_*(x_\epsilon)) - \Phi(x_\epsilon, y^+)}_{\textbf{part 2}} - g(\hat{y}_*(x_\epsilon)) + g(y^+) \\
&\leq \langle\nabla_y\Phi(x_\epsilon, \hat{y}_*(x_\epsilon)), \tilde{y} - \hat{y}_*(x_\epsilon)\rangle - g(\tilde{y}) + g(y^+) \\
&\quad + \langle\nabla_y\Phi(x_\epsilon, \hat{y}_*(x_\epsilon)), \hat{y}_*(x_\epsilon) - y^+\rangle + \frac{L_{yy}}{2}\|\hat{y}_*(x_\epsilon) - y^+\|^2 \\
&= \langle\nabla_y\Phi(x_\epsilon, \hat{y}_*(x_\epsilon)), \tilde{y} - y^+\rangle - g(\tilde{y}) + g(y^+) + \frac{L_{yy}}{2}\|\hat{y}_*(x_\epsilon) - y^+\|^2 \\
&\leq \frac{1}{\alpha}\langle\tilde{y} - y^+, y^+ - \hat{y}_*(x_\epsilon)\rangle + \frac{L_{yy}}{2}\|\hat{y}_*(x_\epsilon) - y^+\|^2 \\
&= -\frac{L_{yy}}{2}\|\hat{y}_*(x_\epsilon) - y^+\|^2 + L_{yy}\langle\tilde{y} - \hat{y}_*(x_\epsilon), y^+ - \hat{y}_*(x_\epsilon)\rangle \\
&\leq L_{yy}\mathcal{D}_{\mathcal{Y}}\|y^+ - \hat{y}_*(x_\epsilon)\|,
\end{aligned}
$$
$$(144)$$

where in the first inequality, we use concavity and smoothness of $\Phi(x_\epsilon, \cdot)$ for **part 1** and **part 2**, respectively; in the second inequality, we use eq. (143); in the last equality, we set $\alpha = L_{yy}^{-1}$; and in the last inequality, we use Cauchy-Schwarz inequality and the fact that $\sup_{y_1, y_2 \in \mathbf{dom}\, g} \|y_1 - y_2\| \le \mathcal{D}_{\mathcal{Y}}$. Next, if we use eq. (144) within eq. (142), it follows that

$$
\max_{y \in \mathcal{Y}} \mathcal{L}(x_\epsilon, y) - \max_{y \in \mathcal{Y}} \mathcal{L}(\mathbf{prox}_{\lambda\phi}(x_\epsilon), y) - \frac{1}{2\lambda} \|\mathbf{prox}_{\lambda\phi}(x_\epsilon) - x_\epsilon\|^2
$$

$$
\le L_{yy}\mathcal{D}_{\mathcal{Y}}\|y^+ - \hat{y}_*(x_\epsilon)\| + \frac{\|\nabla_x \mathcal{L}(x_\epsilon, y^+)\|^2}{2\mu_x} \tag{145}
$$

$$
\le \hat{\mu}_y \mathcal{D}_y^2 + \frac{\epsilon^2}{24\mu_x} + \frac{L_{xy}^2}{L_{yy}^2} \cdot \frac{\hat{\mu}_y^2}{\mu_x} \cdot D_y^2,
$$

where the last inequality follows from eq. (139) and eq. (140) with $\alpha = L_{yy}^{-1}$.

Finally, if we use eq. (145) within eq. (141) and substitute $\lambda = (\gamma + \mu_x)^{-1}$, it follows that

$$
\frac{\mu_x}{2(\gamma + \mu_x)^2} \|\nabla\phi_\lambda(x_\epsilon)\|^2 \le \hat{\mu}_y \mathcal{D}_y^2 + \frac{\epsilon^2}{24\mu_x} + \frac{L_{xy}^2}{L_{yy}^2} \cdot \frac{\hat{\mu}_y^2}{\mu_x} \cdot D_y^2. \tag{146}
$$

Thus, choosing the free parameter $\mu_x = \gamma$ implies that

$$
\|\nabla\phi_\lambda(x_\epsilon)\|^2 \le 8\gamma\hat{\mu}_y\mathcal{D}_y^2 + \frac{\epsilon^2}{3} + 8\frac{L_{xy}^2}{L_{yy}^2} \cdot \hat{\mu}_y^2 \cdot D_y^2. \tag{147}
$$

Thus, we get $\|\nabla\phi_\lambda(x_\epsilon)\| \le \epsilon$ for $\hat{\mu}_y = \min\left\{\frac{\epsilon^2}{24\gamma\mathcal{D}_y^2}, \frac{L_{yy}}{L_{xy}} \cdot \frac{\epsilon}{2\sqrt{6}\mathcal{D}_y}\right\}$. $\qquad\square$

### G.1    Proof of Theorem 5

*Proof.* To get a worst-case complexity, as in the previous sections, let

$$
L \triangleq \max\{L_{xy}, L_{yx}, L_{xx}, L_{yy}\}, \ \delta \triangleq \max\{\delta_x, \delta_y\}, \ \gamma = L.
$$

Assumption 2 implies that $\nabla_y\hat{\Phi}$ and $\nabla_x\hat{\Phi}$ are Lipschitz such that for all $x, x' \in \mathcal{X}$ and $y, y' \in \mathbf{dom}\, g$,

$$
\|\nabla_y\hat{\Phi}(x, y) - \nabla_y\hat{\Phi}(x', y')\| \le L_{yx}\|x - x'\| + \hat{L}_{yy}\|y - y'\|,
$$

$$
\|\nabla_x\hat{\Phi}(x, y) - \nabla_x\hat{\Phi}(x', y')\| \le L_{xx}\|x - x'\| + L_{xy}\|y - y'\|,
$$

where $\hat{L}_{yy} = L_{yy} + \hat{\mu}_y$. Therefore, the proof immediately follows from Lemma 27 and Theorem 1, considering SAPD+ with VR-flag = **false** is applied on (13) with $\hat{\mu}_y = \min\left\{\frac{\epsilon^2}{24\gamma\mathcal{D}_y^2}, \frac{L_{yy}}{L_{xy}} \cdot \frac{\epsilon}{2\sqrt{6}\mathcal{D}_y}\right\}$. $\qquad\square$

## H    Details of fair classification example

In the experiment of fair classification, $\{(\mathbf{a}_i, b_i)\}_{i=1}^n$ denotes the (data,label) pairs of the labeled image data set. $a_i \in \mathbb{R}^{d_1 \times d_2 \times c}$, and $b_i$ is a label associated with one of the $K$-classes, i.e., $b_i \in \mathcal{C} \triangleq \{C_j\}_{j=1}^K$ with $K \le n$. We employ the classifier

$$
h(\cdot\,; \mathbf{x}) : \mathbf{a}_i \in \mathbb{R}^{d_1 \times d_2 \times c} \to \mathbf{p}_i \in \mathbb{R}^K,
$$

where $\mathbf{p}_i = (p_{ij})_{j=1}^K$ s.t. $\sum_{j=1}^K p_{ij} = 1$ and $p_{ij} \ge 0$ for $j = 1, 2, .., K$, and $\mathbf{x}$ is the parameters of the classifier. Specifically, $h(\cdot; \mathbf{x})$ is a CNN with the structure as follows:

$$
[input] \to [conv - elu - maxpool] \times 3 \to [fc - elu] \times 2 \to [softmax]
$$

where *exponential linear unit* (elu) [8] is the smoothed variant of *rectified linear units* (relu) activation function. Furthermore, given the input $\{(\mathbf{a}_i, b_i)\}_{i=1}^n$ and the output $\{\mathbf{p}_i\}_{i=1}^n$, the loss functions $\{l_j\}_{j=1}^K$ used in eq. (15) are

$$
\ell_j(\{(\mathbf{a}_i, b_i)\}_{i=1}^n; \mathbf{x}) = -\frac{1}{N_j} \sum_{i=1}^n \log(p_{ij})\mathbf{1}_{C_j}(b_i)
$$

where $N_j$ is the number of data with label $C_j$, i.e., $N_j = \sum_{i=1}^{n} \mathbf{1}_{C_j}(b_i)$ and

$$\mathbf{1}_{C_j}(b_i) = \begin{cases} 1 & \text{if } b_i = C_j \\ 0 & \text{o.w.} \end{cases}$$

and $p_{ij}$ is the $j$-th element of $\mathbf{p}_i$, and $\mathbf{p}_i = h(\mathbf{a}_i; \mathbf{x})$.

# I    Additional analyses on the related work

In some of the existing work on WCSC problems, particularly [17, 16, 26, 37], except for $\kappa_y = L/\mu_y$, the individual effects of $L$ or $\mu_y$ are not explicitly stated in the final complexity bounds. To better compare existing bounds with ours, it is necessary to state the complexity bound dependence on $L$ and $\mu_y$. For example, Huang *et al.* [17, 16] assume that $\frac{1}{\mu_y} \leq L$, that is equivalent to $L \geq \sqrt{\kappa_y}$; however, a constant factor depending on $L$ was ignored in their oracle complexity result. Moreover, Huang *et al.*[17] employ a different convergence metric and claim that they obtain a competitive result. It turns out that their convergence metric is scaled by an algorithmic constant and when their results are converted into GNP metric, i.e., $\|\nabla\phi(\cdot)\|$, this constant adversely affects their complexity bounds. A similar issue with the claimed complexity bounds also exists in [16], where the complexity bound are computed after the objective function is rescaled. In this section, to provide a fair comparison,

- we give an explicit oracle complexity bound for the related works in [17, 16, 26, 37];
- we discuss those parts in their analysis that are not convincing, and try our best to provide the corrected and optimized complexity bounds based on their analysis.

Without loss of generality, for the sake of easier comparison, we consider the *smooth minimax* problems, i.e., $\min_{x \in \mathcal{X}} \max_{y \in \mathcal{Y}} \mathcal{L}(x, y) = \Phi(x, y)$. We first fix the notation to unify the discussion for the WCSC setting, i.e., $\mathcal{L}(x, y)$ is weakly convex in $x$ and strongly concave in $y$.

Recall that $\phi(x) \triangleq \max_{y \in \mathcal{Y}} \mathcal{L}(x, y)$; thus, $\phi(\cdot)$ is differentiable and we use $\|\nabla\phi(\cdot)\|$ as the convergence metric. In addition, we let $\phi_* \triangleq \inf_{x \in \mathcal{X}} \phi(x)$ and recall that $y_*(\cdot) = \text{argmax}_{y \in \mathcal{Y}} \mathcal{L}(\cdot, y)$. Moreover, for simplicity of the notation, we consider the worst-case complexity bounds using $L$, i.e.,

$$L = \max\{L_{xy}, L_{yx}, L_{xx}, L_{yy}\}, \quad \kappa_y = \frac{L}{\mu_y}, \quad \delta = \max\{\delta_x, \delta_y\}, \quad \gamma = L. \tag{148}$$

## I.1    Revisit of [17, Theorem 1]

In this section, we provide the oracle complexity of Huang *et al.*[17, Theorem 1] using the metric $\|\nabla\phi(\cdot)\|$ for the Stochastic Mirror Descent Algorithm (SMDA), stated in [17, algorithm 1]. Let $\tau, \sigma$ be the primal and dual stepsizes, respectively, $\eta$ be the momentum parameter, $b$ be the large batchsize, and $u$ be convexity modulus of the Bregman distance generating function. We also list our notational convention in table 2 for reader's convenience.

Below, we restate the convergence result of SMDA for the class of Bregman distance functions such that $D_t(x, x') \triangleq (x - x')^\top H_t(x - x')/2$ for some $H_t \succ 0$ –this class of Bregman functions are used for all the numerical experiments reported in [17].

**Theorem 9.** *[17, Thoerem 1] Suppose Assumptions 1, 2, 3 hold with $f(\cdot) = g(\cdot) = 0$. Let $\{x_t, y_t\}_{t=1}^{T}$ be generated by SMDA, stated in [17, Algorithm 1], employing a stochastic first-order oracle to sample stochastic partial derivatives. For parameters chosen as $\eta \in (0, 1]$, $\tau \in (0, \min\{\frac{3u}{4L(1+\kappa_y)}, \frac{9\eta u\mu_y \sigma}{800\kappa_y^2}, \frac{2\eta\mu_y u\sigma}{25L^2}\}]$ and $\sigma \in (0, \frac{1}{6L}]$, let $\eta_t = \eta$, $\tau_t = \tau$ and $\sigma_t = \sigma$ for $t \geq 0$. Then, for any given initial point $(x_0, y_0)$, it holds that*

$$\frac{1}{T} \sum_{t=1}^{T} \mathbb{E}[\|\mathbf{G}^t\|] \leq \frac{4\sqrt{2(\phi(x_0) - \phi_*)}}{\sqrt{3T\tau u}} + \frac{4\sqrt{2}\Delta_0}{\sqrt{3T\tau u}} + \frac{10\delta}{\sqrt{3bu}} + \frac{20\delta\sqrt{\eta\sigma}}{3\sqrt{\tau u\mu_y b}}, \tag{149}$$

*where $\phi(x) = \max_y \mathcal{L}(x, y)$, $\phi_* = \inf_{x \in \mathcal{X}} \phi(x)$, $\Delta_0 = \|y_0 - y_*(x_0)\|$, $y_*(x_0) = \text{argmax}_{y \in \mathcal{Y}} \mathcal{L}(x_0, y)$, $\mathbf{G}^t = H_t^{-1}\nabla\phi(x_t)$, and $H_t$ is a diagonal matrix such that $H_t \succeq u\mathbf{I}$ for all $t \geq 1$ and $u > 0$.*

| Notation in [17] | Notation in our paper | Meaning |
|:---:|:---:|:---:|
| $\gamma$ | $\tau$ | primal stepsize |
| $\lambda$ | $\sigma$ | dual stepsize |
| $L_f$ | $L$ | Lipschitz constant as in (148) |
| $\mu$ | $\mu_y$ | concavity modulus of $\mathcal{L}(x,\cdot)$ |
| $\kappa$ | $\kappa_y$ | condition number |
| $\sigma$ | $\delta$ | variance bound for the SFO |
| $b_1$ | $b'$ | small batch size for VR methods |
| $\rho$ | $u$ | convexity modulus of Bregman distance generating function |

Table 2: Important notation for [17] and this paper.
**Table notes.** (1) SFO: stochastic first-order oracle. (2) $u$ is only used in the analysis provided in this section.

**Remark 11.** *When $f(\cdot) = g(\cdot) = 0$, it follows from the update rules and the definition of Bregman distance function in [17, eq.(12-13), eq.(22-23)] that*

$$\mathbf{G}^t = H_t^{-1} \nabla \phi(x_t),$$

*where $H_t$ is a diagonal matrix such that $H_t \succeq u\mathbf{I}$. Note that*

$$\mathbf{G}^t = \nabla \phi(x_t) \iff H_t = \mathbf{I}.$$

*We noticed that the authors chose the value of $u$ to improve their bounds; but, without addressing its effect on $\mathbf{G}^t$. More precisely, they still use $\|\mathbf{G}^t\|$ as the convergence metric and compare their complexity results with those papers using $\|\nabla\phi(x_t)\|$ as the convergence metric.*

In the following corollary, we will provide the optimal complexity for SMDA based the result in eq. (149), i.e., [17, Thoerem 1].

**Corollary 1.** *Suppose Assumptions 1, 2, 3 hold with $f(\cdot) = g(\cdot) = 0$, and $\frac{1}{\mu_y} \leq L$ hold[8]. Consider the setting of Theorem 9, then SMDA [17, Algorithm 1] can generate $x_\epsilon$ such that $\mathbb{E}\left[\|\nabla\phi(x_\epsilon)\|\right] \leq \epsilon$ by requiring at most $\mathcal{O}(\frac{\kappa_y^5 \delta^2}{\mu_y^2 \epsilon^4})$ stochastic first-order oracle calls.*

*Proof.* Recall that $H_t \succeq u\mathbf{I}$, $\mathbf{G}^t = H_t^{-1}\nabla\phi(x_t)$ and $H_t$ is a diagonal matrix; therefore, we can obtain a tight upper bound on $\mathbb{E}[\|\nabla\phi(x_t)\|]$ using eq. (149) as follows:

$$\frac{1}{T}\sum_{t=1}^{T}\mathbb{E}[\|\nabla\phi(x_t)\|] \leq \frac{4\sqrt{2(\phi(x_0) - \phi_*)}}{\sqrt{3T}}\sqrt{\frac{u}{\tau}} + \frac{4\sqrt{2}\Delta_0}{\sqrt{3T}}\sqrt{\frac{u}{\tau}} + \frac{10\delta}{\sqrt{3b}} + \frac{20\delta\sqrt{\eta\sigma}}{3\sqrt{\mu_y b}}\sqrt{\frac{u}{\tau}}. \quad (150)$$

If we use their parameter choices, i.e., $\eta \in (0,1]$,

$$\sigma = \mathcal{O}\left(\frac{1}{L}\right), \quad \tau = u\,\min\left\{\frac{3}{4L(1+\kappa_y)}, \frac{9\eta\mu_y\sigma}{800\kappa_y^2}, \frac{2\eta\mu_y\sigma}{25L^2}\right\}, \quad u = \mathcal{O}(L^\nu), \quad (151)$$

for some free parameter $\nu \geq 0$, then we get

$$\frac{u}{\tau} = \max\left\{\frac{4L(1+\kappa_y)}{3}, \frac{800\kappa_y^2}{9\eta\mu_y\sigma}, \frac{25L^2}{2\eta\mu_y\sigma}\right\} = \Omega(\kappa_y^3), \quad (152)$$

where the second term leads to $\kappa_y^3$. It is essential to note that $\tau$ choice in (151) implies that $u/\tau$ ratio is independent of $u$; hence, the parameter $u$ indeed does not affect the bound on the right-hand-side of eq. (150). Therefore, contrary to what is suggested in [17], choosing different values for $u$ through picking different $\nu \geq 0$ values indeed is not useful for proving tighter bounds in GNP metric $\|\nabla\phi(x_k)\|$ in this simple scenario using their parameter choices.

---

[8]The assumption $\frac{1}{\mu_y} \leq L$ is also made in [17].

Note eq. (150) can be simplified as

$$\frac{1}{T}\sum_{t=1}^{T}\mathbb{E}[\|\nabla\phi(x_t)\|] \leq \mathcal{O}\Big(\sqrt{\frac{\kappa_y^3(\phi(x_0)-\phi_*)}{T}} + \frac{\delta}{\sqrt{b}} + \frac{\delta\kappa_y^2}{\sqrt{b}L}\Big).$$

Thus, for any $\epsilon > 0$, to find point $x_t$ such that $\mathbb{E}[\|\nabla\phi(x_t)\|] \leq \epsilon$, one should choose $t \geq T$ for

$$T = \mathcal{O}\Big(\frac{\kappa_y^3}{\epsilon^2}(\phi(x_0)-\phi_*)\Big), \quad b = \mathcal{O}\Big(\frac{\kappa_y^4\delta^2}{L^2\epsilon^2}\Big),$$

which leads to the oracles complexity of

$$2bT = \mathcal{O}\Big(\frac{\kappa_y^7\delta^2}{L^2\epsilon^4}\Big) = \mathcal{O}\Big(\frac{\kappa_y^5\delta^2}{\mu_y^2\epsilon^4}\Big).$$

$\square$

## I.2 Revisit of [17, Theorem 3]

In this section, we provide the oracle complexity of Huang *et al.* [17, Theorem 3] using the metric $\|\nabla\phi(\cdot)\|$ for the Stochastic Mirror Descent Algorithm with variance reduction (SMDA-VR), stated in [17, algorithm 2]. Let $\tau, \sigma$ be the primal and dual stepsizes, respectively, $\eta$ be the momentum parameter, $b$ be the large batchsize, $b'$ be the small batchsize, $q$ be the period for sampling large batch size (i.e., once every $q$ batches is large), and $u$ be the strongly-convex constant of the Bregman distance generating function. We also list our notational convention in table 2 for reader's convenience.

Below, as we did in the previous section for SMDA, we restate the convergence result of SMDA-VR for the class of Bregman distance functions such that $D_t(x, x') = (x - x')^\top H_t(x - x')/2$ for some $H_t \succ 0$ –this class of Bregman functions are used for all the numerical experiments reported in [17].

**Theorem 10.** *[17, Thoerem 3] Suppose Assumptions 1, 2, 3 hold with $f(\cdot) = g(\cdot) = 0$. Let $\{x_t, y_t\}_{t=1}^T$ be generated by SMDA-VR, stated in [17, Algorithm 2], employing a stochastic first-order oracle to sample stochastic partial derivatives. For parameters chosen as $\eta \in (0, 1]$, $\tau = (0, \min\left\{\frac{3u}{4L(1+\kappa_y)}, \frac{\eta\mu_y\sigma u}{38L^2}, \frac{3u}{19L^2\eta}, \frac{u\eta}{8}, \frac{9u\eta\mu_y\sigma}{400\kappa_y^2}, \right\}]$ and $\sigma \in (0, \min\left\{\frac{1}{6L}, \frac{9\mu_y}{100\eta^2L^2}\right\}]$, let $\eta_t = \eta$, $\tau_t = \tau$ and $\sigma_t = \sigma$ for $t \geq 0$ and $b' = q$. Then, for any given initial point $(x_0, y_0)$, we have*

$$\frac{1}{T}\sum_{t=1}^{T}\mathbb{E}[\|\mathbf{G}^t\|] \leq \frac{4\sqrt{2(\phi(x_0)-\phi_*)}}{\sqrt{3T\tau u}} + \frac{4\sqrt{2}\Delta_0}{\sqrt{3T\tau u}} + \frac{2\sqrt{2}\delta}{\sqrt{\tau u\eta bL}}. \tag{153}$$

*where $\phi(x) = \max_y \mathcal{L}(x, y)$, $\phi_* = \inf_{x\in\mathcal{X}}\phi(x)$, $\Delta_0 = \|y_0 - y_*(x_0)\|$, $y_*(x_0) = \arg\max_{y\in\mathcal{Y}}\mathcal{L}(x_0, y)$, $\mathbf{G}^t = H_t^{-1}\nabla\phi(x_t)$, and $H_t$ is a diagonal matrix such that $H_t \succeq u\mathbf{I}$ for some $u > 0$.*

In the following corollary, we will provide the optimal complexity for SMDA-VR based the result in eq. (153), i.e., [17, Thoerem 3].

**Corollary 2.** *Suppose Assumptions 1, 2, 3 hold with $f(\cdot) = g(\cdot) = 0$, and $\frac{1}{\mu_y} \leq L$ hold[9]. Consider the setting of Theorem 10, then SMDA-VR [17, Algorithm 2] can generate $x_\epsilon$ such that $\mathbb{E}[\|\nabla\phi(x_\epsilon)\|] \leq \epsilon$ by requiring at most $\mathcal{O}(\frac{\kappa_y^5\delta^2}{\mu_y\epsilon^3})$ stochastic first-order oracle calls.*

*Proof.* Recall that $H_t \succeq u\mathbf{I}$, $\mathbf{G}^t = H_t^{-1}\nabla\phi(x_t)$ and $H_t$ is a diagonal matrix; therefore, we can obtain a tight upper bound on $\mathbb{E}[\|\nabla\phi(x_t)\|]$ using eq. (153) as follows:

$$\frac{1}{T}\sum_{t=1}^{T}\mathbb{E}[\|\nabla\phi(x_t)\|] \leq \frac{4\sqrt{2(\phi(x_0)-\phi_*)}}{\sqrt{3T}}\sqrt{\frac{u}{\tau}} + \frac{4\sqrt{2}\Delta_0}{\sqrt{3T}}\sqrt{\frac{u}{\tau}} + \frac{2\sqrt{2}\delta}{\sqrt{\eta bL}}\sqrt{\frac{u}{\tau}}. \tag{154}$$

If we use their parameter choices, i.e., $\eta \in (0, 1]$,

$$\sigma = \mathcal{O}\Big(\frac{1}{\kappa_y L}\Big), \quad \tau = u\min\left\{\frac{3}{4L(1+\kappa_y)}, \frac{\eta\mu_y\sigma}{38L^2}, \frac{3}{19L^2\eta}, \frac{\eta}{8}, \frac{9\eta\mu_y\sigma}{400\kappa_y^2}\right\}, \quad u = \mathcal{O}(L^{1+\nu}), \tag{155}$$

---
[9]The assumption $\frac{1}{\mu_y} \leq L$ is also made in [17].

for some free design parameter $\nu \geq 0$, then we get

$$\frac{u}{\tau} = \max\left\{\frac{4L(1+\kappa_y)}{3}, \frac{38L^2}{\eta\mu_y\sigma}, \frac{19L^2\eta}{3}, \frac{8}{\eta}, \frac{400\kappa_y^2}{9\eta\mu_y\sigma}\right\} = \Omega(\kappa_y^4),$$

where the last term leads to $\kappa_y^4$. It is essential to note that $\tau$ choice in (155) implies that $u/\tau$ ratio is independent of $u$; hence, the parameter $u$ indeed does not affect the bound on the right-hand-side of eq. (154). Therefore, contrary to what is suggested in [17], for the simple scenarios considered here choosing different values for $u$ through picking different $\nu \geq 0$ values indeed is not useful for proving tighter bounds in GNP metric $\|\nabla\phi(x_k)\|$ with their parameter choices.

Note that eq. (154) can be simplified as

$$\frac{1}{T}\sum_{t=1}^{T}\mathbb{E}[\|\nabla\phi(x_t)\|] \leq \mathcal{O}\left(\sqrt{\frac{\kappa_y^4(\phi(x_0) - \phi_*)}{T}} + \delta\frac{\kappa_y^2}{\sqrt{bL}}\right).$$

Thus, for any $\epsilon > 0$, to find point $x_t$ such that $\mathbb{E}[\|\nabla\phi(x_t)\|] \leq \epsilon$, one should choose $t \geq T$ for

$$T = \mathcal{O}\left(\frac{\kappa_y^4(\phi(x_0) - \phi_*)}{\epsilon^2}\right), \quad b = \mathcal{O}\left(\frac{\kappa_y^4\delta^2}{L^2\epsilon^2}\right),$$

which leads to the oracle complexity of

$$4b'T + 2bT/q = \mathcal{O}\left(\frac{b'\kappa_y^4}{\epsilon^2} + \frac{\kappa_y^8\delta^2}{L^2\epsilon^4}/q\right).$$

Since their parameter choice requires $b' = q$, to optimize the above bound, we let $b' = q = \mathcal{O}\left(\frac{\kappa_y^2}{L\epsilon}\right)$, which leads to

$$\mathcal{O}\left(\frac{\kappa_y^6\delta^2}{L\epsilon^3}\right) = \mathcal{O}\left(\frac{\kappa_y^5\delta^2}{\mu_y\epsilon^3}\right).$$

$\square$

## I.3 Revisit of [26, Theorem 1]

Recall that $\phi(x) = \max_y \mathcal{L}(x,y)$ and $\phi_* = \inf_{x\in\mathcal{X}}\phi(x)$. In this paper, the total oracle complexity to find point $x_\epsilon$ such that $\mathbb{E}[\|\nabla\phi(x_\epsilon)\|] \leq \epsilon$ is given by

$$\mathcal{O}(\kappa_y^2\epsilon^{-2}\log(\kappa_y/\epsilon)) + \mathcal{O}(T/q \cdot b) + \mathcal{O}(T \cdot b' \cdot m) \tag{156}$$

where[10]

$$T = \left\lceil\frac{100\kappa_y L\Delta_f}{9\epsilon^2}\right\rceil, \ q = \lceil\epsilon^{-1}\rceil, \ b = \left\lceil\frac{2250}{19}\delta^2\kappa_y^2\epsilon^{-2}\right\rceil, \ b' = \left\lceil\frac{3687}{76}\kappa_y q\right\rceil, \ m = \lceil 1024\kappa_y\rceil. \tag{157}$$

Given an arbitrary initial point $x_0$, let $y_0$ be obtained by inexactly solving $\max_y \mathcal{L}(x_0, y)$, and they define $\Delta_f = \mathcal{L}(x_0, y_0) - \frac{134\epsilon^2}{\kappa_y L} - \phi_*$. In (157), the other parameters are defined as follows: $b$ is the large batchsize, $b'$ is the small batchsize, $q$ is the period such that once every $q$ outer iterations, SREDA calls for a large batchsize, $T$ is the number of the outer iterations and $m$ is the number of the inner iterations –each outer iteration requires $m$ inner iterations and each inner iteration calls for a small batchsize. Then eq. (156) becomes $\mathcal{O}(\frac{L\kappa_y^3}{\epsilon^3})$.

## I.4 Revisit of [37, Theorem 1]

Recall that $\phi(x) = \max_y \mathcal{L}(x,y)$ and $\phi_* = \inf_{x\in\mathcal{X}}\phi(x)$. In this paper, the precise parameter selection for [37, Theorem 1] is provided in [37, Theorem 3] of the supplementary material. Using these parameter choice implies that the total oracle complexity to find point $x_\epsilon$ such that $\mathbb{E}[\|\nabla\phi(x_\epsilon)\|] \leq \epsilon$ is given by

$$T \cdot b' \cdot m + \left\lceil\frac{T}{q}\right\rceil \cdot b + T_0, \tag{158}$$

---

[10]In [26], there is a typo in the choice of $b = \lceil\frac{2250}{19}\delta^2\kappa_y^{-2}\epsilon^2\rceil$. Here, we provide the correct one.

for an arbitrary initial point $x_0$, where the number of outer iterations, $T$, the number of the inner iterations per each outer iteration, $m$, are set as follows:

$$T = \max\left\{\frac{3345\kappa_y}{\epsilon^2},\ 6600(1+\kappa_y)L\frac{(\phi(x_0)-\phi_*)}{\epsilon^2}\right\}, \quad b = \frac{9366\delta^2\kappa_y^2}{\epsilon^2},$$

$$b' = \frac{\kappa_y}{\epsilon}, \quad m = 52\kappa_y - 1, \quad q = \frac{2}{13(1+\kappa_y)}\frac{\kappa_y}{\epsilon}, \quad T_0 = \mathcal{O}(\kappa_y\log(\kappa_y)).$$

Above $b$ is the large batchsize, $b'$ is the small batchsize, $q$ is the period such that once every $q$ outer iterations, a large batch size is sampled rather than a small batch size. Then eq. (158) becomes $\mathcal{O}(\frac{L\kappa_y^3}{\epsilon^3})$.

## I.5 Revisit of [16, Theorem 12]

In this section, we provide the oracle complexity of [16, Theorem 12] using the metric $\|\nabla\phi(\cdot)\|$ for the Accelerated first-order Momentum Descent Ascent (`ACC-MDA`) algorithm, stated in [16, algorithm 3]. Let $\tau, \sigma$ be the primal and dual stepsizes, respectively, $\{\eta_t\}$ be the momentum parameter sequence, and $b$ be the batchsize. We also list our notational convention in table 3 for reader's convenience.

| Notations in [16] | Notations in our paper | Meaning |
|:---:|:---:|:---:|
| $\gamma$ | $\tau$ | primal stepsize |
| $\lambda$ | $\sigma$ | dual stepsize |
| $L_f$ | $L$ | Lipschitz constant as in (148) |
| $L_g$ | $L(1+\kappa_y)$ | $L$-smooth constant of $\phi(x)$ |
| $\tau$ | $\mu_y$ | concavity modulus of $\mathcal{L}(x,\cdot)$ |

Table 3: Important notations for [16] and this paper.

Below, we restate the convergence result of `ACC-MDA` reported in [16].

**Theorem 11.** *[16, Thoerem 12] Suppose Assumptions 1, 2, 3 hold with $f(\cdot) = g(\cdot) = 0$. Let $\{x_t, y_t\}_{t=1}^T$ be generated by `ACC-MDA` algorithm, stated in [16, Algorithm 3], when applied to the smooth minimax problem $\min_{x\in\mathcal{X}}\max_{y\in\mathcal{Y}}\mathcal{L}(x,y) = \Phi(x,y)$. For some given $p > 0$, let $\eta_t = \frac{p}{(\psi+t)^{1/3}}$ for all $t \geq 0$, $\tau \in (0,\ \min\{\frac{\sigma\mu_y}{2L}\sqrt{\frac{2b}{8\sigma^2+75\kappa_y^2b}}, \frac{\psi^{1/3}}{2L(1+\kappa_y)p}\}]$ and $\sigma \in (0, \min\{\frac{1}{6L}, \frac{27b\mu_y}{16}\}]$ such that $\psi \geq \max\{2, p^3, (c_1p)^3, (c_2p)^3\}$ for some $c_1 \geq \frac{2}{3p^3} + \frac{9\mu_y^2}{4}$ and $c_2 \geq \frac{2}{3p^3} + \frac{75L^2}{2}$. Then for any given $x_0$, we have*

$$\frac{1}{T}\sum_{t=1}^T \mathbb{E}[\|\nabla\phi(x_t)\|] \leq \frac{\sqrt{2M''}\psi^{1/6}}{T^{1/2}} + \frac{\sqrt{2M''}}{T^{1/3}}, \tag{159}$$

*where $\phi(x) = \max_y \mathcal{L}(x,y)$, $\Delta_0 = \|y_0 - y_*(x_0)\|^2$, $y_*(x_0) = \arg\max_{y\in\mathcal{Y}}\mathcal{L}(x_0,y)$, $M'' = \frac{\phi(x_0)-\phi_*}{\tau p} + \frac{9L^2\Delta_0}{p\sigma\mu_y} + \frac{2\psi^{1/3}\delta^2}{b\mu_y^2p^2} + \frac{2(c_1^2+c_2^2)\delta^2p^2}{b\mu_y^2}\ln(\psi+T)$, and $\phi_* = \inf_{x\in\mathcal{X}}\phi(x)$.*

**Remark 12.** *[16, Remarks 13 and 14] When $b = \mathcal{O}(\kappa_y^\nu)$ for $\nu > 0$ and $\kappa_y^\nu \leq \frac{8}{81L\mu_y}$, they claim that they can obtain the gradient complexity of $\tilde{\mathcal{O}}(\kappa_y^3\epsilon^{-3})$ if $\nu = 3$, and $\tilde{\mathcal{O}}(\kappa_y^{2.5}\epsilon^{-3})$ if $\nu = 4$. However, for the assumption $\kappa_y^\nu \leq \frac{8}{81L\mu_y}$ to hold in general, one needs to rescale the original objective function $\mathcal{L}(x,y)$ with some $s \in (0,1]$ to define*

$$\mathcal{L}_s(x,y) \triangleq s \cdot \mathcal{L}(x,y). \tag{160}$$

*Then the Lipschitz constant of $\nabla\mathcal{L}_s$, strongly concavity modulus of $\mathcal{L}(x,\cdot)$ for any $x \in \mathcal{X}$ and the variance bound of the stochastic oracle for $\nabla_x\mathcal{L}_s$ and $\nabla_y\mathcal{L}_s$ can be written as $sL$, $s\mu_y$, and $s^2\delta^2$, respectively. We notice that the effect of scaling $\mathcal{L}$ on the problem parameters is not discussed in [16] and eq. (159) is directly used to derive the convergence result assuming $\kappa_y^\nu \leq \frac{8}{81L\mu_y}$. As a consequence, the complexity results of $\tilde{\mathcal{O}}(\kappa_y^3\epsilon^{-3})$, $\tilde{\mathcal{O}}(\kappa_y^{2.5}\epsilon^{-3})$ do not hold without loss of generality unless the original function $\mathcal{L}$ satisfies the restrictive assumption of $\kappa_y^\nu \leq \frac{8}{81L\mu_y}$.*

In the following discussion, we analyze the effect of scaling $\mathcal{L}$ on the complexity bounds whenever $\kappa_y^\nu \leq \frac{8}{81L\mu_y}$ is not satisfied for the original objective function $\mathcal{L}$, and we provide complexity bounds holding without loss of generality that are optimized by choosing $\nu > 0$ properly. Now, consider implementing ACC-MDA on an appropriately scaled problem $\min_x \max_y \mathcal{L}_s(x, y)$ where $\mathcal{L}_s$ is defined in (160). Let

$$L_s \triangleq sL, \quad \mu_s \triangleq s\mu_y, \quad \delta_s \triangleq s\delta. \tag{161}$$

Note that the condition numbers of $\mathcal{L}_s$ and $\mathcal{L}$ are the same, and are equal to $\kappa_y$, i.e., $\kappa_y = \frac{L}{\mu_y} = \frac{L_s}{\mu_s}$. In the upcoming discussion, suppose that $s \in (0, 1]$ is chosen such that $\kappa_y^\nu \leq \frac{8}{81L_s\mu_s}$.

To facilitate the complexity analysis and make the upcoming discussion easier, first we restate Theorem 11 for the function $\mathcal{L}_s$, where we used the relation $\nabla\phi$ and the derivative of $\max_y \mathcal{L}_s(\cdot, y)$; indeed, the derivative of $\max_y \mathcal{L}_s(\cdot, y)$ is equal to $s\nabla\phi(\cdot)$, where $\phi(x) = \max_y \mathcal{L}(x, y)$.

**Theorem 12.** *[16, Thoerem 12] Suppose Assumptions 1, 2, 3 hold with $f(\cdot) = g(\cdot) = 0$. Let $\{x_t, y_t\}_{t=1}^T$ be generated by ACC-MDA algorithm, stated in [16, Algorithm 3], when applied to the smooth minimax problem $\min_{x \in \mathcal{X}} \max_{y \in \mathcal{Y}} \mathcal{L}_s(x, y) = s \cdot \mathcal{L}(x, y)$. For some given $p \geq 0$, let $\eta_t = \frac{p}{(\psi+t)^{1/3}}$ for all $t \geq 0$, $\tau \in (0, \min\{\frac{\sigma\mu_s}{2L_s}\sqrt{\frac{2b}{8\sigma^2+75\kappa_y^2 b}}, \frac{\psi^{1/3}}{2L_s(1+\kappa_y)p}\}]$ and $\sigma \in (0, \min\{\frac{1}{6L_s}, \frac{27b\mu_s}{16}\}]$ such that $\psi \geq \max\{2, p^3, (c_1'p)^3, (c_2'p)^3\}$ for some $c_1' \geq \frac{2}{3p^3} + \frac{9\mu_s^2}{4}$ and $c_2' \geq \frac{2}{3p^3} + \frac{75L_s^2}{2}$. Then for any given $x_0$, we have*

$$\frac{1}{T}\sum_{t=1}^T \mathbb{E}[\|\nabla\phi(x_t)\|] \leq \frac{1}{s}\left(\frac{\sqrt{2M_s''}\psi^{1/6}}{T^{1/2}} + \frac{\sqrt{2M_s''}}{T^{1/3}}\right), \tag{162}$$

*where $\phi(x) = \max_y \mathcal{L}(x, y)$, $\Delta_0 = \|y_0 - y_*(x_0)\|^2$, $y_*(x_0) = \arg\max_{y \in \mathcal{Y}} \mathcal{L}(x_0, y)$, $M_s'' = \frac{s(\phi(x_0)-\phi_*)}{\tau p} + \frac{9L_s^2\Delta_0}{p\sigma\mu_s} + \frac{2\psi^{1/3}\delta_s^2}{b\mu_s^2 p^2} + \frac{2(c_1'^2+c_2'^2)\delta_s^2 p^2}{b\mu_s^2}\ln(\psi+T)$, and $\phi_* = \inf_{x \in \mathcal{X}} \phi(x)$.*

Next, following the analysis in [16, Remarks 10 and 13], we provide a particular parameter choice for ACC-MDA so that it is applicable to the setting of Theorem 12.

**Lemma 28.** *Under the premise of Theorem 12. Suppose $\kappa_y^\nu \leq \frac{8}{81L_s\mu_s}$, $b = \kappa_y^\nu$ for some $\nu > 0$, and*

$$\frac{\sigma\mu_s}{2L_s}\sqrt{\frac{2b}{8\sigma^2+75\kappa_y^2 b}} \leq \frac{\psi^{1/3}}{2L_s(1+\kappa_y)p}. \tag{163}$$

*If $\sigma = \min\{\frac{1}{6L_s}, \frac{27b\mu_s}{16}\}$ and $\tau = \min\{\frac{\sigma\mu_s}{2L_s}\sqrt{\frac{2b}{8\sigma^2+75\kappa_y^2 b}}, \frac{\psi^{1/3}}{2L_s(1+\kappa_y)p}\}$, then $\psi = \Theta(\max\{1, L_s^6\})$ satisfies the condition in Theorem 12 and*

$$\sigma = \Theta(b\mu_s), \quad \tau^{-1} = \Theta\left(\frac{\kappa_y^3}{bL_s}\right). \tag{164}$$

**Remark 13.** *The conditions $\kappa_y^\nu \leq \frac{8}{81L_s\mu_s}$ and eq. (163), and the choice $b = \kappa_y^\nu$ are as suggested in [16].*

*Proof.* Since $\kappa_y^\nu \leq \frac{8}{81L_s\mu_s}$, we have $\sigma = \frac{27b\mu_s}{16} = \Theta(b\mu_s)$. Furthermore, eq. (163) implies that we can simplify $\tau$ as

$$\tau = \frac{\sigma\mu_s}{2L_s}\sqrt{\frac{2b}{8\sigma^2+75\kappa_y^2 b}}.$$

Then it follows that,

$$\tau^{-1} = \Theta\left(\frac{\kappa_y}{b\mu_s}\sqrt{b\mu_s^2+\kappa_y^2}\right) = \Theta\left(\frac{\kappa_y^2}{bL_s}(\kappa_y^{\nu/2}\mu_y + \kappa_y)\right) = \Theta\left(\frac{\kappa_y^2}{b}\left(\kappa_y^{\nu/2-1} + \frac{\kappa_y}{L_s}\right)\right) = \Theta\left(\frac{\kappa_y^3}{bL_s}\right),$$

where we use the relation $\kappa_y^\nu \leq \frac{8}{81L_s\mu_s} \Rightarrow L_s^2 \leq \frac{8}{81}\kappa_y^{1-\nu}$ for the last equality. Next, from eq. (163) and the requirement on $\psi$ in Theorem 12, a sufficient condition $\psi$ is

$$\psi \geq \max\left\{2, \, p^3, \, (c_1'p)^3, \, (c_2'p)^3, \, \left(\sigma\mu_s(1+\kappa_y)p\sqrt{\frac{2b}{8\sigma^2+75\kappa_y^2 b}}\right)^3\right\}, \tag{165}$$

Now we consider the components of $\max$ operator in (165). Note positive constant $p$ can be chosen independent of other problem parameters, e.g., $p = 1$. Furthermore, the requirement on $c_1', c_2'$ can be satisfied for

$$c_1' = \Theta(\mu_s^2), \quad c_2' = \Theta(L_s^2). \tag{166}$$

Finally, using $\sigma = \frac{27b\mu_s}{16}$ together with $b = \kappa_y^\nu$ yields that

$$\sigma\mu_s(1+\kappa_y)p\sqrt{\frac{2b}{8\sigma^2 + 75\kappa_y^2 b}} = \Theta\left(\kappa_y b\mu_s^2\sqrt{\frac{1}{b\mu_s^2 + \kappa_y^2}}\right) = \Theta\left(L_s^2\kappa_y^{\nu-1}\sqrt{\frac{1}{\kappa_y^{\nu-2}L_s^2 + \kappa_y^2}}\right)$$

$$= \Theta\left(L_s^2\kappa_y^{\nu-1}\sqrt{\frac{1}{\kappa_y^2}}\right) \leq \Theta(1),$$

where we use the relation $\kappa_y^\nu \leq \frac{8}{81L_s\mu_s} \Rightarrow L_s^2 \leq \frac{8}{81}\kappa_y^{1-\nu}$ for the last equality and the last inequality. Therefore, using the above relations within eq. (165), we observe that one can set

$$\psi^{1/3} = \Theta(\max\{1, L_s^2\}), \tag{167}$$

which completes the proof. $\qquad\square$

Next, we will use the parameters in Lemma 28 to provide an optimized complexity for `ACC-MDA` [16, Algorithm 12] to generate $x_\epsilon$ such that $\mathbb{E}\left[\|\nabla\phi(x_\epsilon)\|\right] \leq \epsilon$.

**Corollary 3.** *Suppose Assumptions 1, 2, 3 hold with $f(\cdot) = g(\cdot) = 0$, and $\kappa_y^\nu > \frac{8}{81L\mu_y}$ for the original function $\mathcal{L}$. Running* `ACC-MDA` *on $\min_x \max_y \mathcal{L}_s(x, y)$ for*

$$s = \frac{2\sqrt{2}}{9}\frac{1}{L}\kappa_y^{(1-\nu)/2}, \tag{168}$$

*and $b = \kappa_y^\nu$, one can generate $x_\epsilon$ such that $\mathbb{E}\left[\|\nabla\phi(x_\epsilon)\|\right] \leq \epsilon$ requiring at most $\tilde{O}(\frac{L^{1.5}\kappa_y^{3.5}}{\epsilon^3})$ stochastic first-order oracle calls.*

*Proof.* It follows from Theorem 12 that that

$$\frac{1}{T}\sum_{t=1}^T \mathbb{E}[\|\nabla\phi(x_t)\|] \leq \frac{\sqrt{2M_s''}}{s}\left(\frac{\psi^{1/6}}{T^{1/2}} + \frac{1}{T^{1/3}}\right). \tag{169}$$

Based on Lemma 28, let $\psi = \Theta(\max\{1, L_s^6\})$; thus, $\frac{\psi^{1/6}}{T^{1/2}} \leq \frac{1}{T^{1/3}}$ when $T$ is large enough. Therefore, for all sufficiently small $\epsilon > 0$, $\frac{1}{T^{1/3}} \leq \frac{s}{\sqrt{2M_s''}} \cdot \frac{\epsilon}{2}$ implies that $\frac{\psi^{1/6}}{T^{1/2}} \leq \frac{s}{\sqrt{2M_s''}} \cdot \frac{\epsilon}{2}$, and we get

$$\frac{1}{s}\frac{\sqrt{2M_s''}}{T^{1/3}} \leq \frac{\epsilon}{2} \implies \min_{t\in\{0,\ldots,T\}}\mathbb{E}[\|\nabla\phi(x_t)\|] = \epsilon. \tag{170}$$

Moreover, note that eq. (168) implies $\kappa_y \leq \frac{8}{81L_s\mu_s}$; thus, we can choose $\tau, \sigma, b$ as in Lemma 28 which satisfy

$$\sigma = \Theta(b\mu_s), \quad \tau^{-1} = \Theta\left(\frac{\kappa_y^3}{bL_s}\right), \quad b = \kappa_y^\nu.$$

Recall that $c_1', c_2'$ chosen as in (166) and $\psi$ chosen as in (167) satisfy all the required conditions in Theorem 12; hence, $M_s'' = \frac{s(\phi(x_0)-\phi_*)}{\tau p} + \frac{9L_s^2\Delta_0}{p\sigma\mu_s} + \frac{2\psi^{1/3}\delta_s^2}{b\mu_s^2 p^2} + \frac{2(c_1'^2+c_2'^2)\delta_s^2 p^2}{b\mu_s^2}\ln(\psi+T)$ implies that

$$\frac{1}{s^2}M_s'' = \Theta\left(\frac{\kappa_y^3}{sbL_s} + \frac{\kappa_y^2}{s^2 b} + \frac{\delta_s^2}{s^2 b\mu_s^2}\max\{1, L_s^2\} + \frac{\kappa_y^2 L_s^2\delta_s^2}{s^2 b}\ln(\psi+T)\right)$$

$$= \tilde{\Theta}\left(\frac{\kappa_y^3}{s^2 bL} + \frac{\kappa_y^2}{s^2 b} + \frac{\delta^2}{s^2 b\mu_y^2}\max\{1, s^2 L^2\} + \frac{s^2\kappa_y^2 L^2\delta^2}{b}\right). \tag{171}$$

Moreover, to satisfy eq. (170), one needs to choose $T \geq \Theta\left(\left(\frac{1}{s^2}M_s''\right)^{3/2}\frac{1}{\epsilon^3}\right)$. Since the total oracle complexity is $bT$, we obtain that

$$bT \geq \tilde{\Theta}\left(\frac{1}{\epsilon^3}\left(\frac{\kappa_y^{9/2-\nu/2}}{s^3 L^{3/2}} + \frac{\kappa_y^{3-\nu/2}}{s^3} + \frac{\delta^3 \kappa^{-\nu/2}}{s^3\mu_y^3}\max\{1, s^3 L^3\} + s^3\kappa_y^{3-\nu/2}L^3\delta^3\right)\right). \quad (172)$$

From eq. (168), i.e., $s^2 = \frac{8}{81}\frac{1}{L^2}\kappa_y^{1-\nu}$, it follows that

$$bT \geq \tilde{\Theta}\left(\frac{1}{\epsilon^3}\left(L^{3/2}\kappa_y^{\nu+3} + L^3\kappa_y^{\nu+3/2} + \delta^3\kappa^{\nu+3/2}\max\{1, \kappa_y^{\frac{3-3\nu}{2}}\} + \kappa_y^{9/2-2\nu}\delta^3\right)\right). \quad (173)$$

When $\nu \geq 1$, we have

$$bT \geq \tilde{\Theta}\left(L^{3/2}\kappa_y^{\nu+3} + L^3\kappa_y^{\nu+3/2} + \delta^3\kappa_y^{3-\nu/2} + \kappa_y^{9/2-2\nu}\delta^3\right),$$

the optimal value is achieved at $\nu = 1$ and $bT \geq \Theta\left(\frac{L^{1.5}\kappa_y^4}{\epsilon^3}\right)$; when $\nu < 1$, we have

$$bT \geq \tilde{\Theta}\left(L^{3/2}\kappa_y^{\nu+3} + L^3\kappa_y^{\nu+3/2} + \delta^3\kappa_y^{\nu+3/2} + \kappa_y^{9/2-2\nu}\delta^3\right),$$

the optimal value is achieved at $\nu = \frac{1}{2}$ and $bT \geq \tilde{\Theta}\left(\frac{L^{1.5}\kappa_y^{3.5}}{\epsilon^3}\right)$, which completes the proof. $\qquad \square$

**Remark 14.** *In [16], Huang* et al. *claims the oracle complexity of $\tilde{\mathcal{O}}(\kappa_y^3\epsilon^{-3})$ for $\nu = 3$, and $\tilde{\mathcal{O}}(\kappa_y^{2.5}\epsilon^{-3})$ for $\nu = 4$. However, our analysis leading to eq. (173) demonstrates that the complexities would be $\tilde{\mathcal{O}}(\frac{L^{1.5}\kappa_y^6}{\epsilon^3})$ for $\nu = 3$ and $\tilde{\mathcal{O}}(\frac{L^{1.5}\kappa_y^7}{\epsilon^3})$ for $\nu = 4$.*