# OpenReview forum: "SAPD+: An Accelerated Stochastic Method for Nonconvex-Concave Minimax Problems"
_NeurIPS.cc/2022/Conference — NeurIPS 2022 Accept_

### Official Review · Reviewer_DZXf · 2022-07-11

**Rating:** 5
**Confidence:** 4
**Soundness:** 3 good
**Presentation:** 3 good
**Contribution:** 3 good

**Summary:**

This paper is concerned with weakly convex-strongly concave (WCSC) and weakly convex-merely concave (WCMC) saddle-point problems. A new stochastic method, SAPD+, is proposed, which is based on the inexact proximal point method. Compared with the existing works, the proposed method does not require compactness for neither the promal nor the dual domain, and can achieve the best $\kappa$ dependence. This result is nice and outstands much of the work. In addition, they also proposed a variance-reduced version of SAPD+ and achieves the best known complexity.


**Questions:**

1. Note that function $Phi(\dot, y)$ is assumed to be weakly convex and smooth. Since a smooth function is necessarily weakly convex, the reviewer is wondering whether the assumption that the function $Phi(\dot, y)$ is weakly convex is redundant and should be removed?

2. In the work of Davis and Drusvyatskiy [8], an additional upper bound of the gradient norm is required for weakly convex objective functions. The reviewer is wondering how this assumption is relaxed in this present work? Please explain.

3. Regarding the assumption on the compactness.
(1) In Section 2.2, the authors relax the compactness assumption by introducing an additional upper bound on the subgradient norm. This additional assumption should be mentioned in the Contribution, since the boundedness of feasible domains is replaced by the boundedness of subgradient.
(2) The reviewer is also wondering whether Assumption 5 is more realistic than Assumption 4. More advantages of using this assumption instead of the assumption on the compactness should be explained in detail.

4. Does the VR parameters (b', b, q) have influences on the performances? It is not clear from the present work.

5. Some typos to be fixed:
(1) Line 200 ``Suppose Assumption 1, 2, 3 and 4....'' should be ``Suppose Assumptions 1, 2, 3 and 4....''

(2) Line 285 ``Suppose Assumption 1, 3, 6 and 7....'' should be ``Suppose Assumptions 1, 3, 6 and 7....''

**Ethics Review Area:**

["I don’t know"]

**Strengths And Weaknesses:**

originality: This paper is well organised in general. In Introduction, the motivation, problem formulation and literature review are included. Contributions are listed in a clear way. The main results and the proofs are easy to follow. The reviewer only has one minor comment related to the originality of this work: In the conclusion, it is suggested that some future directions should be included.

quality: The results are technically sound and theoretically supported. The algorithm proposed is also new.

clarity: The proof is thorough and clear. The simulation and the comparison are adequate.

significance: This paper takes the stocastic setting of the nonconcave-convex mini-max saddle point problem, which, compared with the determinstic setting, is more realistic and practical since the modern datasets have large-dimensions and computing its exact gradients is infeasible. In addition, this paper relaxes the commonly adopted assumption that both the promal and the dual domain should be compact, which is a significant improvement compared with existing works. The assumptions and results obtained in the work are well explained relative to existing works.

---

> ### Author Response · Authors · 2022-08-02
> **Response to reviewer DZXF**
>
> Thanks for the insightful comments and suggestions.\
> Q1: The reviewer is right, i.e., even if we remove the weak convexity assumption on $\Phi$, our results will continue to be true. That said, smoothness constant $L$ is usually much larger than the weak convexity modulus, i.e., $\gamma$, and we wanted to show the complexity dependence on $L$ and $\gamma$, separately. Moreover, we describe how parameters of SAPD+ should be chosen depending on $L$ and $\gamma$ separately–see eq.(11). This flexibility helped us improve the performance of SAPD+ both in theory and practice as our results can exploit the scenario when $\gamma$ is known and $\gamma \ll L$ rather than directly letting $\gamma=L$. We should also point out that although we assume potentially nonsmooth functions $f$ and $g$ to be convex, our proof directly follows even if we assume $f$ and $g$ to be weakly convex.
>
> Q2: This is an important question. The short intuitive answer is related to how we can remove the compactness assumption, generally used in other related works. Even if the primal/dual domains can be unbounded, under some conditions on the subdifferentials of $f$ and $g$ (Assumption 5), we can show that the iterate sequence remains bounded in the L2 sense. This property of the SAPD+ iterates enables us to show the desired result without assuming an upper bound on the gradient norm, e.g., when $f=g=0$ or $f,g$ are indicator functions of convex sets. In case we assume compactness, an upper bound on the gradient norm is immediately implied because of the smoothness of $\Phi$.
>
> Q3: (1) The reviewer is right, we will state in contributions that compactness can be exchanged with Assumption 5 on $f$ and $g$. (2) We think that neither assumption is superior to the other. It is important to emphasize that we use $\inf$ in Assumption 5, not $\sup$. Thus, we do not require a uniformly bounded subdifferential for $f$ and $g$. Using $\inf$ is helping us to extend our result to indicator functions of convex sets that are not necessarily bounded – please also see Remark 5. In the revised version we will extend this discussion.
>
> Q4: This is a significant point.\
> (1) Theorem 4 provides a lower bound on $b$ to ensure the convergence of SAPD+, i.e., the convergence is guaranteed when $b$ is large enough. Indeed, the choice of $b$ controls the variance term. In Lemma 21, the variance term $\Xi^{VR}$ satisfies $\mathcal{O}(1/b)$, i.e., a larger $b$ implies a smaller $\Xi^{VR}$. In addition, Remark 4 says that for any given $b$, our complexity result depends on $(b’,q)$ as follows: $\mathcal{O}(\kappa_y b’+\frac{\kappa_y^2b}{b’}+\kappa_y^2q+\frac{\kappa_y b}{q})$. To minimize this bound, the optimal choice is $b’_*=\mathcal{O}(\sqrt{\kappa_y b})$ and $q_*=\mathcal{O}(\sqrt{\frac{b}{\kappa_y}})$ as functions of $b$.\
> (2) In practice, following the theoretical results above, we first choose a large $b$, then determine the value of $(b’,q)$. Indeed, we find that the product of $b’_*$ with $q_*$ is $\mathcal{O}(b)$, as a result, in practice we adjust the values of $(b’,q)$ in a way so that $b’q=\mathcal{O}(b)$. Our experimental results validate that this parameter choice works well in practice.
> When we experiment with different $(b,b’,q)$ values, we observed the following marginal effects in practice, which were expected based on the discussion in (1) above:\
> (a)For fixed $(b',q)$, as we increase $b$, the variance gets smaller; but, the convergence gets slower as well.\
> (b)For fixed $(b,q)$, as we increase $b’$, the convergence gets faster for a while, and then gets slower after a threshold. As $b’$ gets larger, the variance gets smaller in our observed range of $b’$.\
> (c)For fixed $(b,b’)$, as we increase $q$, the convergence gets faster at first, and then gets slower after a threshold. For larger $q’$, the variance gets larger in our observed range of $q$.\
> We will add a remark that explains this point further in the revised version.
>
> Q5: Thanks for pointing out the typos. We will fix them.
>
> Future works: We appreciate the reviewer's reminder and will add two future directions in the conclusion:\
> (1) One is to extend our results to weakly convex-weakly concave(WCWC) setting. The recent work of “The Complexity of Constrained Min-Max Optimization’’ shows that, for general smooth WCWC objectives, the computation of even approximate first-order local min-max solutions is intractable–this motivates the identification of structural assumptions on $\Phi$ so that these intractability barriers can be bypassed. We suggest to identify proper structures for nonconvex-nonconcave $\Phi$ that is amenable to efficient first-order primal-dual methods, e.g., Ostrovskii et al.[34] require the existence of a solution to a Minty VI.\
> (2) Another one is to consider line search when Lipschitz constants are unknown. Although some recent works consider line search for deterministic convex-concave problems, stochastic line search for saddle point (SP) problems has not been considered yet.

---

### Official Review · Reviewer_JXF9 · 2022-07-11

**Rating:** 6
**Confidence:** 1
**Soundness:** 3 good
**Presentation:** 3 good
**Contribution:** 2 fair

**Summary:**

The authors analyse the performance of a stochastic method (SADP+ and variance reduced SAPD+) for solving weakly convex-strongly concave  and weakly convex merely concave saddle-point problems. Under milder assumptions than previous works (no compactness needed), they prove that SADP+ (and its VR counterpart) obtain complexities which have better dependencies in the condition number for WCSC problems and a $\log$ improvement for WCMC problems.

**Questions:**

- how would you justify the originality / significance of your results when building on the works of [14] [42]

**Strengths And Weaknesses:**

Please keep in mind that this submission is unfortunately *not* in my area of expertise.

Though the paper is very dense and compact, it is is overall well written and easy to follow,  The contributions are clearly stated and all the results are introduced and commented. Table 1 provides (what seems to be) an extensive summary of the relevant related results. They show an improvement of the order $\kappa_y^2$ for WCSC problems ($\kappa_y$ for WCMC problems) which seems significant. I am however not capable of judging whether relaxing the compacting assumption is a significant improvement over previous results.

The numerical experiments are convincing: several algorithms are compared, the two settings are clearly explained and hyperparameter selection is given each time.

My major concern regarding the submission is the importance and significance of their results. The algorithm they consider is the natural stochastic version of the APD algorithm proposed in A PRIMAL-DUAL ALGORITHM FOR GENERAL CONVEX-CONCAVE SADDLE POINT PROBLEMS [14] (which is oddly not cited in the main text). Furthermore SADP+ was already analysed in ROBUST ACCELERATED PRIMAL-DUAL METHODS FOR COMPUTING SADDLE POINTS [ 42] for strongly convex-strongly concave problems. Hence I have concerns regarding the originality of the paper.

---

> ### Author Response · Authors · 2022-08-02
> **Response to reviewer JXF9**
>
> We completely understand the reviewer’s question regarding the originality / significance of our results that are built on the works of [14] and [42]. SAPD in [42] is an extension of [14] to the stochastic gradient setting when the function is SCSC (strongly convex-strongly concave) – APD in [14] is deterministic and does not cover SCSC setting. The algorithmic structure both in [14] and [42] are very similar; however, extending these results to the more general setting of weak convexity is not trivial and requires a completely different algorithmic structure and analysis techniques for obtaining the results in our NeurIPS submission as detailed in the appendix (provided in our supplementary file).
>
> To be precise, for WCSC(weakly convex-strongly concave) setting, to be able to show $\kappa_y^2$ and $\kappa_y$ improvement over algorithms with the best known complexity results in the literature so far, we had to carefully analyze an inexact proximal point method (iPPM) and its interaction with SAPD for solving SCSC subproblems – this is completely new as iPPM framework does not appear neither in [14] nor in [42]. The originality/significance of this paper stems from (1) our tight analysis of iPPM in a stochastic setting establishing complexity results with the best condition number dependency and also (2) our showing $\log(1/\epsilon)$ improvement for WCMC(weakly convex-merely concave) problems. In our analysis, we also devise a new first-time bound that permits us to translate convergence in the gradient norm of the Moreau envelope (GNME) metric to the gradient norm of the primal function (GNP) metric with explicit dependency on eps. This allows us to obtain stronger and precise guarantees in the GNP metric. To summarize, our paper provides a new careful analysis of iPPM in both GNME and GNP metrics while only using SAPD as an efficient sub-solver. We appreciate the reviewer’s question, we will add further discussions about this in the revised version.

---

### Official Review · Reviewer_yayL · 2022-07-11

**Rating:** 7
**Confidence:** 4
**Soundness:** 3 good
**Presentation:** 2 fair
**Contribution:** 3 good

**Summary:**

This paper proposed SAPD+ algorithm for differentiable weakly-convex-strongly/merely-concave (WCSC / WCMC) minimax problems, which is based on iteratively solving SCSC subproblems; while further assuming smoothness of the gradient estimator, the proposed algorithm can be incorporated with SPIDER-type technique for solving the problems. The proposed algorithm achieves new SOTA complexities in all three settings.

**Questions:**

The results here are pretty interesting. While I found it hard to check thoroughly the long proof. My main questions:

1. The algorithm idea is pretty simple, just iteratively augment it to SCSC, and use existing oracle to approximately solve it. The conciseness looks exciting to me. I found that there are also some other literature like PG-SMD [35] and Catalyst [41] which shares similar idea. While here this paper is the first one to achieve such sharp complexity. It would be great if authors can provide any comparison to existing literature, and intuitive understanding on why it achieves acceleration here (and others no).

2. I personally may think that here "weakly convex" is a little misleading, Assumption 2 on the Lipschitz smoothness has already implied weak convexity (as mentioned in Remark 3), while the difference should be the modulus of weak convexity. I may view weak convexity be more common in nonsmooth optimization (e.g., [35]), but here the coupling term $\Phi$ is differentiable (ignoring regularizers $f$ and $g$).

In general I think the results here are important and significantly improve over existing results. While the presentation can be further polished for better readability.

**Limitations:**

Yes

**Strengths And Weaknesses:**

Strength:
1. Claim an algorithm with new SOTA complexity
2. The algorithm is pretty simple
3. Further relax the common compactness assumption

Weakness:
1. The proof is messy, also too many notations (maybe unavoidable though), I felt hard and confused to follow the idea when reading.

---

> ### Author Response · Authors · 2022-08-02
> **Response to reviewer yayL**
>
> Thank you for your comments and suggestions. We completely understand the difficulties with reading the proof. We will simplify the notations and to polish the presentation in the revised version.
>
> Q1: Yes, you are right. Our SAPD+ algorithm employs the inexact proximal point method that is also employed in other works, e.g., PG-SMD[35], [39] by Yan et al., and Catalyst[41]. To get a sharper complexity than those in other related work, our method exploits the following ideas:\
> (1) Comparison with [35,39]: We use an accelerated algorithm, i.e., SAPD, to solve the strongly convex-strongly concave(SCSC) sub-problems, while [39] by Yan et al. and PG-SMD [35] adopt unaccelerated methods, e.g., PG-SMD [35] uses stochastic mirror descent (SMD) that is equivalent to the gradient descent ascent algorithm when one employs the Euclidean setting. Using SAPD allows us to solve the subproblems more efficiently and this is one of the reasons why we can achieve a better complexity compared to [35] and [39].
>
> (2) Comparison with [41]: We consider a general class of problems with stochastic gradients and non-smooth $f$ and $g$, while Catalyst[41] considers smooth deterministic saddle point(SP) problems and their finite sum counterparts. We should point out that for our special case of deterministic weakly convex-strongly concave(WCSC) SP problems, Catalyst[41] is faster. Indeed, for smooth deterministic WCSC SP problems, our complexity is $\mathcal{O}(\kappa_y L \epsilon^{-2})$ while Catalyst[41] requires $\mathcal{O}(\sqrt{\kappa_y}log^2(\kappa_y) L \epsilon^{-2})$. That said, our main contribution is for the stochastic setting and unlike our results, the work in [41] does not provide any guarantees for stochastic problems or when there are non-smooth regularizers.
>
> (3) Our analysis is based on the expected gap metric for the subproblems, which allows us to effectively trade off between convergence of bias and variance terms for stochastic problems in case of variance reduction. This is one of the reasons we can achieve a better complexity and kappa dependence for stochastic problems when using variance reduction.
>
> (4) To our knowledge, our analysis is the first one to rigorously characterize the computational effort required for translating the convergence in GNME (gradient norm of Moreau Envelope) to that in GNP (gradient norm of the primal function). This is beneficial in two key aspects: (a) tighter analysis by using GNME (because of the nice properties of the Moreau Envelope); (b) negligible cost to translate the analysis from GNME to GNP.
>
> To be more precise, recall that we define $\phi(\cdot)=\max_{y\in\cal{Y}}\cal{L}(\cdot,y)$, and $\phi_{\lambda}(\cdot)$ is the Moreau Envelope of $\phi$. Convergence in GNP means $||\nabla\phi(\cdot)||\leq \epsilon$ and convergence in GNME means $||\nabla\phi_{\lambda}(\cdot)||\leq \epsilon$. In this paper, we first adopt GNME to analyze our algorithms in order to get sharper complexity bounds. Indeed, Theorem 2 (see Lemma 14 and Lemma 15 for details) shows that we can convert a GNME guarantee to a GNP guarantee by incurring only little additional cost compared to the cost of obtaining a GNME guarantee such that the overall worst-case complexity (in terms of worst-case dependency on the target accuracy) remains the same for both metrics. Indeed, when there is a non-smooth $f$, we show equivalence between the metrics based on GNME and the generalized gradient mapping, which is equivalent to GNP when $f=0$. Moreover, although the proof of Theorem 2 is algorithm-dependent, we emphasize that our proof technique can be applied to related future works that adopt GNME metric for convergence.
>
> In a high-level summary, the success of our approach relies on adopting an efficient SCSC sub-solver, using a tight analysis of GNME metric within the iPPM framework, and effectively converting guarantees from the GNME metric to the GNP metric. We will give this discussion in the revised version.
>
> Q2: The reviewer is absolutely right that the Lipschitz smoothness implies weak convexity (as mentioned in Remark 3) such that the modulus of weak convexity $\gamma$ is bounded by the Lipschitz constant $L$. That said, $L$ is usually much larger than $\gamma$, and we wanted to show the complexity dependence on $L$ and $\gamma$ separately. Moreover, in our suggested parameter choice, eq. (11), we allow the parameters of SAPD+ to depend on $L$ and $\gamma$ separately. This has also been useful to us in our experiments with real data when determining the value of $\gamma$ instead of directly letting $\gamma=L$. We will add a remark that explains this point further in the revised version, we thank the reviewer for pointing this out.

---

> > ### Comment · Reviewer_yayL · 2022-08-09
> > **Thank you and Further**
> >
> > I appreciate authors' response, which further compares existing literature to provide some intuitions on the improvement. I encourage authors to add such discussion (also proof sketch) into the paper.
> >
> > ---
> > **Follow-up on Q1 before**:
> > Can I say the key improvement only lies in the complexity for solving the inner loop? Because I found that compared to the Catalyst [41] you mentioned, the outer loop is basically the same ($\mathcal{O}(L\epsilon^{-2})$). While for the inner loop, here the SAPD [42] for SCSC can shave the $\kappa_y$-dependence in the "stochastic part" ($\delta^2/\epsilon^2$) compared to existing other SCSC literature, e.g., *Hsieh et al. "On the convergence of single-call stochastic extra-gradient methods." NeurIPS 2019* (Theorem 5 therein, they are using point distance, which corresponds to your primal-duality gap by multipling the smoothness parameter $L$.).
> >
> > I am interested in it because the algorithm here suggests a framework based on iPPM, and if there exists a better algorithm for SCSC than SAPD in the stochastic regime, maybe we can further improve the complexity.
> >
> > ---
> > **Some further**:
> > I believe the results here are significant, so I vote 7 now. But the proof here is too lengthy and messy in my honest opinion, and lacks some intuitive sketch, which is challenging for a conference venue desiring a thorough check on the proof.
> >
> > TBH I cannot confirm that the results in the paper are all correct, in such short reviewing time frame, which also corresponds to my readability concern. I may suggest that this work needs a thorough polishement on the typesetting, and deserves a submission to MP/SIOPT/JMLR for a better review (of course, at a cost of time though).
> >
> > ---
> >
> > **Some further points**:
> > 1. Line 669, what is "$\pm$" there?
> > 2. Line 875 & 879, do you miss anything inside the inner product? Now it is $\langle\Delta, x\rangle$, but in Lemma 9, it is $\langle\Delta, x-x'\rangle$ (also similar for $y$).
> > 3. The definition of $K_N(\cdot)$ seems to be unavailable in the main context? I did not find it until Section A.1.
> > 4. For **basic** readability, I may suggest that many many long (and nontrivial) expressions and additional definitions in the appendix should be put into a separate \equation environment, rather than squeezing inline, e.g., Line 577-579.
> > 5. A nomenclature list will be highly appreciated.

---

> > > ### Author Response · Authors · 2022-08-09
> > > **Further Response to Reviewer yayL**
> > >
> > > We appreciate all the feedback. We will add the discussion on existing literature and the proof sketch into the paper.
> > >
> > > **Follow-up on Q1 before**: We understand the reviewer’s point here; that said, the outer iter complexity of $\mathcal{O}(L\epsilon^{-2})$ in Catalyst[41] is for the deterministic setting while in our submission we get the same iteration complexity for the stochastic setting – it is not clear if the outer iter complexity analysis in Catalyst paper extends to the stochastic setting. On the other hand, we absolutely agree that using a solver for the subproblems that is more efficient than SAPD can indeed improve the overall complexity in the stochastic setting. However, we should point out that SAPD already has some optimality properties. For the variance term (which depends on the noise level), SAPD [42] has a complexity of $\tilde{\mathcal{O}}((\delta_{x}^{2} / \mu_{x}+\delta_{y}^{2} / \mu_{y}) / \epsilon)$ which is optimal up to a log factor. For the bias term (which characterizes how fast the initial conditions are forgotten), as mentioned in *On Lower Iteration Complexity Bounds for the Saddle Point Problems* by Zhang el al., the lower bound for SCSC subproblems is $\Omega((\sqrt{\frac{L_{xx}}{\mu_x}+\frac{L^2_{yx}}{\mu_x\mu_y}+ \frac{L_{xx}}{\mu_y}})\ln(1/\epsilon))=\Omega(\kappa\ln(1/\epsilon))$  while SAPD achieves $\mathcal{O}((\frac{L_{xx}}{\mu_x}+\sqrt{\frac{L^2_{yx}}{\mu_x\mu_y}}+ \frac{L_{xx}}{\mu_y})\ln(1/\epsilon))=\mathcal{O}(\kappa\ln(1/\epsilon))$. Therefore, SAPD is optimal in terms of $\kappa$ and $\epsilon$ dependence and an algorithm for SCSC subproblems with a better complexity may only give an improvement in second-order terms in terms of $\kappa$. We will clarify this point further in the revised version.
> > >
> > > **Proof sketch**: We admit that our proof is lengthy. To make it more readable, we will provide a sketch of the proof in the revised version. On a high level, our proof can be divided into two main parts: (1) inner loop and (2) outer loop analysis.
> > >
> > > (1) Inner loop: We analyze the convergence in terms of expected gap function $\mathcal{G}^t$ in Lemma 2. Our key inequality is given in eqs. (49) and (50), which contain the **variance terms** $P_k^t,Q_k^t$ and the **bias term** $U_N^t(x,y)$. To complete the proof, we analyze the variance terms in Lemmas 7-9 and bound $P_k^t,Q_k^t$ in eqs. (50b) and (50c). Moreover, to analyze the bias term, we construct a telescoping sum in the proof of Lemma 2 by using a linear matrix inequality~(LMI), i.e., eq. (51). We provide Lemmas 10 and 11 to explain the equivalence between different LMIs, i.e., eqs. (16), (39) and (40), to support the proof and conclude eq. (52). Finally, combining the analysis of the variance terms and the bias term, we obtain the desired inner loop complexity.
> > >
> > > (2) Outer loop: Since the convergence for inner loop is in terms of $\mathcal{G}^t$, our goal is to study the relationship between $\mathcal{G}^t$ and GNME $||\nabla_x\phi_{\lambda}(x)||$. Indeed, Lemmas 3-5 allows us to translate the expected gap result of inner loops to the convergence in terms of GNME for the outer loops. In Theorem 6, we provide the convergence result in the GNME metric and the requirements on the parameters, which leads to the complexity bound in Theorem 1.
> > >
> > > **Q1**: In lines 648 and 649, we defined two auxiliary sequences $\tilde{y}^+_{k}$ and $\tilde{y}_k^-$. The $\pm$ in line 669 means the equation holds for both  $\tilde{y}_k^+$  and $\tilde{y}_k^-$.
> > >
> > > **Q2**: We bound $Q_k^t$ terms in eq. (98) and eq. (50c) in different ways. Bounding $Q_k^t$ in eq. (98) requires Lemma 20 while bounding $Q_k^t$ in eq. (50c) requires Lemma 9. For simplicity, here we only talk about the first term of $Q_k^t$ , i.e., $<\Delta_k^{\Phi^t,x},\hat{x}_{k+1}^t - x_{k+1}^{t}>$, and a similar argument also holds for other terms of $Q_k^t$.
> > >
> > >  Indeed, in the Lemma 20, we assume that $f,g=0$, thus, the iteration relation is quite simple, i.e.,
> > > $$
> > > x_{k+1}^t = x_k^t - \tau \nabla_x \Phi^t (x_{k}^t, y_{k+1}^t).
> > > $$
> > > Also note that $\mathbb{E}[<\Delta_k^{\Phi^t,x},\hat{x}_{k+1}^t >] =0$.  As a result, in Lemma 20, we can just consider the term
> > >
> > > $$
> > > \mathbb{E}[<\Delta_k^{\Phi^t,x}, x_{k+1}^t>].
> > > $$
> > >
> > > However, in eq.(98) and Lemma 9, without the assumption that $f=g=0$,  although it is still true that
> > > $$
> > > \mathbb{E}[<\Delta_k^{\Phi^t,x}, \hat{x}_{k+1}^t>]=0,
> > > $$
> > >
> > > we don’t have the simple relation of $x_{k+1}^t$ anymore because of the proximal operator on $f$. As a result, we have to consider the term
> > >
> > > $$\mathbb{E}[<\Delta_k^{\Phi^t,x}, \hat{x}_{k+1}^{t} - x_{k+1}^t>$$
> > > as a whole part, and use Cauchy inequality as well as Lemma 7 to compute the bound.
> > >
> > > **Q3**: We will add the expression of $K_N(\cdot)$ before Algorithm 1.
> > >
> > > **Q4**: We completely agree with you; we will move some of the long expressions to a separate \equation environment for enhancing readability.
> > >
> > > **Q5**: Thanks for the suggestion, we will provide a nomenclature list in the revised version.

---

> > > > ### Comment · Reviewer_yayL · 2022-08-09
> > > > **Response**
> > > >
> > > > Thank you for the quick response, which basically clarifies all confusions. Authors can integrate the above sketch into the revision.
> > > >
> > > > Also you mentioned this is the first analysis based on GNME in WCSC. I understand that Moreau envelope will smoothen the primal function in WCMC case. But in WCSC, the primal function is already smooth. You mentioned that the analysis based on GNME is **"beneficial in two key aspects"**, because of the **nice properties** of ME. I did not fully understand this benefit in WCSC case, would you mind elaborate this point a bit? Thank you.
> > > >
> > > > I am interested in it because, provided that the transformation cost between GNME and GNP is negligible, then with a lower bound for WCSC problems in terms of GNP convergence (if any), then solving WCSC in terms of GNME should be the same difficult as GNP. So I may conclude that the reason why existing literature with GNP measurement have worse results just stems from analysis technique, rather than a systematic advantage of GNME over GNP.

---

### Meta-Review · Area_Chair_JMig · 2022-08-25

**Recommendation:** Accept
**Confidence:** Less certain

**Metareview:**

The authors propose a new algorithm that attains the state-of-the-art rates for an important non-convex concave problem template, which includes distributionally robust learning problems as a special case.

The main contribution of the work is the new proof technique that goes beyond [42] to obtain the new rates. However, the supplementary material (49 pages!) that contains the proof is quite messy and it is difficult to verify it.

On one hand, the appendix needs to be completely re-written so that one can verify the full proof. On the other hand,  the authors provided a proof sketch that makes sense during the rebuttal.  The numerical demonstrations do support the authors' case. As a result, after discussions with the SAC, the AC decided to give the authors the benefit of the doubt.

We advise the authors to completely clean up the proof in the appendix for camera ready and make it accessible.

**Award:**

No

---

### Decision · Program_Chairs · 2022-09-14

Accept